# The translational landscape of HIV-1 infected cells reveals key gene regulatory principles

Anuja Kibe [1], Stefan Buck [1,2], Anne-Sophie Gribling-Burrer [1,3], Orian Gilmer[1], Patrick Bohn [1], Tatyana Koch[1], Chiara Noemi-Marie Mireisz[3], Andreas Schlosser[4], Florian Erhard[2,5], Redmond P. Smyth [1,3] & Neva Caliskan [1,6] ✉

Human immunodeficiency virus-1 (HIV-1) uses a number of strategies to modulate viral and host gene expression during its life cycle. To characterize the transcriptional and translational landscape of HIV-1 infected cells, we used a combination of ribosome profiling, disome sequencing and RNA sequencing. We show that HIV-1 messenger RNAs are efficiently translated at all stages of infection, despite evidence for a substantial decrease in the translational efficiency of host genes that are implicated in host cell translation. Our data identify upstream open reading frames in the HIV-1 5′-untranslated region as well as internal open reading frames in the *Vif* and *Pol* coding domains. We also observed ribosomal collisions in *Gag-Pol* upstream of the ribosome frameshift site that we attributed to an RNA structural fold using RNA structural probing and functional analysis. Antisense oligonucleotides designed to alter the base of this structure decreased frameshift efficiency. Overall, our data highlight the complexity of HIV-1 gene regulation and provide a key resource for decoding of host–pathogen interactions upon HIV-1 infection. Furthermore, we provide evidence for a RNA structural fold including the frameshift site that could serve as a target for antiviral therapy.

HIV-1 is a complex retrovirus that synthesizes multiple proteins through tightly regulated post-transcriptional and translational mechanisms. Following HIV-1 entry, reverse transcription and integration, the resulting proviral DNA is transcribed by host RNA polymerase II. The ensuing retroviral pre-mRNAs undergo alternative splicing to produce three classes of viral mRNAs: (1) unspliced full-length transcripts encoding structural and enzymatic polyproteins, Gag and Gag-Pol; (2) partially spliced transcripts yielding envelope and accessory proteins Env, Vif, Vpr and Vpu; and (3) fully spliced transcripts expressing regulatory and accessory proteins Rev, Tat and Nef[1–3]. Translation of unspliced and partially spliced products is temporally regulated through the Rev

protein, which binds to the Rev response element structure to facilitate their nuclear export[4,5]. HIV-1 mRNAs are capped, polyadenylated and rely on canonical cap-dependent translation, although under certain conditions, noncanonical mechanisms are also used[6–8]. For example, hypermethylation of the $m^7G$ cap in unspliced and partially spliced HIV-1 mRNAs may enhance translation of late-stage proteins[9,10]. Internal ribosome entry sites (IRESs) in the 5′-untranslated region (UTR) of HIV-1 RNAs as well as the coding region of Gag have also been proposed[11–19], with 5′-UTR IRES activity reportedly increasing under cellular stress and later stages of infection[20,21]. Recently upstream open reading frame (uORF)-mediated non-AUG translation events were also reported in

[1]Helmholtz Institute for RNA-based Infection Research, Helmholtz Centre for Infection Research (HIRI-HZI), Würzburg, Germany. [2]Faculty of Informatics and Data Science, University of Regensburg, Regensburg, Germany. [3]Institute of Molecular and Cellular Biology (CNRS), UPR 9002, University of Strasbourg, Strasbourg, France. [4]Rudolf Virchow Center for Experimental Biomedicine, University of Würzburg, Würzburg, Germany. [5]Institute for Virology and Immunobiology, University of Würzburg, Würzburg, Germany. [6]Faculty of Biology and Preclinical Medicine, University of Regensburg, Regensburg, Germany. ✉e-mail: neva.caliskan@ur.de

HIV-1, further demonstrating the virus's ability to exploit diverse translation strategies[22].

Translation of the full-length HIV-1 transcript is also regulated at the elongation step. The Gag-Pol polyprotein, which comprises viral enzymes such as protease, reverse transcriptase and integrase, is translated through programmed −1 ribosomal frameshifting (−1FS)[23]. −1FS occurs with 5–10% efficiency and relies on two conserved elements: a slippery sequence ((SS) UUUUUUA) and a downstream RNA structure, generally accepted to be a stem loop[23–25]. These elements ensure the correct expression ratio of Gag and Gag-Pol, critical for viral replication[26]. Although individual aspects of HIV-1 gene expression—such as alternative splicing, noncanonical initiation and programmed frameshifting—have been studied extensively, the global translational landscape of host and viral mRNAs during infection remains poorly understood[3,27–32].

In this work, we combined ribosome (Ribo-seq) and disome (Disome-seq) profiling with RNA sequencing (RNA-seq) from cytoplasmic extracts of HIV-1 infected T cells to comprehensively analyze host and viral translation during infection. Our results reveal that certain host, but not viral, transcripts were blocked during translation initiation. Ribo-seq detected extensive non-AUG translation initiation sites on the HIV-1 5′-UTR and internal open reading frames (iORFs) in the *Vif* and *Pol* genes. In addition, we observed notable ribosome stalling upstream of the canonical frameshift motif. Disome-seq revealed colliding ribosomes at this site, attributed to an extended RNA structure validated by structural probing and functional analysis. The extended frameshift RNA fold is crucial for maintaining frameshifting efficiency (FE), and targeting it with antisense oligonucleotides (ASOs) reduced frameshifting by ~40%. Our work sheds light on the translational dynamics of both host and viral mRNAs during HIV-1 infection and describes an RNA structural fold that regulates viral gene expression and represents a potential target for antiviral therapy.

## Results

### Altered host transcriptome and translatome during infection

To investigate the translational landscape of HIV-1 infected cells we infected human SupT1 cells, a T cell lymphoblastic lymphoma-derived cell line, with vesicular stomatitis virus glycoprotein pseudotyped HIV-1 NL4-3 Gag-iGFP ΔEnv virus, which allows only a single round of infection with HIV-1 (Fig. 1a). Flow cytometry confirmed green fluorescent protein (GFP) expression in more than 80% of cells 24 h post infection (hpi) (Extended Data Fig. 1a). Lysates from mock and HIV-1 infected cells were processed at 8, 16 and 24 hpi for RNA-seq and Ribo-seq analysis with duplicate experiments showing strong reproducibility (Fig. 1a and Extended Data Fig. 1e,f). Length distribution ribosome-protected fragments (RPF) were in the expected range peaking at 30 nucleotides (Extended Data Fig. 1b)[33–35]. To determine the reading frame distribution and predict the most likely P-site of every RPF, the probabilistic analysis pipeline (PRICE) was used[36]. Our metagene analysis revealed that the majority of RPF that map to coding sequences (CDS) are in the 0 frame, indicating efficient nuclease digestion (Extended Data Fig. 1c). Also, notable enrichment of reads was observed in the annotated coding regions, with only a small percentage of the RPFs mapping to 5′-UTRs (Extended Data Fig. 1d).

At 8 hpi, ~0.6% of RNA-seq reads derived from HIV-1 mRNAs, whereas ribosome-associated viral RNA was ~0.05%. By 16 hpi and 24 hpi, HIV-1 reads increased to ~1% and ~6% for RNA-seq and to ~0.5% and ~2.5% for Ribo-seq, respectively (Extended Data Fig. 1g). To examine host transcriptional and translational changes, we performed an integrative analysis of the Ribo-seq and RNA-seq at each time point with respect to the uninfected mock sample (Fig. 1b)[37]. At 8 hpi, 1,245 genes were differentially expressed in either RNA-seq, Ribo-seq or both, with 258 (~20%) showing proportional changes in RNA and translation levels, indicating no change in translational efficiency (TE) (depicted in blue) (Fig. 1b). However, ~60% (787 genes), demonstrated changes in translation as revealed by Ribo-seq (depicted in red), without corresponding

RNA-level changes, reflecting a shift in TE. At 16 and 24 hpi, most changes (75%) were proportional between RNA and translation, suggesting transcriptional regulation dominates at later stages, with fewer genes showing changes in TE (~5% (16 hpi) and ~12% (24 hpi)) (Fig. 1b).

Next, we conducted Gene Ontology (GO) analyses of significantly upregulated and downregulated genes at each time point based on Ribo-seq data, categorizing the top 15 pathways by fold changes into broader umbrella terms (Supplementary Table 1 and Methods). This revealed an increase in pathways related to cholesterol metabolism, cell motility, signal transduction and immune response (Supplementary Table 1 and Fig. 1c). Cholesterol metabolism, critical for viral entry, is likely stimulated by the viral Nef protein[38–42]. Early infection downregulated apoptotic and energy metabolism pathways, consistent with Nef-mediated prevention of apoptosis[43], whereas later stages showed suppression of translation-related genes, including those involved in ribosomal RNA processing and ribosome assembly. This suppression prompted further investigation of TE changes across all genes (Fig. 1c and Supplementary Table 1). Among these, *PCM1* gene was consistently upregulated at all time points, where the encoded protein was reported to interact with long noncoding RNAs to modulate interferon levels, although its role in HIV-1 infection remains unclear[44]. Immune response genes, including *HELLS* and *TOP2B*, as well as stress-response genes *SMC4* and *SMCDH1*, where the encoded proteins are Tat interactors, showed persistent TE upregulation[45]. Interestingly, at 8 hpi, *ATF4*, whose encoded protein is a key regulator of cellular adaptive stress response and HIV-1 regulation, was translationally downregulated, although its TE normalized at later stages[46,47]. Moreover, TE of 14 translation-associated genes, including genes encoding for elongation factors and ribosomal proteins like *RPL35*, *RPS21*, *RPS8* and *RPS14*, consistently decreased across all time points (Extended Data Fig. 2a).

Overall, these results highlight extensive host transcriptional and translational reprogramming upon HIV-1 infection, particularly affecting cellular stress, rRNA processing and translation.

### HIV-1 suppresses host translation at the initiation level

Because our data indicated disruption of translational processes, we examined the impact of HIV-1 infection on global host translation by analyzing RNA profiles from uninfected and infected cell lysates loaded on sucrose gradients at different time points (Fig. 2a). At 16 and 24 hpi, monosomes (80S) increased, whereas polysomes decreased, leading to lower polysome-to-monosome ratios than at 8 hpi, indicating slight, but measurable inhibition of translation initiation at later stages (Fig. 2b and Extended Data Fig. 2b). At 8 hpi, the polysome-to-monosome ratio did not change compared with the mock in both replicates, suggesting that the translation initiation inhibition occurs at later stages of infection (Fig. 2b and Extended Data Fig. 2b).

We then analyzed the distribution of specific RNAs in gradient fractions using a real-time quantitative polymerase chain reaction (qPCR) and calculated their polysome-to-monosome ratios (Extended Data Fig. 2 and Methods). Interestingly, we observed a difference in the distribution of host and viral RNAs. Host mRNAs (*GAPDH*, *ACTB*) were enriched in polysomes at 8 hpi, indicating active translation, but shifted to monosomes by 16 and 24 hpi, reflecting reduced translation (Fig. 2c and Extended Data Fig. 2c,d). By contrast, HIV-1 mRNAs were actively translated throughout infection. Partially spliced isoforms showed transient changes at 16 hpi but recovered by 24 hpi, whereas unspliced RNAs redistributed to lighter fractions at 24 hpi (Fig. 2d and Extended Data Fig. 2d). The increased availability of untranslated unspliced RNA could provide genomes for assembly into viral particles, because active translation of unspliced RNA is proposed to inhibit viral packaging[48]. Moreover, TEs of host mRNAs (*GAPDH*, *ACTB*) decreased over time, whereas TE of viral mRNAs (*Gag*, *Pol*) increased, peaking at 16 hpi (Extended Data Fig. 2e). Together, these analyses demonstrate a gradual decline in host global translation initiation upon HIV-1 infection, possibly because of decreased translation as a

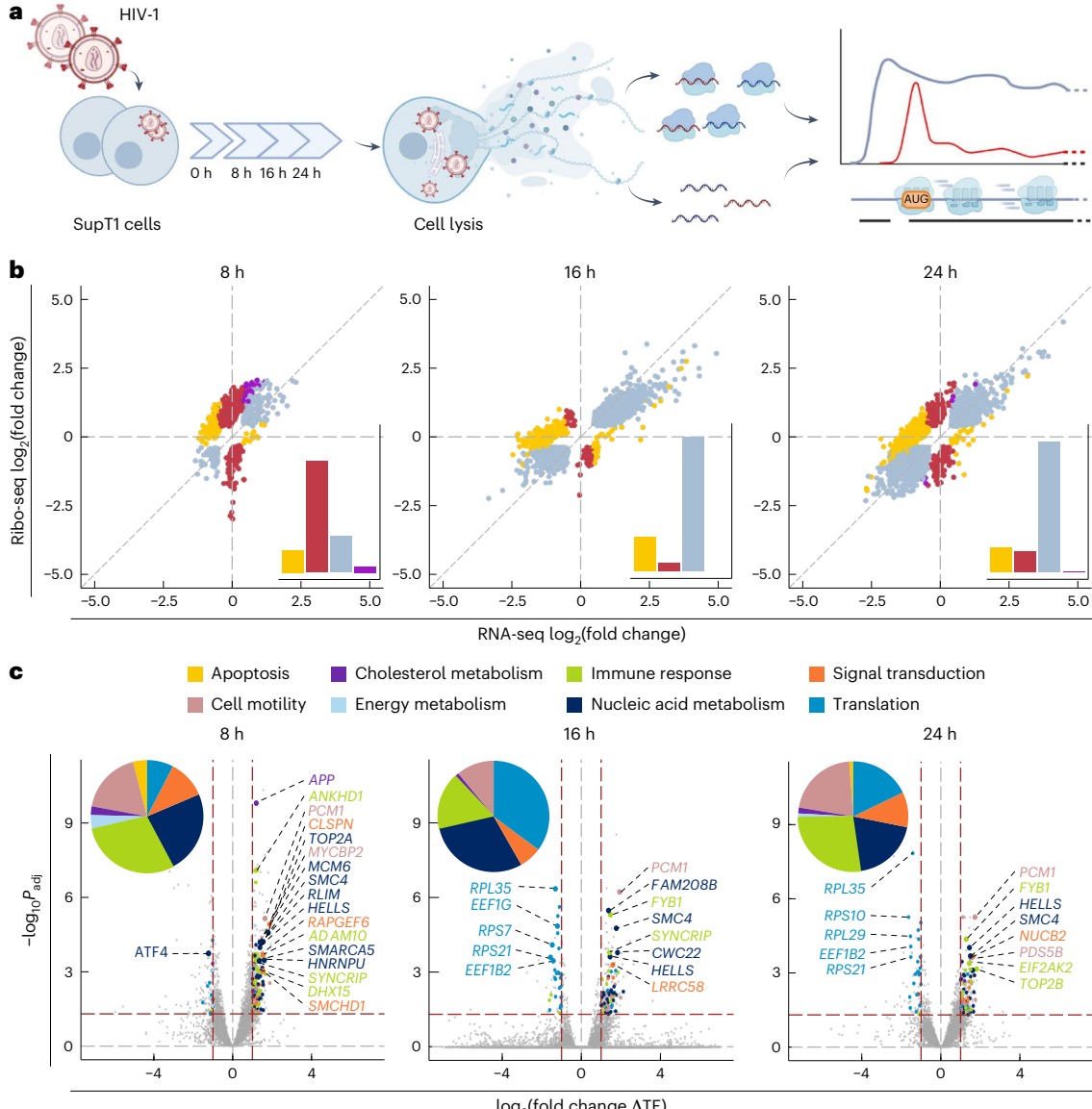

**Fig. 1 | Global transcriptional and translational changes in HIV-1 infected cells. a**, Schematic representation of the procedure used to monitor transcript abundance and translation in HIV-1 infected cells. Briefly, SupT1 cells were infected or not with HIV-1-iGFP (NL4.3 strain). At 0, 8, 16 and 24 hpi, cells were lysed to recover the cytoplasmic fraction and prepared Ribo-seq and RNA-seq libraries were subjected to high-throughput sequencing. **b**, Scatterplots of the log$_2$(fold change) in cytoplasmic RNA-seq and Ribo-seq levels of HIV-1 infected cells compared with the mock sample (8 h uninfected cells) at each time point of infection. Only genes that were called to be significantly differentially expressed in the deltaTE pipeline (8 hpi, $n = 1,254$; 16 hpi, $n = 1,312$; 24 hp.: $n = 2,257$) are shown ($P_{adj} < 0.05$, Wald's test $P$ value Benjamini–Hochberg corrected). Genes are colored based on fold changes in Ribo-seq, RNA-seq and TE (yellow: significant change in TE that counteracts the change in RNA, buffering the effect of transcription; red: significant change in Ribo-seq, with no change in RNA-seq

leading to change in TE; blue: significant change in RNA and Ribo at the same rate, with no change in TE; purple: significant change in TE that acts with the effect of transcription). Inset: bar plots of percentage of differentially expressed genes colored as described above. **c**, Volcano plots of differential TE of HIV-1 infected cells compared with the uninfected sample at each time point of infection. $P_{adj}$ (Wald's test $P$ value Benjamini–Hochberg corrected) values calculated by DESeq2 in the deltaTE pipeline. Genes with a log$_2$(fold change) > 1 in TE are colored based on GO terms (biological process) grouped as umbrella terms (Methods), and genes with a log$_2$(fold change) > 1.2 and −log$_{10}P_{adj}$ > 3.15 in TE are labeled. Inset: pie charts representing the umbrella terms of differentially regulated genes colored in **c** at the respective time point. Each slice of the pie chart represents the percentage of genes in the particular term. See also Extended Data Fig. 1. Illustration in **a** created using BioRender.com.

cellular stress response. However, HIV-1 ensures its own mRNA evades this suppression, resulting in increased translation efficiencies as the infection progresses.

**Codon-resolved analysis of stalling on host transcriptome**

Given the observed suppression of host translation by HIV-1, we investigated the potential mechanisms underlying this effect, including alterations of the codon usage and cellular transfer RNA pools induced

by HIV-1 infection. To accurately identify and quantify stalling events on individual codons, we implemented a new algorithm as illustrated in Extended Data Fig. 3a and described in Methods.

A-site stalling was observed on Ile (I) codons, increasing slightly (0.25–0.34 log$_2$(fold change)) at later infection stages, whereas Ala (A) and Asn (N) codons showed consistent stalling throughout (Extended Data Fig. 3b). P-site stalling was mostly observed at Asp (D) codons throughout infection, which was reported to be a common stalling

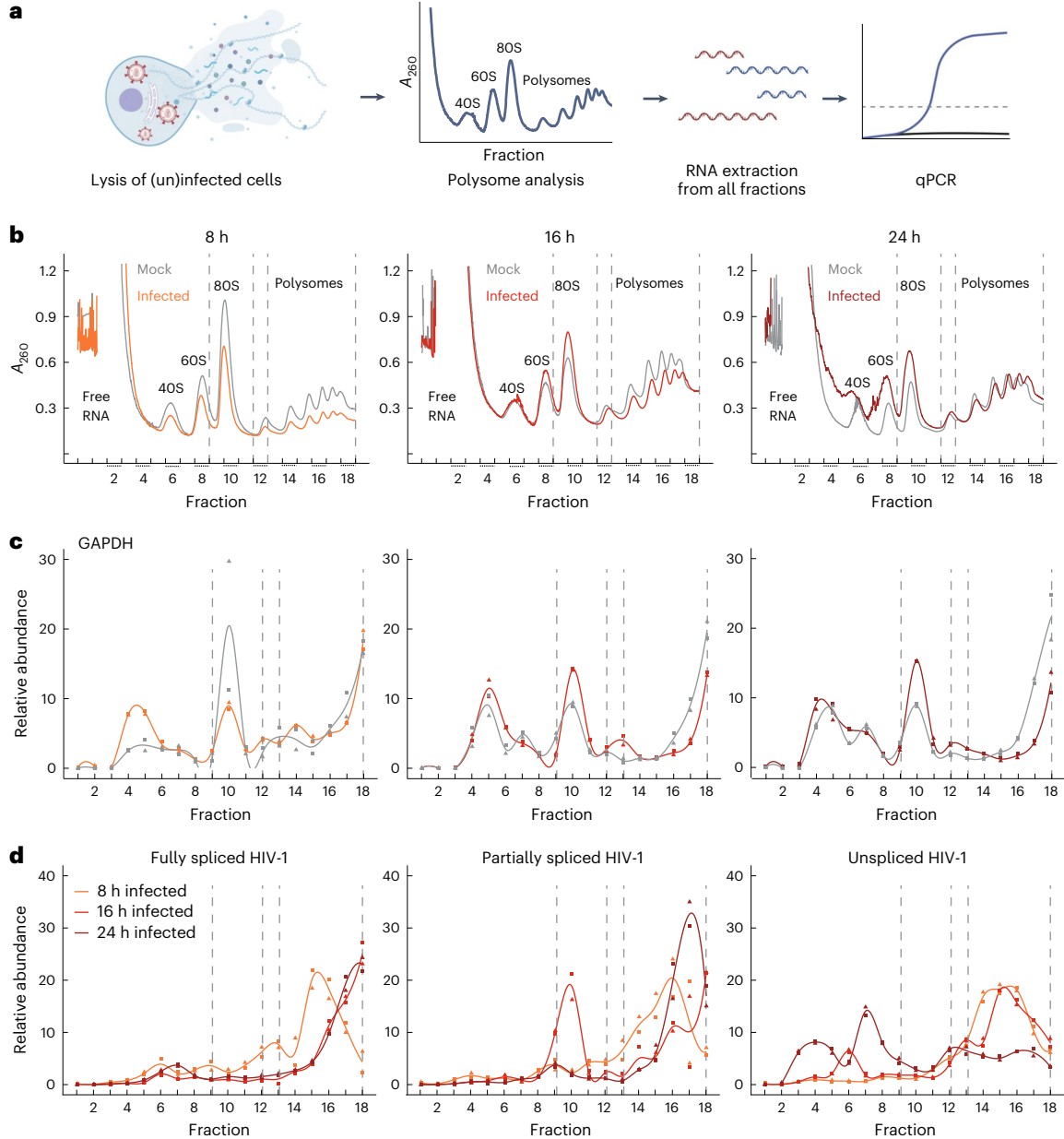

**Fig. 2 | Effect of HIV-1 infection on global translation profiles of host and viral RNAs. a**, Schematic representation of the experiment to monitor the effect of HIV-1 infection on overall translation using polysome profiling and qPCR analysis. Briefly, cytoplasmic lysates of uninfected and/or infected SupT1 cells at different time points of infection were sedimented through 5–45% sucrose gradients, and fractions were collected while continuously monitoring absorbance at $\lambda = 260$ nm. RNA was isolated from each of these fractions for subsequent qPCR analysis of specific genes. **b**, Polysome profile of infected cells compared with the mock sample at each time point of infection ($n = 2$; representative profile of one replicate is shown). **c,d**, Relative abundance of GAPDH (**c**) and HIV-1 (**d**) RNAs along the polysome profile at each time point of infection ($n = 2$). The gray line represents mock control (uninfected cells) at the respective time point. The depicted curve represents a smoothed interpolation derived from mean values of two qPCR replicates and both replicate values are shown as points. Dashed lines mark the fractions containing 80S (9–12) and polysome (13–18). See also Extended Data Fig. 2. Illustration in **a** created using BioRender.com.

site across species, including humans (Extended Data Fig. 3c)[49]. These data indicate that, whereas P-site stalling is unaffected on the course of infection, A-site stalling at the Ile codon awaits further investigation.

**Changes in HIV-1 translation patterns during infection**

Having characterized the impact of HIV-1 infection on host translation and ribosome stalling, we proceeded to investigate the translation patterns of viral-specific ORFs based on codon and frame-resolved RPF signatures. As described earlier, viral RNA was consistently detected throughout the infection process, and ribosome footprints were seen on all canonical viral CDS at later time points (Fig. 3a and Extended Data Fig. 1g). High ribosome density was observed in the 5′-UTR, likely because of its shared usage among splice isoforms and potential ribosome pausing in the highly structured 5′-UTRs of the HIV-1 transcripts, which were also shown across species in Ribo-seq datasets[50] (Fig. 3a). Dense clustering of RPFs in overlapping HIV-1 gene regions was analyzed using PRICE to determine the most likely translated protein. At 8 hpi, the number of reads was insufficient to map to each HIV-1 gene. At 16 hpi, Ribo-seq reads primarily corresponded to the 'early' regulatory protein Rev, translated from fully spliced HIV-1 transcripts (Fig. 3b). Partially spliced transcripts showed high Vpu expression at 16 and 24 hpi, supporting its role in enhancing progeny virion release at later

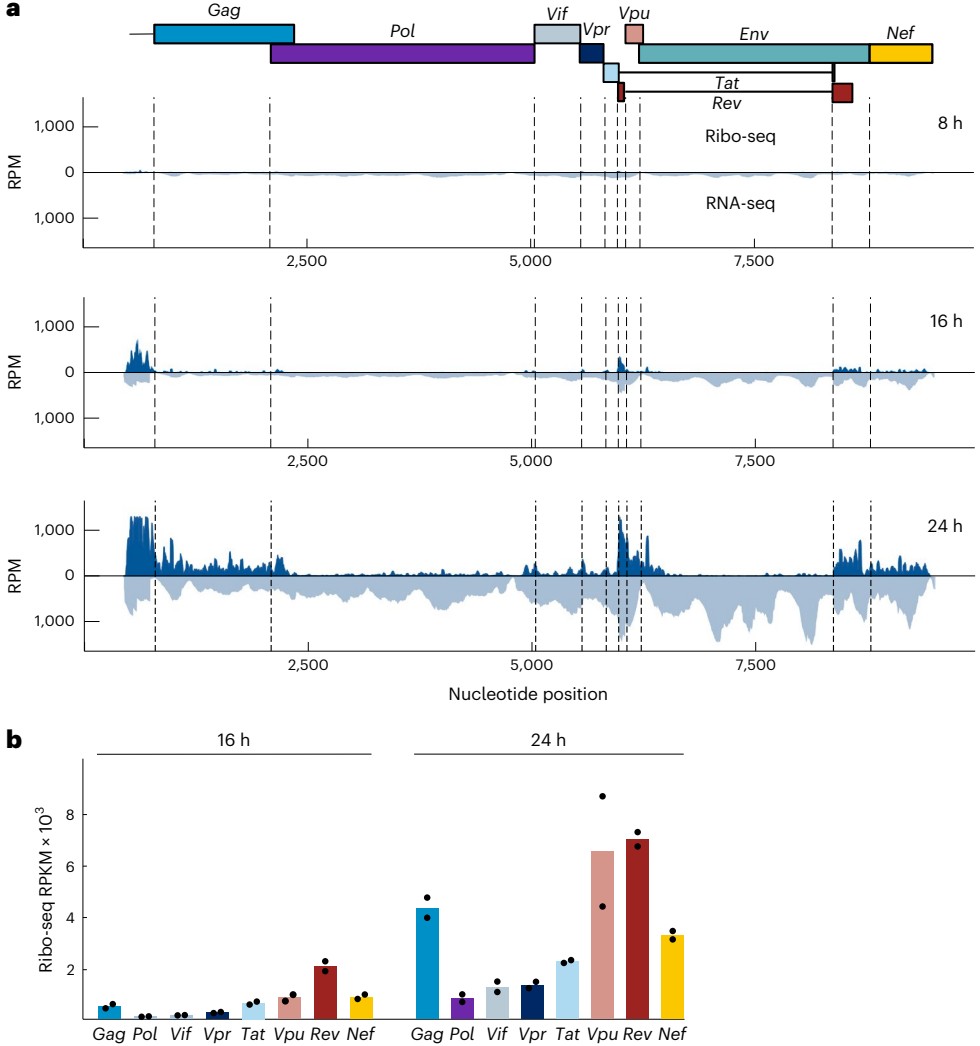

**Fig. 3 | Change in the HIV-1 translational landscape upon infection. a**, Ribo-seq and RNA-seq depth profiles at different time points upon infection. Canonical start codons of viral transcripts are depicted by dashed lines. **b**, Read counts normalized to reads per kilobase per million mapped reads (RPKM), mapping to each HIV gene in Ribo-seq at 16 and 24 hpi. Dots represent individual values from replicates. $n$ = 2 independent experiments. RPM, reads per million.

stages[51]. From 16 to 24 h, we marked the largest increase (almost tenfold) in the HIV-1 reads corresponding to *Gag*, translated from unspliced transcripts, reflecting its critical role in viral assembly (Fig. 3b).

### Discovery of iORFs in *Pol* and *Vif* genes

Because the majority of the HIV-1 RPFs originate from the 5′-UTR, we investigated translation initiation sites, performing Ribo-seq with har-ringtonine, which accumulates ribosomes at initiation sites. Quality control confirmed successful experiments, with the majority of RPFs mapping to host CDS in the 0 frame with a length of 29–30 nucleotides (Extended Data Fig. 4a–c). Numerous short ORFs were detected in the host genome (Supplementary Table 2), including known regula-tory uORFs, such as those for *MDM2* gene translation, validating our approach (Extended Data Fig. 4d). In the HIV-1 genome, prominent peaks in the 5′-UTR were associated with near-cognate AUG codons, indicative of noncanonical initiation, giving rise to −1, 0 or +1 frame uORFs encoding for peptides of ~3–42 amino acids (Fig. 4a,b). Among these, three uORFs were validated by PRICE ($P < 0.05$) (Fig. 4b (high-lighted in red) and Supplementary Table 2).

In addition, two hitherto unknown HIV-1 iORFs were identified in the *Pol* and the *Vif* coding regions (PRICE $P < 0.005$) (Fig. 4c,d and Sup-plementary Table 2). The *Pol* iORF is found in the noncoding exon 2, 166 nucleotides upstream of the canonical *Pol* ORF. It starts with a UUG and could generate different peptides ranging from 23 to 48 amino acids depending on splicing events from the D2 donor splice site (position 4,962) to different acceptor sites namely A2, A3, A4a/b/c or A5 (Fig. 4c and Extended Data Fig. 5a–c). Sequence alignment showed universal conservation of the UUG start codon, D2 splice site and UGA stop codon across 4,903 HIV-1 strains (Extended Data Fig. 4e). Consistent Ribo-seq reads at D2–A3 and D2–A5 junctions support translation of this iORF (Supplementary Table 3).

The *Vif* iORF, located in noncoding exon 3, starts with a GUG in the +1 reading frame relative to *Vif* and codes for nine amino acids. Located downstream of the A2 acceptor site, this iORF involves the D3 donor splice site, which enables generation of multiple fusion peptides via splicing to A3, A4a/b or A5 sites (Fig. 4d and Extended Data Fig. 5a,c). Similar to the *Pol* iORF, the start codon, splice site and stop codon are universally conserved for the *Vif* iORF (Extended Data Fig. 4e). Next, to confirm *Vif* iORF translation, we used a fluorescence reporter assay in which mCherry is produced only if translation begins at the GUG start codon of the *Vif* iORF. As a control, mCherry was placed in the canonical *Vif* frame (Methods and Extended Data Fig. 4f,g). Through this setup, we observed ~25% of mCherry expression relative to the control, confirm-ing *Vif* iORF translation in cells (Extended Data Fig. 4f,g).

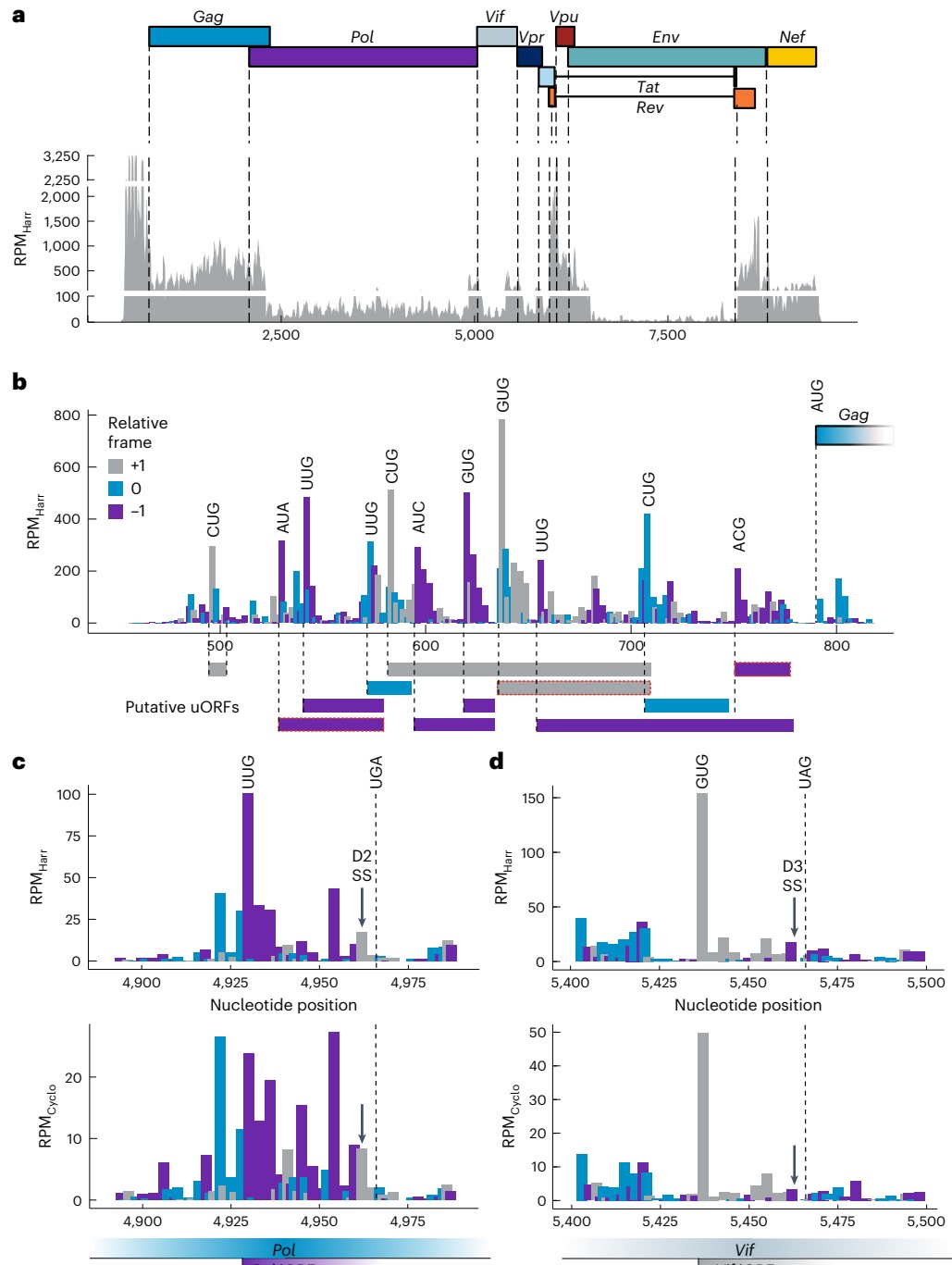

**Fig. 4 | Translation initiation sites in viral transcripts. a**, HIV-1 translational landscape of harringtonine-treated samples at 24 hpi. P-site codons were predicted with PRICE. Canonical start codons of viral transcripts are depicted by dashed lines. **b**, Close-up view of the HIV-1 5′-UTR showing the distribution of ribosome P-sites and the position of putative non-AUG start codons. Potential uORFs are shown below. uORFs predicted by PRICE ($P < 0.05$) are highlighted in red. Codon bars are colored according to the relative frame (*Gag* ORF is in frame 0). *P* values are calculated in the PRICE pipeline using standard parameters

(Methods)[136]. **c**, Potential iORF in the viral *Vif* gene (*Vif* ORF is in frame 0) predicted by PRICE. The iORF encodes for nine amino acids. However, the splice donor site D2 is located just before the annotated stop codon. **d**, Potential iORF in the viral *Pol* gene (*Pol* ORF is in frame 0) predicted by PRICE. The iORF encodes for twelve amino acids. However, the splice donor site D3 is located just before the annotated stop codon. RPM$_{Cyclo}$, reads per million cycloheximide-treated samples; RPM$_{Harr}$, reads per million harringtonine-treated samples. See also Extended Data Figs. 4 and 5.

Intriguingly, splicing events in transcripts (D1–A1/D1–A2) may position UUG/GUG initiation codons of the *Pol* or *Vif* iORFs upstream of canonical AUG codons, allowing ribosome recruitment in the 5′-UTR (Extended Data Fig. 5c,d). Translation of these iORFs across D2–A4a/b/c or D3–A4a/b/c sites could express an as-yet undescribed reading frame in the exon coding for Tat and Rev (Extended Data Fig. 5e). MS analysis

detected a peptide derived from the *Pol* iORF (SSSEQSDSSSFSIK), validating its translation. In addition, a second peptide was detected from the *Env* ORF, where, for biosafety reasons, translation from the canonical *Env* ORF is excluded in the HIV-1 strain used (Supplementary Table 4 and Extended Data Fig. 5f,g). Long-read nanopore sequencing identified Rev4-5 transcripts as likely sources for *Pol* iORF and Rev7-12

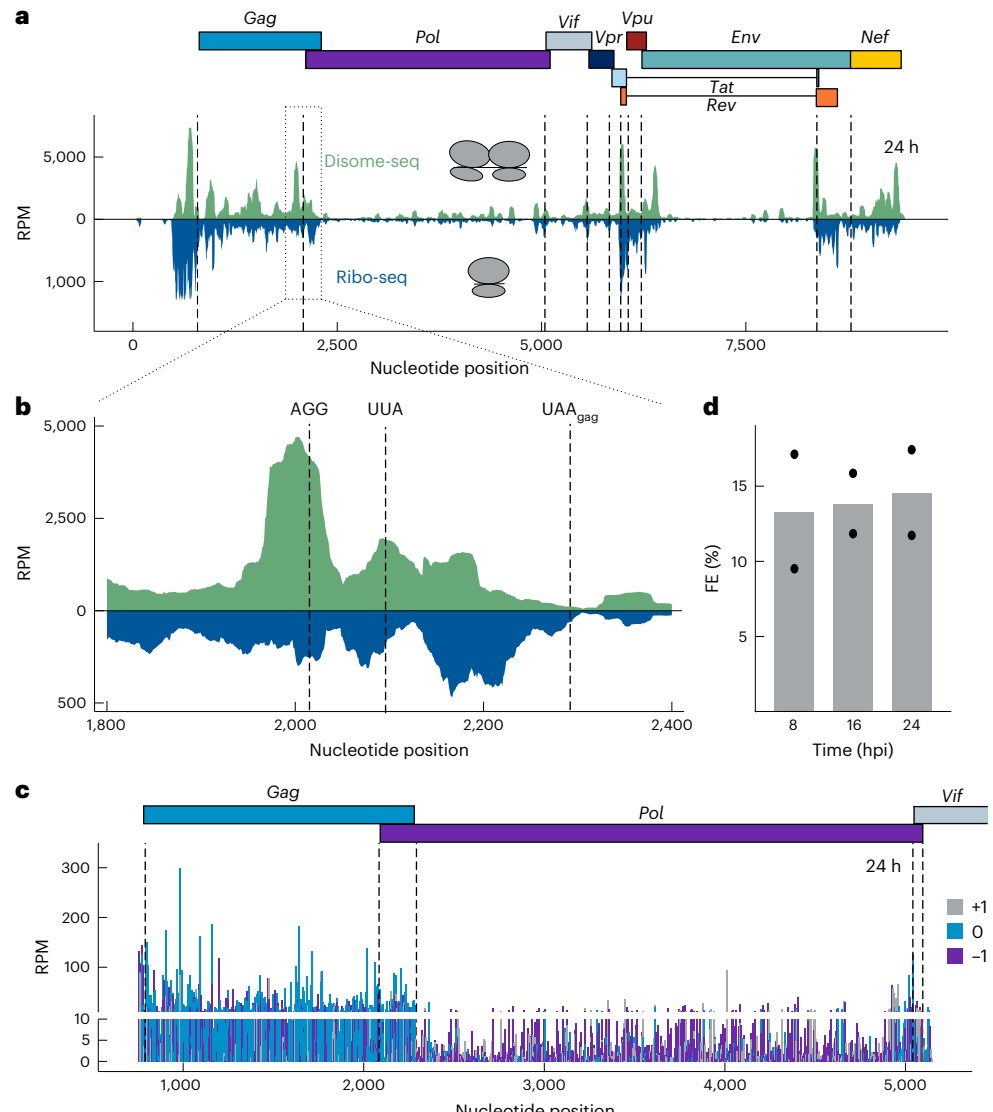

**Fig. 5 | Investigation of pause sites near the HIV-1 −1 frameshift site by ribosome and disome profiling. a**, Disome-seq (green) and Ribo-seq (blue) depth profiles of HIV-1 at 24 hpi. The frameshift region is boxed. **b**, Zoom image of the HIV-1 frameshift region at the interface of the *Gag* and *Pol* genes. **c**, Translational landscape of the *Gag/Pol* gene region. P-site codons were predicted with PRICE. P-site codon bars are colored according to the relative frame (*Gag* ORF is in frame 0). The *y* axis is cut at 10 reads per million. **d**, Percentage of ribosome FE at each time point of infection. Dots represent individual values from replicates. *n* = 2 independent experiments. See also Extended Data Fig. 6.

for *Vif* iORF, placing these iORFs upstream of canonical Rev start codons (Supplementary Table 5). Notably, the only alternative source of the peptide would be the transcript Gp41 2 containing the cryptic D2b–A5 splice junction (Extended Data Fig. 5h). This was ruled out with the RPF data, where there are 5 and 11 overlapping reads for the D3–A4a and D3–A4b splice sites respectively, compared with only a single read overlapping the D2b–A5 splice site, further supporting iORF translation as the primary source (Supplementary Table 3). Future studies should explore the mechanisms of near-cognate start codon initiation and the functional roles of iORF translation products in HIV-1 infection.

**Ribosome stacking upstream of the HIV-1 frameshift site**

In addition to the identified noncanonical iORFs, HIV-1 is already known to use noncanonical translation events such as the −1FS, achieved through the means of a frameshift stimulatory site (FSS) comprising an SS and an RNA secondary structure, spanning nucleotides 2,084–2,130 at the *Gag-Pol* overlapping gene. Past studies using this frameshift element in vitro indicated ribosomal pausing over the P-site UUA codon of

the slippery site (UUUUUA), although exactly where the RNA secondary structure begins is still under debate[27,28,52,53].

Motivated by previous work highlighting the importance of ribosome collisions for viral frameshifting[54], we complemented our analysis by Disome-seq on mock and infected samples, by selecting 50–80-nucleotide protected fragments[54–57]. In Disome-seq, read lengths were seen to be broader than in classical Ribo-seq with peaks of around 54–57 nucleotides and 60–63 nucleotides, a range consistent with the expected RPF for disomes, as reported previously (Extended Data Fig. 6b)[55,58]. Similar to our Ribo-seq experiments, the replicates of Disome-seq showed high consistency (Extended Data Fig. 6a). The Disome-seq read depth profile mapping to HIV-1 showed distinct stalling peaks that might be crucial for correct cotranslational folding of the proteins (Fig. 5a)[58]. Specifically in *Gag-Pol*, there was a notable accumulation of disomes approximately 70 nucleotides upstream of the canonical programmed ribosome frameshift SS (Fig. 5a,b). Beyond the *Gag*_UAA 0 frame stop codon, both monosome and disome coverages decreased drastically, because only −1 frame ribosomes

continue translation beyond this site (Fig. 5c). Next, using the ratio of ribosome footprints in the *Pol* and *Gag* CDS, we estimated the FE of HIV-1 in infected T cells. Approximately, 12–17% of the *Gag-Pol* coverage was observed at *Pol* at both 16 and 24 hpi, slightly higher than reported in previous in vitro studies[23]. FE also remained constant throughout the course of infection (Fig. 5d). We analyzed the disome read length distribution near the FSS, observing that disome reads ending between nucleotides 1,990 and 2,050, located upstream of the FSS, peaked at 60–64 nucleotides. This peak suggests 'true' disomes representing ribosomal stacking or collisions[57]. At and beyond the FSS (2,050–2,110), we observed a similar length distribution. However, we see a larger fraction of shorter footprints with a median length of approximately 54–57 nucleotides, which could be attributed to ribosome quality control intermediates[59]. Despite the similar distribution, the reduced disome reads imply less ribosome stacking at and beyond FSS (consistent Ribo-seq reads at these positions) (Fig. 5b and Extended Data Fig. 6c).

### RNA fold at Gag-Pol frameshift site finetunes gene expression

To investigate ribosome stalling upstream of the *Gag-Pol* frameshift site, we performed codon-resolved analysis using the Ribo-seq data. Ribosomes showed minor pausing at the P-site UUA (Leu) of the slippery site and a prominent pause at the AGG (Arg) codon 74 nucleotides upstream, consistent with disome observations (Figs. 5b and 6a). Alignment of 4,903 HIV-1 wild-type (WT) sequences confirmed conservation of both the canonical SS and the amino acid sequence Pro–Arg–Lys–Lys (PRKK), encoded by the -CCUAGG/AAAAAG/A-nucleotide sequence, at the pause site (Fig. 6a, lower). To validate the pause sites, we performed ribosome pausing assays in vitro in rabbit reticulocyte, using reporter mRNAs mimicking the native HIV-1 genomic context lysates[60,61] (Methods and Extended Data Fig. 6d,e). A transient pause was observed at the AGG codon, suggesting temporary ribosome stacking. Interestingly, we noted a persistent protein product near the slippery site in vitro, indicating a potential ribosome drop-off at this position (Extended Data Fig. 6d,e).

To understand the structural basis of the ribosome stalling event, we performed dimethyl sulfate (DMS) probing in HIV-1 infected SupT1 cells. This analysis revealed an extended structure of the HIV-1 frameshift site, which broadly agrees with the in vitro probed structure reported in ref. 27. The extended RNA structure is predicted to fold in a three-way junction with a large central bulge. We find good support for the anchoring stem (nucleotides 2,022–2,031 with 2,152–2,161) and the conventional stem loop (nucleotides 2,099–2,110 with 2,115–2,126). However, neither the alternate lower stem (nucleotides 2,037–2,039 with 2,146–2,148) described in ref. 27, nor the lower part of the conventional stem loop (nucleotides 2,092–2,096 with 2,130–2,134) are well supported, indicating that these stems may not be folded in the majority of molecules in their native, dynamic state in the cell (Fig. 6b).

To assess the functional relevance of extended RNA fold, we used our frameshift reporter assay in HEK293 cells[62]. The extended fold showed a FE of 5.4 ± 0.1% (Fig. 6c). Deleting the fold (Δext), leaving only the conventional SS and stem loop, reduced FE by ~30% (3.7 ± 0.5%), whereas targeting the lower stem with ASO1 similarly reduced FE by ~40% compared with nontargeting (Fig. 6c). The AGG pause site is proximal to nucleotides AAAAAG, which could be a putative slippery site (SS*) (Fig. 6a). Frameshifting at SS* could lead to premature termination at a downstream stop codon. Mutating SS* (SS*$_{mut}$) resulted in a slight decrease in FE (4.2 ± 0.8%). However, mutating both the canonical SS and the −1 frame stop codons still allowed for ~16% FE relative to WT (FE = 0.9 ± 0.4), indicating potential −1FS at SS*. All in all, these results confirm the functional importance of the extended RNA fold, and the existence of a low efficiency alternative frameshift event (FS*). This supports the notion of an additional regulatory layer for Gag-Pol expression to be explored in future studies.

## Discussion

In this work, we performed complementary high-resolution genomic and molecular analysis to elucidate translational dynamics of viral and host RNAs during HIV-1 infection. Our study shows that the initial host response is translationally regulated. Despite downregulation of host translation, HIV-1 mRNAs are efficiently translated at all stages. We identified uORFs in the HIV-1 5′-UTR and iORFs in the *Vif* and *Pol* coding domains, along with ribosome collisions upstream of the frameshift site, linked to a regulatory RNA structure in *Gag-Pol*. These findings highlight new regulatory elements in the HIV-1 genome that finetune viral gene expression.

One of the striking observations of this study is that translation of specific mRNAs encoding for ribosomal proteins, including *RPS17*, *RPS21*, *RPS14*, *RPS8*, *RPL34* and *RPL11*, was consistently downregulated, with *RPL13a*—encoding for a ribosomal protein implicated in antiviral responses—showed reduced translation at 16 and 24 hpi[63]. Such regulation suggests heterogeneity in ribosome composition may favor HIV-1 RNA translation over host RNAs, as proposed for other viruses[60,61,64].

Furthermore, our results demonstrate that HIV-1 suppresses host translation initiation, while evading this effect through alternative mechanisms. In our study, elevated RPF densities at the 5′-UTR aligns well with previously reported IRES-mediated ribosome recruitment as well as alternative initiation from non-AUG codons. Such uORFs are extensively used by other viruses as well as eukaryotes[65–67]. Perhaps, peptides translated from these alternative short uORFs could modulate the TE of HIV-1 *Gag*. For instance, they may inhibit translation of the main ORF by restricting the fraction of initiating and/or scanning 40S ribosomal subunits that can reach the canonical start codon[67–69]. It is also proposed that cryptic translation products of the HIV-1 5′-UTR could generate new antigens that can be recognized by the immune responses[22]. Although proteomic detection of such peptides is challenging because of low abundance or instability, their role in immune recognition warrants further study, because all translation products can serve as antigens, even if they do not accumulate in cells[70]. Indeed, supporting this, a recent study also reported the presence of uORFs in HIV-1, although the reported peptides differ from those discovered in our study[22]. However, a caveat of Ribo-seq is that reads from the viral 5′-UTR may arise from non-RPF footprints because of highly structured regions or protein interactions resistant to enzymatic degradation. Further studies are needed to explore viral uORFs and ribosome-induced structural changes in the 5′-UTR, which may regulate the switch from early to late HIV-1 translation.

In addition to uORFs, another important finding was the alternative initiation in *Vif* and *Pol* giving rise to iORFs, confirmed through proteomics and long-read sequencing. These peptides, formed by a combination of splicing and alternative reading frames, may modulate canonical protein translation, as reported previously for other viruses[71,72]. Moreover, fusion proteins, which would contain the *Vif* iORF at the N-terminal or truncated versions of canonical proteins can also modulate localization of the protein in the cell, and thus affect protein functions. Together with novel splice isoforms, these iORFs expand the regulatory potential of HIV-1, presenting intriguing targets for future studies.

Our distinctive approach with disome analysis revealed notable ribosome stacking at the 3′-end of *Gag* upstream of the *Gag-Pol* frameshift site, a departure from other viruses such as SARS-CoV-2 and TMEV in which pausing occurs at the frameshift site itself[54,73]. Further analysis of the RNA element in the vicinity of the pause site pointed to the presence of a three-way junction that included the frameshift stem loop. We also validated the presence of the alternative structure using RNA structure probing in infected cells, which somewhat resembles the predicted three-helix junction proposed in a previous in vitro study[27]. Here, we suggest several scenarios on how this pause can regulate Gag and Pol levels during the HIV-1 life cycle. First, the pause may be caused by ribosome collisions, which can trigger ribosome-mediated

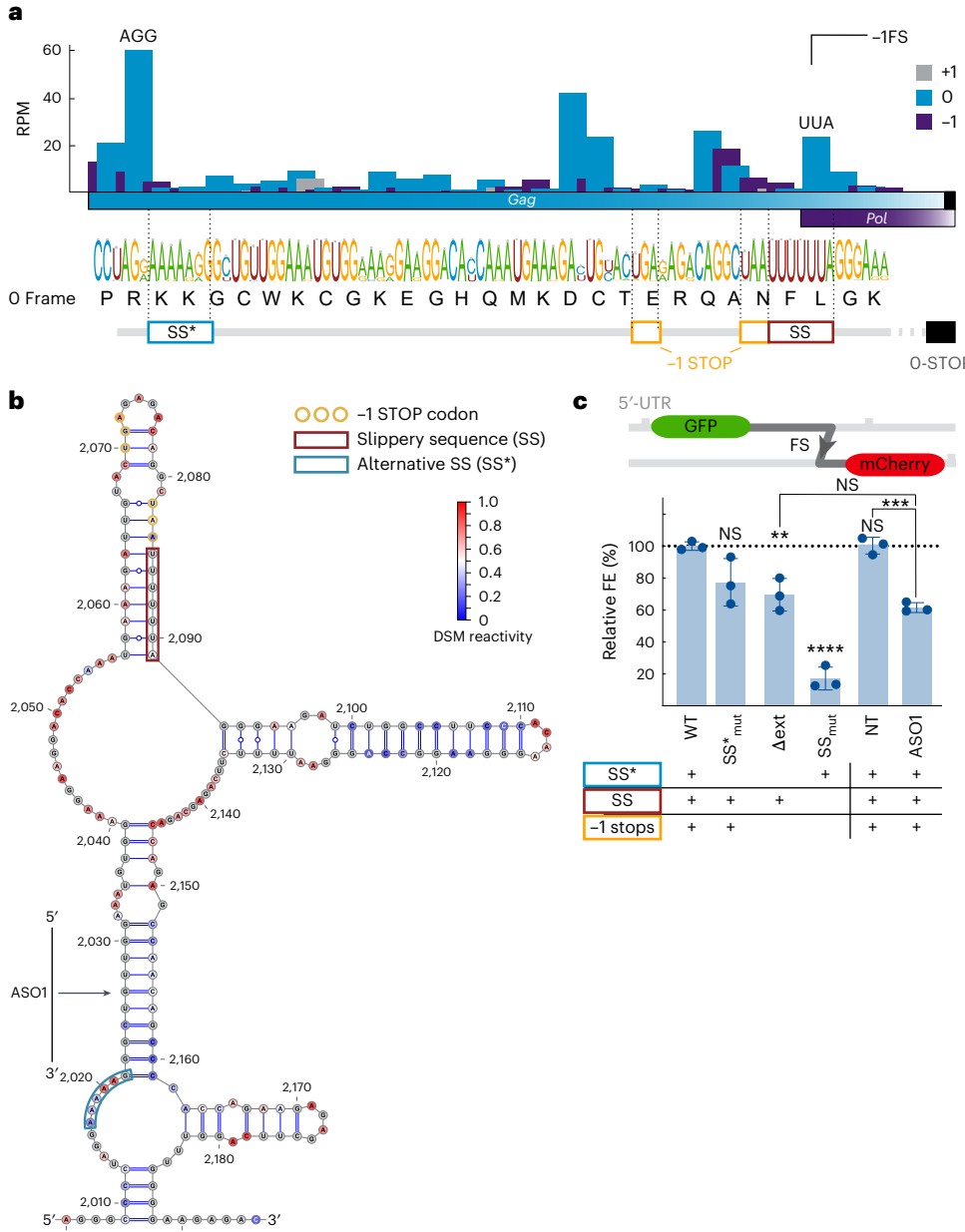

**Fig. 6 | Experimental and functional analysis of extended HIV-1 frameshift site.**
**a**, Translational landscape surrounding the HIV-1 frameshift site (FS) at 24 hpi (mean of both replicates). P-site codons were predicted using PRICE analysis. P-site codon bars are colored according to their relative frame (*Gag* ORF is in the 0 frame). Codons of the highest peak upstream of the SS (AGG) and in the SS (UUA) are denoted. The sequence logo of 4,903 HIV sequences is shown with the amino acids sequence associated with the 0 frame. Below these sequences, the potential alternative frameshift SS (SS*), the −1 frame stop codons and the canonical SS are shown in blue, yellow and red boxes, respectively. **b**, RNA structure model of the region surrounding the HIV-1 frameshift site, from nucleotides 1,551 to 1,741, as predicted by DMS data. The DMS reactive nucleotides are colored red, whereas unreactive nucleotides are blue. Gray nucleotides correspond to those without DMS data. The ASO1 binding site is shown. **c**, Schematic representation of the dual-fluorescence frameshift reporter constructs and quantification of

the relative FE of different HIV-1 frameshift site mutants. EGFP and mCherry are separated by both a self-cleaving 2A peptide and a stop codon in-frame with the EGFP. As a result, 0 frame translation produces only EGFP, whereas −1FS produces both EGFP and mCherry. The ratio of mCherry to EGFP fluorescence is used to quantify the FE. Each mutant characteristic is highlighted below the bar graph. Included is a control in which both SS and SS* are mutated to define the baseline mCherry signal, with subsequent data adjusted for this baseline. HIV-1 Δext corresponds to the structure containing nucleotides 1,629 to 1,672. The nontargeting (NT) ASO and ASO1 were used against the WT HIV-1 frameshift site. Data points represent the mean ± s.d. ($n = 3$ independent experiments). *P* values were calculated using an ordinary unpaired one-sided analysis of variance comparing every mutant FE to the WT (Exact *P* values: WT versus Δext, $P = 0.008$; SSmut, $P < 0.0001$; NT versus ASO1, $P = 0.005$). NS, not significant.

quality control mechanisms leading to ribosomal drop-off and potential release of the nascent peptide[74]. Second, we also marked an alternative slippery site positioned near the pause site, inducing low levels of frameshifting. Translation through this alternative slippery site would terminate before the canonical HIV-1 frameshift site. Interestingly, both these scenarios would lead to a truncated Gag product. The Gag variant

would lack the p6 Gag part of the Pr55 Gag precursor, which contains the late domain required for efficient viral budding. It has been shown to be essential for incorporating Vpr and Pol into the virions[75-77]. Hence, the product may either simply be degraded or produce a functional Gag variant that could be important to regulating budding, Vpr levels or Pol protein in the virions.

ASO-targeting of the lower stem of the alternative fold of the HIV-1 RNA decreased FE, pointing to a functional role for the upstream RNA of the canonical FSS. The extended HIV-1 RNA fold can regulate ribosome speed and density at the canonical frameshift site. In that case, the upstream sequence may promote ribosome stacking, facilitating the pause over the canonical SS. Pausing may also cause ribosomal traffic jams upstream of the canonical frameshift site to ensure that elongating ribosomes are sufficiently distanced allowing the frameshift RNA hairpin to fold between consecutive rounds of translation elongation. In line with that, a model was previously presented in which changes in the rate of initiation affected the propensity of frameshifting by altering the distance between elongating ribosomes on the mRNA, which influences the frequency of encounters between these ribosomes and the frameshift stimulatory signal[21]. Detailed studies are warranted to unpack the precise mechanics of this pause, its interrelation with HIV-1 frameshifting, HIV-1 Gag-Pol protein levels and its broader implications for the HIV-1 life cycle. All in all, ribosome stacking upstream of the frameshift site may represent an additional layer of regulation to maintain the critical ratio of structural and enzymatic proteins during HIV-1 infections.

Taken together, our work enhances understanding of post-transcriptional regulation of both host and viral mRNAs during HIV-1 infection. Although certain aspects of the complex translational control process of HIV-1 remain to be further investigated, our work presents new paradigms of noncanonical mechanisms in viral gene expression and paving the way for new antiviral strategies.

## Online content

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

## Methods

### Cell culture
HEK293 cells (gift of J. Vogel, Helmholtz Institute for RNA-based Infection Research Helmholtz Center for Infection Research (HIRI-HZI)) were maintained in DMEM (Gibco, catalog no. 41966052) supplemented with 10% FBS (Gibco, catalog no. 10500064), 100 µg ml⁻¹ streptomycin and 100 U ml⁻¹ penicillin (Gibco, catalog no. 15140122). SupT1 cells (National Institutes of Health (NIH) HIV Reagent Program, National Institute of Allergy and Infectious Diseases: Sup-T1 Cells, ARP-100, contributed by D. Ablashi) were maintained in RPMI (Gibco, catalog no. 11875093) supplemented 10% FBS, 1× L-glutamine (Gibco, catalog no. 25030024), 100 µg ml⁻¹ streptomycin and 100 U ml⁻¹ penicillin. Cell lines were maintained at 37 °C with 5% $CO_2$.

### HIV-1 virus production and infection
HEK293T cells were seeded in a 15-cm² plate with 15 ml of media 24 h before transfection. Cells were then cotransfected with 10 µg of pNL4-3 ΔEnv-iGFP (NIH AIDS Research and Reference Reagent program, catalog no. 11100) and 2 µg of pCMV VsVg (gift from R. Smyth, HIRI-HZI) using polyethylenimine (Polysciences Europe, catalog no. 24765-1). At 48 h post transfection, supernatant was collected and centrifuged at 2,000g for 20 min to remove cell debris. The supernatant was filtered and loaded onto the 20% sucrose cushion (50 mM Tris–HCl (Carl Roth, catalog no. HN95.1) pH 7.4, 100 mM NaCl (Carl Roth, catalog no. 9265.5), 0.5 mM EDTA (Sigma-Aldrich, catalog no. 46931320001)) at a ratio 4:1 and centrifuged at 2,853g for 20 h. The virus pellet was resuspended in phosphate-buffered saline (PBS), treated with DNase I (Th. Geyer, catalog no. 11824306) for 1 h at 37 °C, distributed into microfuge tubes and kept at −80 °C until further use.

For ribosome, disome and polysome profiling experiments, SupT1 cells were either mock treated (PBS; Sigma, catalog no. P3813) or infected with 50 µl of virus suspension per 40 million cells by spinoculation at 1,500g for 30 min at 37 °C, in the presence of 8 µg ml⁻¹ polybrene (Merck, catalog no. TR-1003-G). Cells were pelleted for 3 min at 37 °C, washed once with warm media and resuspended in 40 ml of RPMI medium.

### Ribosome profiling and RNA-seq
Ribosome profiling samples were prepared according to the protocol outlined by McGlincy and Ingolia[78], with modifications for suspension cell lines. At each time point of infection, cells were treated with 100 µg ml⁻¹ cycloheximide (VWR, catalog no. A0879) in DMSO (Carl Roth, catalog no. A994.2) at 37 °C for 5 min. For harringtonine samples, cells were treated with 2 µg ml⁻¹ harringtonine in DMSO at 37 °C for 5 min, followed by cycloheximide. Cells were immediately pelleted at 1,000 rpm for 3 min, washed once with cycloheximide containing PBS and the cell pellet was snap-frozen in liquid nitrogen and stored at −80 °C until further use. Cell pellets were thawed in lysis buffer (20 mM Tris–Cl pH 7.4, 150 mM NaCl, 5 mM MgCl₂ (Carl Roth, catalog no. 2189.1), 1 mM DTT (Biomol, catalog no. D8070.10), 100 µg ml⁻¹ cycloheximide, 1% Triton-X (Merck, catalog no. T8787), 25 U ml⁻¹ Turbo DNase (Invitrogen, catalog no. AM2238)) and triturated ten times through a 26G needle. The lysate was cleared with centrifugation for 10 min at 4 °C and the supernatant was recovered. rRNA contamination was removed using riboPOOLs rRNA depletion kit (siTOOLs Biotech, catalog no. dp-K024-000042) following the manufacturer's instructions.

For RNA-seq, total RNA was isolated from the cell lysate using TRIzol LS (Invitrogen, catalog no. 10296010), following the manufacturer's instructions. RNA-seq libraries from these samples were constructed using CORALL Total RNA-seq Library Prep Kit (Lexogen, catalog no. 183-184), according to the manufacturer's instructions. The quality of the library was assessed by using a BioAnalyzer via the High Sensitivity DNA Kit (Agilent, catalog no. 5067-4626). Sequencing experiments were performed by Core Unit Systems Medicine (University of Würzburg) with an Illumina NextSeq 500 Mid-output SE150 instrument or with an Illumina NextSeq 2000 P3 SE100 instrument.

### Polysome and qPCR analysis
Mock-treated and infected SupT1 cells at respective time points (40 million cells per sample, two replicates per time point for both mock and infected) were treated with 100 µg ml⁻¹ cycloheximide in DMSO at 37 °C for 5 min. Cells were immediately pelleted at 1,000 rpm for 3 min, washed once with cycloheximide containing PBS and the cell pellet was lysed in ice-cold lysis buffer. Cell pellets were lysed with 500 µl of lysis buffer, and the lysate was clarified by centrifugation at 170,000g for 10 min at 4 °C. Polysome buffer (20 mM Tris–HCl pH 7.4, 150 mM NaCl, 5 mM MgCl₂, 1 mM DTT, 100 µg ml⁻¹ cycloheximide) was used to prepare all sucrose solutions. Sucrose density gradients (5–45% w/v sucrose; VWR, catalog no. IB37160) were freshly prepared using a Gradient Master (BioComp Instruments) according to the manufacturer's instructions. The lysate was then applied to a 5–45% sucrose continuous gradient and centrifuged at 217,873g for 3 h, at 4 °C. The absorbance at 254 nm was monitored and recorded and 570-µl fractions were collected using a gradient collector (BioComp instruments). RNA was isolated from 300 µl of each fraction using TRIzol LS reagent according to the manufacturer's instructions. Next, the RNA was column purified using the NucleoSpin Gel and PCR Cleanup kit (Macherey-Nagel, catalog no. 740609.25) and reverse transcription was done using SuperScript IV reverse transcriptase (Invitrogen, catalog no. 18090010) primed by random hexamers. qPCR reactions were set-up using POWER SYBR green Master Mix (Invitrogen, catalog no. A25777) according to the manufacturer's instructions and were analyzed using the CFX96 Touch Real-Time PCR Detection System (Bio-Rad). To create the polysome as well as qPCR smooth profiles, an interpolation was made from the mean values, so that the final curve had 1,000 data points. The integrals were estimated from the interpolated splines by the trapezoidal rule, using the equation:

$$\sum_{i=1}^{n} (x_i - x_{i-1}) * \frac{(y_{i-1} + y_i)}{2}.$$

### Next Generation Sequencing analysis
For computational processing of the ribosome profiling and RNA-seq reads, PRICE pipeline was utilized[36]. In addition to performing all preprocessing and prefiltering steps, this pipeline determines the P-site codon for each individual Ribo-seq read with a probabilistic model and uses this information to statistically predict canonical as well as noncanonical ORFs. First, a JSON file specifying the path to the demultiplexed FASTQ sequencing data and mapping parameters was prepared. With the correctly prepared JSON file, bash scripts streamlining the pipeline were created as instructed (https://github.com/erhard-lab/price). Briefly, the main steps executed by the pipeline are as follows: adapter trimming using cutadapt (parameters: -a AGATCGGAAGAGCACACG -e 0.3); unique molecular identifier extraction by gedi FastqFilter (parameters: -D -layout 2UI5U151 -min 18); human rRNA and mycoplasma reads removal by mapping to specific libraries with bowtie2 (parameters: -p 8 --un un --local -x $INDEX -U); read mapping to the human- (GRCh38.p10, Ensembl 90) and HIV (accession AF324493) genome by STAR (parameters: --runMode alignReads --runThreadN 8 --alignIntronMax 1 --genomeDir $starindex --genomeLoad LoadAndKeep --limitBAMsortRAM 8000000000 --readFilesIN $fileIN --outSAMmode NoQS --outSAMtype BAM SortedByCoordinate --alignEndsType Extend5pOfRead1 --outSAMattributes nM MD NH); unique molecular identifier deduplication using gedi DedupUMI; P-site and ORF predictions using PRICE. Finally, the bash scripts were executed and the PRICE coverage data, a report folder with mapping statistics and BAM files with the mapped reads were generated by the pipeline.

For RNA-seq the preprocessing steps of the pipeline were manually adjusted. The linker sequences were trimmed and data were filtered using fastp. The RNA-seq reads were mapped to the human and HIV genome using STAR aligner.

## Differential gene expression analysis

For analysis of differential gene expression, the mapped reads were first assigned to annotated genomic features using the featureCounts tool[79]. Next, the deltaTE pipeline was utilized to perform statistical differential expression analysis on the featureCounts matrices between every infection time point to the mock sample (8 h uninfected SupT1 cells)[37]. The pipeline was ultimately executed as instructed (https://github.com/SGDDNB/translational_regulation). The count matrices, fold change tables, as well as some interactive plots can be found as supplementary files under the Gene Expression Omnibus accession GSE244468. For GO analysis the online tool of the Panther classification system was used[80]. GO analysis was performed with a custom background list and an over-representation test for 'GO biological process complete' was performed (Fisher's exact test and false discovery rate correction). Gene lists to be analyzed included significantly differentially expressed genes identified by deltaTE analysis. At each time point, two separate analyses for upregulated or downregulated genes, based on the Ribo-seq log$_2$(fold change), were conducted. Uploaded gene lists were provided with Ensembl identifiers. For each enrichment analysis, the top 15 enriched GO processes of hierarchy level 1, identified by their fold change, were classified into umbrella terms (Supplementary Table 1). The pie charts in Fig. 1c were generated using the number of genes with a TE log$_2$(fold change) greater than 1 or less than −1, belonging to a specific umbrella term at each infection time point.

## Bioinformatic analysis of ribosomal stalling patterns

To investigate ribosomal stalling, P-site codon information from PRICE was used. The PRICE probabilistic model determines the most likely P-site for each Ribo-seq read and here the summed coverage for every potential codon is referred to as a 'peak'. For the ribosomal stalling patterns, cycloheximide-treated Ribo-seq datasets were used. All codon coverage peaks were sorted into bins depending on the genomic distance to the next peak so that individual translational active regions or exons end up in a single bin. If the genomic distance between peaks exceeded 100 nucleotides, a new bin was created. Bins with fewer than ten entries were discarded. Next, human genome annotations were utilized to exclude the first five codons of annotated CDS. From the remaining bins, the highest peaks were considered potential stalling sites when the read count was at least two times higher than the mean of all peaks in the bin. Depending on the size of each bin, a different number of maximum peaks was selected, specifically the number of considered peaks was determined by floor division for bins with more than ten entries (number of max values = binsize // 10). For comparison, 1,000 random datasets were created by sampling 1,000 times the same number of random peaks from the previously sorted bins. Finally, the occurrence of every codon was counted and a frequency per 1,000 was derived. The log$_2$(fold change) of the codon frequency observed in the potential stalling sites compared with the mean codon frequency of the random subsamples was computed. The A-site codons downstream were also identified similarly.

## Plasmid construction

The reporter vector for ribosome pausing assays contained β-globin 5′- and 3′-UTRs as well as a 30-nucleotide poly(A) tail. The insert was derived from nucleotides 64 to 2,687 (Δ1,870–1,881) of the HIV-1 genome; a 3× FLAG-tag was introduced at the N terminus to facilitate detection. To generate controls for ribosome pausing, the −1FS site was mutated by disrupting the RNA fold as well as the SS for the 0 frame and FS frame controls or by adding stop codons after the AGG (for AGG$_{pause}$) and UAA (for SS$_{pause}$) codon.

To generate dual-fluorescence reporter constructs, frameshift sites of HIV-1 and corresponding mutant and/or truncated variants were placed between the CDS of enhanced green fluorescent protein (EGFP) and mCherry by site-directed mutagenesis in such a way that EGFP would be produced in the 0 frame and mCherry produced in the −1 frame[81]. EGFP and mCherry were separated by StopGo signals[81] as well as an alpha-helical linker[82]. A construct with no FS insert and mCherry in-frame with EGFP served as a 100% translation control and was used to normalize EGFP and mCherry intensities. A construct with mutated SS* or SS with EGFP in the 0 frame and mCherry in the −1 frame was used to subtract background frameshifting. Sequences of all plasmids and oligos used in this study are given in Supplementary Table 6.

## Ribosome pausing assay

mRNAs were in vitro transcribed using T7 polymerase purified in-house using linearized plasmid DNA as the template. RNAs were translated in nuclease-treated rabbit reticulocyte lysate (Promega, catalog no. A25777) programmed with ~50 μg ml$^{-1}$ template mRNA. A typical reaction mixture was composed of 90% (v/v) rabbit reticulocyte lysate, 20 μM amino acids (lacking methionine) and 0.2 MBq [$^{35}$S]-methionine (Hartmann Analytic, KSM-01ME, catalog no. 19810110). The translational inhibitor harringtonine (5 μg ml$^{-1}$) was added shortly after initiation for synchronization. The translation reaction was stopped by the addition of an equal volume of 10 mM EDTA and 100 μg ml$^{-1}$ RNase A (Invitrogen, catalog no. AM2271). Following incubation at room temperature for 20 min, 3 vol. of 2× Bolt LDS Sample Buffer (Invitrogen, catalog no. B0007) was added, and the samples were denatured at 70 °C for 10 min before loading on 15% sodium dodecyl sulfate–polyacrylamide gel electrophoresis gel. After polyacrylamide gel electrophoresis, gels were dried under vacuum, exposed to phosphor screens (Fujifilm) and scanned with a Typhoon FLA 7000 (GE Healthcare).

## Flow cytometry

HEK293 cells were transiently transfected using polyethylenimine (Polysciences Europe, catalog no. 24765-1) according to the manufacturer's instructions with either the control construct or the −1FS construct encoding for the dual-fluorescence EGFP-mCherry translation reporter as outlined in Fig. 6, gating strategy Supplementary Fig. 1. For cotransfections with ASO, plasmids and ASO (5′-ACATTTCCAACAGCCC-3′) were mixed. To increase the stability of the oligonucleotides, the phosphate backbone was replaced by phosphothioate and three bases at the 5′- and 3′-ends were locked nucleic acids. Cells were collected 24 h post transfection. After washing with PBS, flow cytometry was performed on a NovoCyte Quanteon (ACEA) instrument using NovoExpress software (v.1.6.2). FE was calculated according to the following formula:

$$FE\,(\%) = \frac{\frac{mCherry_{test}}{EGFP_{test}}}{\frac{mCherry_{control}}{EGFP_{control}}} \times 100$$

where mCherry is the mean −1 frame mCherry intensity, EGFP is the mean EGFP intensity, test represents the sample containing the frameshift element and control represents the in-frame control lacking the frameshift element where mCherry and EGFP are produced from the 0 reading frame. Data represent the results of at least three independent experiments. Data were analyzed and plotted in GraphPad Prism (v.9.2.0). For statistical analysis of FE in the presence of ASOs, an ordinary one-sided analysis of variance was followed by a Brown–Forsythe test to ensure equal variance among the samples. Finally, a Dunnett's multiple comparisons test was used to compare test samples to control constructs (nontargeting ASOs).

## DMS mutational profiling

For in vivo DMS modification of HIV-1 guide RNA, SupT1 cells were infected with VsVg pseudotyped HIV-1 (ΔEnv) particles as described above. At 24 h post infection, cells were probed with 85 mM DMS for 6 min at 37 °C, placed immediately on ice and the reaction was quenched by addition of 100 mM β-mercaptoethanol. Cells were pelleted by centrifugation for 5 min at 400g, 4 °C and washed with

PBS. The cell pellet was directly resuspended in 1 ml of TRI Reagent (Invitrogen, catalog no. AM9738) and total RNA was extracted according to the manufacturer's instructions. RNA was purified using NucleoSpin Gel and PCR Clean-Up, according to the manufacturer's instructions.

Reverse transcription was performed on purified RNA mixed with 0.5 mM deoxyribonucleotide triphosphate and 0.25 µM reverse transcription primer (5′-CTTGTCTATCGGCTCCTGCTTC-3′) in a final volume of 12 µl. The RNA was denatured at 65 °C for 5 min and placed on ice. The reverse transcription reaction was performed by addition of 1 mM MnCl$_2$, 1× MarathonRT buffer (50 mM Tris–HCl pH 8.3, 200 mM KCl, 20% glycerol), 0.2 µl of RNasin (Invitrogen, catalog no. AM2694) and 40 U of MarathonRT[93] in a final volume of 25 µl. The reactions were incubated for 8 h at 42 °C.

Newly synthesized complementary DNAs were diluted eight times and PCR amplified using Forward (5′-TCGTCGGCAGCGTCAGATGT GTATAAGAGACAGCTACCATAATGATACAGAAAGGC-3′) and Reverse (5′-GTCTCGTGGGCTCGGAGATGTGTATAAGAGACAGGTCTCTTCC CCAAACCTGAAG-3′) primers. Amplicon quality was checked on an agarose gel.

DNA amplicons were purified by solid-phase reversible immobilization using Mag-Bind TotalPure next generation sequencing beads (Omega Bio-tek, catalog no. M1378-00) following the manufacturer's instructions, with a DNA to beads ratio of 1:0.8. Nextera indexes were added by PCR, 80 ng of purified DNA was mixed with 1× Q5 Buffer, 0.2 mM dNTPs, 0.02 U of Q5 polymerase and 2.5 µl of Nextera Index (Illumina) in a 14-µl reaction. The quality of the libraries was assessed by agarose gel electrophoresis. Equal amounts of each sample were pooled followed by a final solid-phase reversible immobilization purification (DNA to beads ratio of 1:0.8). Paired-end PE150 sequencing was carried out on an Illumina NovaSeq instrument (Novogene). Data were trimmed using cutadapt v.1.18 and aligned to the reference sequence with bowtie2. Parameters for cutadapt were '--nextseq-trim 20 --max-n 0 -a atcctgcaacctgctcttcgcc -A gtagctgtcgagctcctgcgaag', and for bowtie2, '-D 20 -R 3 -N 1 -L 15 -i S,1,0.50'. Further analysis used RNA Framework v.2.7.2 modules rf-count ('-m') and rf-norm ('-rb AC -ni ---scoring-method 3 -norm-method 2') to calculate DMS reactivities by subtracting background mutations and normalizing using 90% wintering normalization. Structure predictions were carried out using RNA Framework rf-fold module ('-ow -dp -sh -g').

## Sample preparation for mass spectrometry

Cells were lysed in PBS containing 0.25% sodium deoxycholate (Sigma-Aldrich, catalog no. 302-95-4), 0.2 mM iodoacetamide (Sigma-Aldrich, catalog no. 144-48-9), 1 mM EDTA, 1× complete protease inhibitors (Roche, catalog no. 04693132001), 1 mM PMSF (Santa Cruz Biotechnology, catalog no. 329-98-6) and 1% n-octyl-beta-D-glucopyranoside (Santa Cruz Biotechnology, catalog no. 29836-26-8) at 4 °C for 1 h. Lysates were clarified by centrifugation at 16,000g for 20 min at 4 °C. Proteins were precipitated overnight at −20 °C with a fourfold volume of acetone and pellets were washed with acetone at −20 °C. Precipitated proteins were dissolved in 20 µl of NuPAGE LDS Sample Buffer (Thermo Fisher Scientific, catalog no. NP0008), reduced in 50 mM DTT for 10 min at 70 °C and alkylated with 120 mM iodoacetamide for 20 min at room temperature in the dark. Separation was performed on NuPAGE 4–12% Bis–Tris gels (Thermo Fisher Scientific, catalog no. WG1401A) with MOPS buffer. Gels were washed three times for 5 min with water, stained with Simply Blue Safe Stain (Thermo Fisher Scientific, catalog no. LC6060) for 1 h and finally washed with water for 1 h. Each gel lane was cut into 15 pieces. After destaining with 30% acetonitrile in 0.1 M NH$_4$HCO$_3$ (pH 8), the gel bands were shrunk with 100% acetonitrile and dried in a vacuum concentrator (Concentrator 5301; Eppendorf). Trypsin (Trypsin Gold; Promega, catalog no. V5280) digestion was performed overnight at 37 °C in 0.1 M NH$_4$HCO$_3$ (pH 8) with 0.1 µg of protease per gel band. The supernatant was removed and peptides were extracted from the gel bands with 5% formic acid.

## Nano-liquid chromatography with tandem mass spectrometry analysis

LC–MS/MS analyses were performed on an Orbitrap Fusion (Thermo Fisher Scientific) mass spectrometer equipped with a PicoView Ion Source (New Objective) and connected to an EASY-nLC 1000 (Thermo Fisher Scientific). Samples were loaded on a trapping column (2 cm × 150 µm ID; PepSep) and separated on a capillary column (30 cm × 150 µm ID; PepSep), both packed with 1.9 µm C$_{18}$ ReproSil. Peptides were separated with a flow rate of 500 nl min$^{-1}$ and a 30-min linear gradient from 3% to 30% acetonitrile and 0.1% formic acid.

Both MS and MS/MS scans were acquired in the Orbitrap analyzer with a resolution of 60,000 for MS scans and 30,000 for MS/MS scans. Higher-energy collisional dissociation fragmentation with 35% normalized collision energy was applied and a Top Speed data-dependent MS/MS method with a fixed cycle time of 3 s was used. Dynamic exclusion with a repeat count of 1 and an exclusion duration of 30 s was applied. Singly charged precursors were excluded from selection. The minimum signal threshold for precursor selection was set to 50,000. Predictive automatic gain control was used with a target automatic gain control value of $4 × 10^5$ for MS scans and $5 × 10^4$ for MS/MS scans. EASY-IC was used for internal calibration.

MS data were analyzed using PEAKS Studio Xpro (Bioinformatics Solutions). Raw data refinement was performed with the following settings: Merge Options, no merge; Precursor Options, corrected; Charge Options, 1–6; Filter Options, no filter; Process, true; Default, true; Associate Chimera, yes. De novo sequencing and database searching were performed with a Parent Mass Error Tolerance of 10 ppm. Fragment Mass Error Tolerance was set to 0.02 Da, and Enzyme was set to none. The following variable modifications were used: Oxidation (M), pyro-Glu from Q (N-term Q), acetylation (protein N-terminal) and carbamidomethylation (C). Data were searched against a fasta database concatenated from UniProt human reference proteome (UP000005640, release 2022-02, all variants) and a custom database containing translated ORF derived from Ribo-seq data (altogether 101,707 protein entries). Identified proteins and peptides were filtered to 1% false discovery rate.

## Nanopore sequencing

To analyze the HIV transcriptome in infected SupT1 cells at 8, 16 and 24 hpi, 30 µg of TRIzol-extracted RNA was treated with 3 µl of Turbo DNase in a 150-µl total volume for 30 min at 37 °C. RNA was purified using a Macherey-Nagel PCR Clean-up kit with NTC buffer according to the manufacturer's instructions and 1 µg of RNA was reverse transcribed with SuperScript IV using primer GTACAGGCAAAAA GCAGCTGCT according to the manufacturer's instructions in a 10-µl reaction. The cDNA was purified using a Macherey-Nagel PCR Clean-up kit using NTC buffer and eluted in 20 µl of H$_2$O. RNA was digested in a 10-µl reaction containing 3 µl of cDNA product with 0.5 µl of RNase A and 0.5 µl of RNase T1 (NEB, catalog no. EN0541) and incubated at 37 °C for 30 min, followed by column purification as described above. Next, PCR amplification of 20 µl of the RNase-digested, purified cDNA in a 100-µl reaction with Primestar GXL (Takara, catalog no. R050A) using primers GGTCTCTCTGGTTAGACCAGATCTGAG and GTACAG GCAAAAAGCAGCTGCT was performed with 12 cycles of denaturation at 98 °C for 10 s, annealing at 55 °C for 15 s and extension at 68 °C for 2 min. PCR products were purified with a 1:1 ratio of Mag-Bind TotalPure next generation sequencing beads according to the manufacturer's recommendations and eluted in 14 µl of H$_2$O.

Next, 80 ng of PCR product in 5 µl was end repaired by addition of 0.7 µl of NEBNext End Repair buffer and 0.3 µl of NEBNext End Repair Enzyme Mix (NEB, catalog no. E6050S), mixed thoroughly by pipetting and incubated for 5 min at 25 °C followed by 5 min at 65 °C. Nanopore native ligation barcoding barcodes (ONT SQK-NBD114.96) were ligated by transfer of 3.5 µl of end-repaired product into a new well, followed by addition of 1.5 µl of barcode and 5 µl of Blunt/TA Ligase

Master Mix (NEB). After incubation at 25 °C for 20 min, 1 µl of EDTA (SQK-NBD114.96) was added, samples were pooled and purified with 0.4 vol. of Ampure XP beads (SQK-NBD114.96). Barcoded DNA was eluted in 31 µl of H$_2$O. To ligate the motor protein, 30 µl of the barcoded DNA was mixed with 10 µl of 5× Quick Ligation Buffer (NEB, catalog no. B6058S), 5 µl of DNA and 5 µl of high-concentration T4 DNA ligase (NEB, catalog no. M0202S) was added. After incubation for 20 min at 25 °C, 0.4 vol. of Ampure XP beads (SQK-NBD114.96) were added. Beads were washed twice with 125 µl of Short Fragment Buffer (SQK-NBD114.96) and eluted in 14 µl of Elution Buffer (SQK-NBD114.96). DNA concentration was quantified with AccuClear Ultra High Sensitivity dsDNA Quantitation Solution (Biotium) and 10 fmol was prepared for sequencing on a R10.4.1 Flongle flow cell (FLO-FLG114) by addition of 15 µl of sequencing buffer and 10 µl of loading beads. Data were acquired on a MinION Mk1b with MinKNOW v.23.04.3.

### Nanopore sequencing data analysis
Acquired nanopore data were base called in 'sup' mode with Guppy v.6.5.7 and filtered for a minimal qscore of 10. Isoforms were identified with Isoquant v.3.3.0 using a GTF file containing all previously reported isoforms[1,83,84]. Only uniquely identified isoforms were counted. To identify those isoforms that may express the novel peptides, the nucleotide sequence of all reported isoforms was searched for the iORFs and/or first AUG and in silico translated using Biopython 1.79 in a custom python script.

### Reporting summary
Further information on research design is available in the Nature Portfolio Reporting Summary linked to this article.

### Data availability
High-throughput sequencing data have been submitted to Gene Expression Omnibus (GEO) and are available under the accession number GSE244468. The mass spectrometry proteomics data have been deposited to the ProteomeXchange Consortium via the PRIDE[85] partner repository with the dataset identifier PXD058232. Further information and requests for resources and reagents should be directed to and will be fulfilled by the lead contact neva.caliskan@helmholtz-hiri.de. Source data are provided with this paper.

### Code availability
Publicly available tools and packages have been used as described in the Methods. For downstream analysis and plotting, custom Python and R scripts have been used and are available upon reasonable request.

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

### Acknowledgements

We thank A. Sparmann for critical reading of the manuscript. We thank N. Guydosh and S. Meydan (NIH) for their helpful suggestions on disome sequencing. R.P.S is funded through Helmholtz Young Investigator Grant (VH-NG-1347) and the Centre for Structural Biology of HIV-1 RNA U54 AI170660. N.C. received funding from the Helmholtz Association and the European Research Council grant no. 948636.

### Author contributions

A.K. performed the ribosome/disome profiling and the reporter assays. A.K. and S.B. analyzed the Ribo-seq and Disome-seq data. T.K. optimized translation assays. A.-S.G.-B. performed the qPCR and dimethyl sulfate RNA structure profiling experiments. R.P.S. analyzed the RNA structure profiling data with the support of O.G. on data interpretation. P.B. performed and analyzed nanopore sequencing experiments. C.N.-M.M. and A.S. performed and analyzed the LC–MS/MS experiments. F.E., R.P.S. and N.C. supervised the study. A.K. and N.C. wrote and edited the paper with the contributions of all authors.

### Funding

### Competing interests
The authors declare no competing interests.

### Additional information
**Extended data** is available for this paper at https://doi.org/10.1038/s41594-024-01468-3.

**Correspondence and requests for materials** should be addressed to Neva Caliskan.

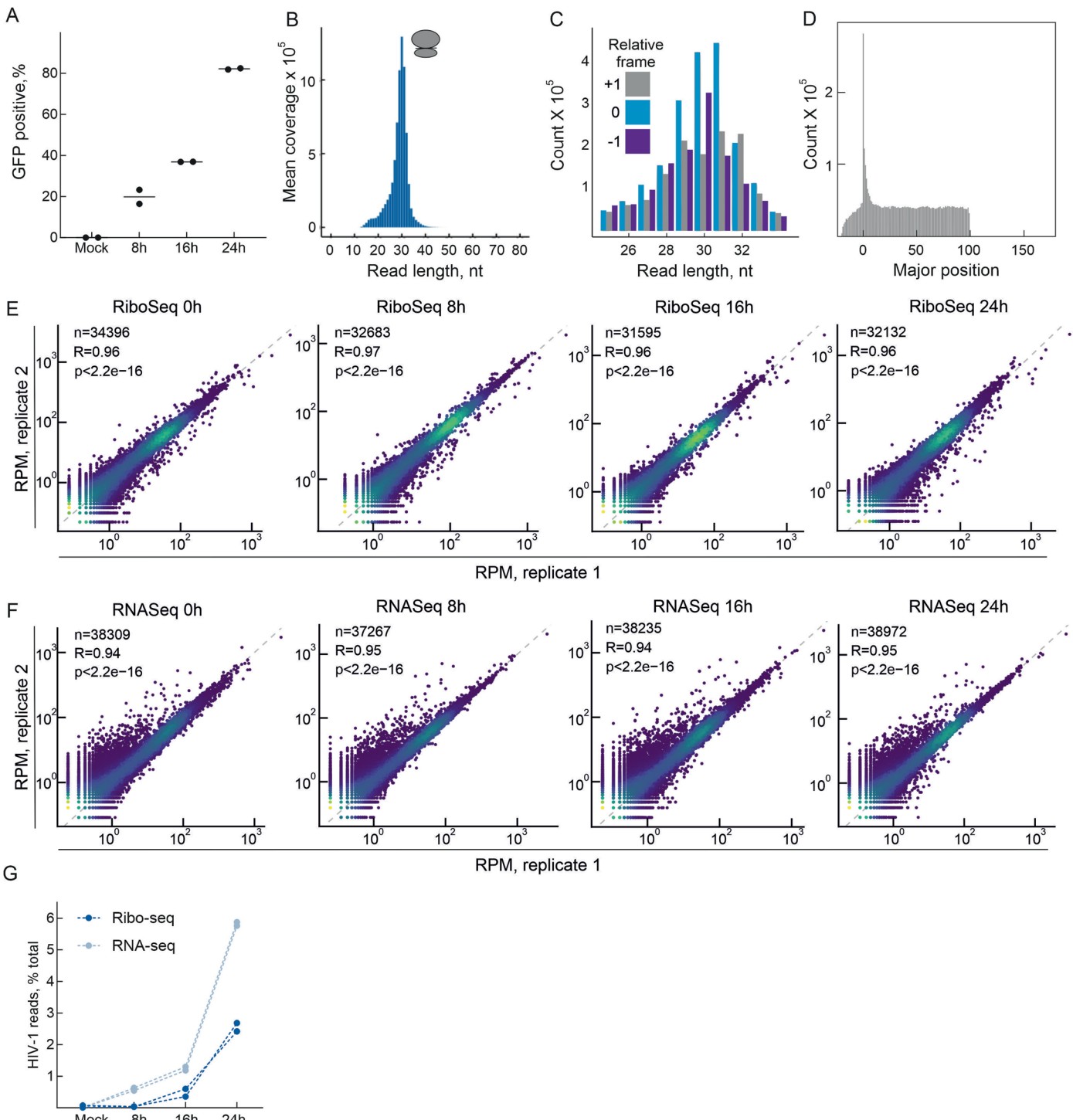

**Extended Data Fig. 1 | Analysis of RiboSeq reads in HIV-1-infected cells.**
(**a**) Flow cytometry-based detection of GFP- expressing SupT1 cells infected
with the HIV-iGFP virus. Dots represent individual values from replicates. $n = 2$
independent experiments. Read length (**b**) and relative frame distribution
(**c**) of all reads overlapping a CDS, obtained from cycloheximide-treated
ribosome profiling libraries prepared from RNaseI- treated SupT1 cell lysates.
(**d**) The number of reads starting within coding transcripts of one replicate.
The x axis is the relative position of the read start within the three regions
(5′-UTR: <0; CDS: 0-100; 3′-UTR: >100). The major isoform is used as reference

for each gene (the longest coding transcript). Scatterplots show the correlation
and reproducibility between duplicates of RiboSeq (**e**) and RNASeq (**f**). Reads
per million (RPM), as assigned to individual genomic features by featureCounts.
Number of reads (n), significance (p value) and Pearson correlation coefficients
(R) are indicated for each scatterplot. Higher point density depicted in blue-
green. (**g**) Proportions of total RNASeq and RiboSeq reads mapping to HIV-1 at
each timepoint of infection. Dots and lines correspond to individual values from
replicates. $n = 2$ independent experiments (RiboSeq experiments).

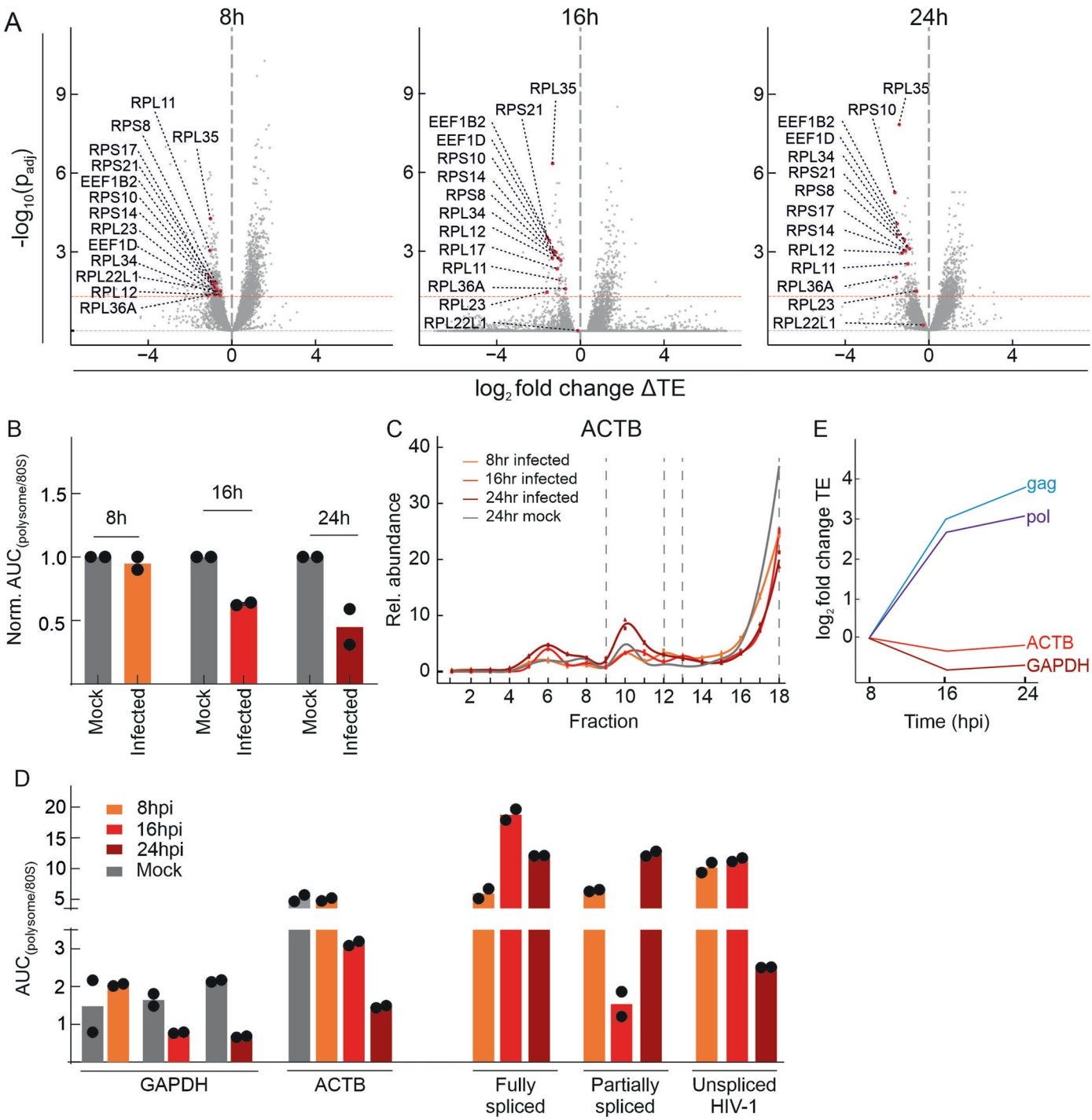

**Extended Data Fig. 2 | Differential translation efficiency of host and viral genes during HIV-1 infection.** (**a**) Volcano plots of differential translation efficiency (TE) of HIV-1 infected cells compared to the uninfected sample at each time point of infection. The downregulated translation-related genes are marked at each time-point. (**b**) Normalized ratio of area under the curve (AUC) calculated for 80S and polysome peaks of the polysome profiles as seen in Fig. 2b, for each timepoint of infection. Dots represent individual values from replicates. $n = 2$ independent experiments. (**c**) Relative abundance of *ACTB* along the polysome profile at each time point of infection. The depicted curve represents a smoothed interpolation derived from mean values of two qPCR replicates and both

replicate values are shown as points. The dashed lines mark the 80S (9–12) and polysome (13–18) containing fractions, respectively. (**d**) Normalized ratio of area under the curve (AUC) calculated for 80S and polysome peaks of the relative abundance mRNA profiles as seen in Fig. 2c, d and Supplementary Fig. 2c, for each timepoint of infection. For GAPDH, mock represents uninfected cells at each timepoint. For ACTB, mock represents 24 h uninfected cells. Dots represent individual values from replicates. $n = 2$ independent experiments. (**e**) Change in translation efficiency (calculated using RiboSeq data) of HIV genes – *Gag, Pol* and host genes – GAPDH, ACTB compared to 8 hours post infection.

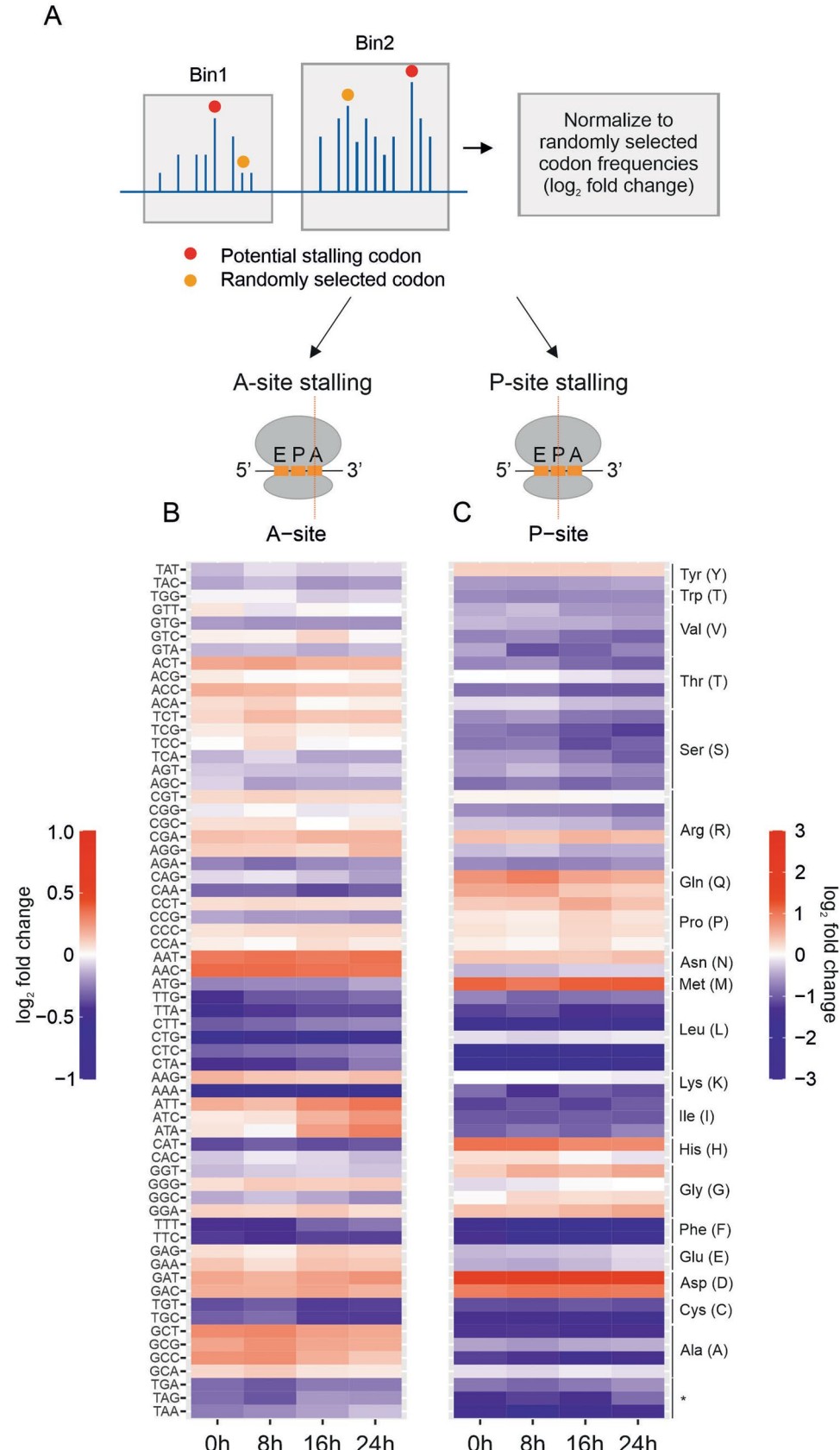

**Extended Data Fig. 3 | Genome-wide analysis of codon-specific ribosome stalling.** (**a**) Schematic representation of the human genome wide, codon specific stalling patterns. Bins are separated based on genomic distance and the first 5 codons of annotated coding sequences are excluded from analysis. Codon specific stalling in the (**b**) A-site and (**c**) P-site on a genome wide scale. Fold change (log₂) between randomized control (sampled 1000 times) and observed stalling frequencies of the codons are plotted at each timepoint.

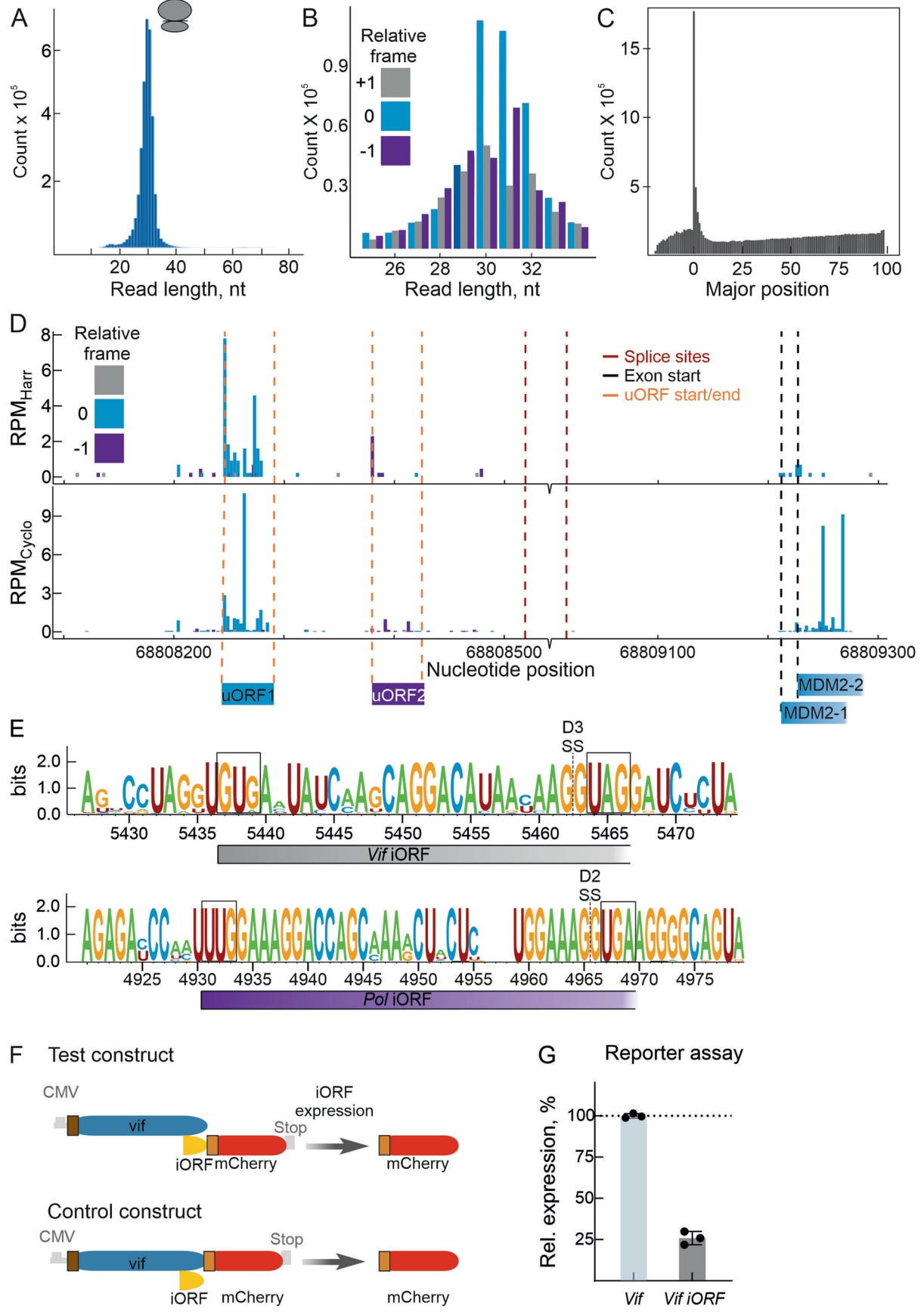

**Extended Data Fig. 4 | See next page for caption.**

**Extended Data Fig. 4 | Alternative initiation in host and HIV-1 ORFs.** Read length (**a**) and relative frame distribution (**b**) of all reads overlapping a CDS, obtained from harringtonine-treated ribosome profiling libraries prepared from SupT1 cell lysates. (**c**) The number of reads starting within coding transcripts for the Harringtonine RiboSeq dataset of one replicate. The x axis is the relative position of the read start within the three regions (5′-UTR: <0; CDS: 0-100; 3′-UTR: >100). The major isoform is used as reference for each gene (the longest coding transcript). (**d**) Close-up view of the 5′UTR of host MDM2 gene, showing the distribution of ribosome P-sites and the position of uORFs in the harringtonine- and cycloheximide-treated RiboSeq. (**e**) Graphical representation of the sequence conservation of the nucleotides corresponding to *Vif* and *Pol* iORF. The sequence logo was created from an alignment of 4903 HIV sequences obtained from the Los Alamos National Laboratory (LANL) HIV database (HIV Sequence Compendium 2021 Apetrei C. et al,) Putative start- and stop-codons are marked by grey boxes. (**f**) Schematics of the *in vitro* fluorescence reporter assay for measuring the expression of iORF activity. (**g**) Quantification of *Vif* iORF expression relative to the canonical 0-frame *Vif* ORF using the *in vitro* fluorescence reporter assay. $n$ = 3 independent experiments. Data from each replicate is shown and error bars indicate standard deviation.

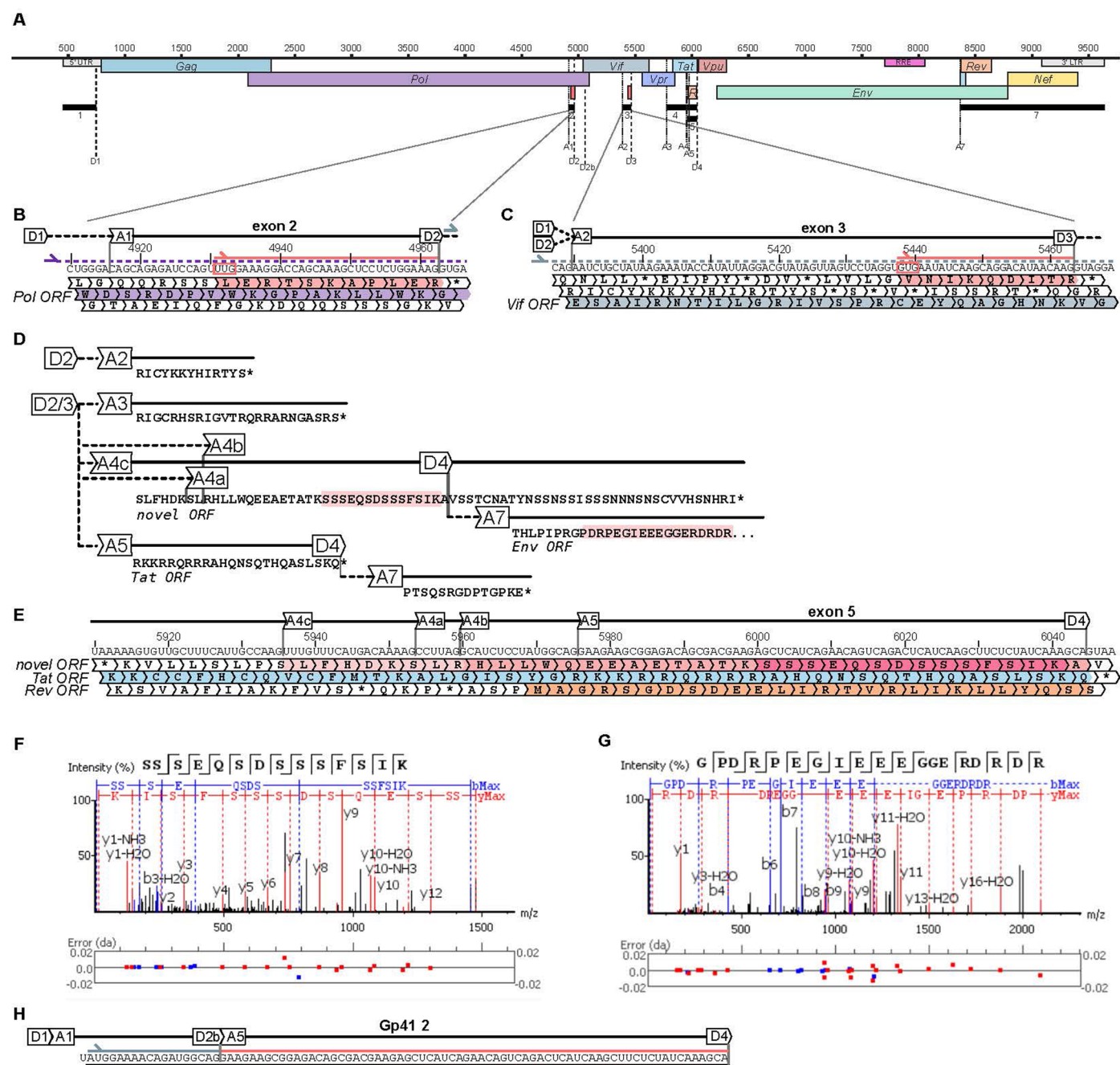

**Extended Data Fig. 5 | Alternative splicing and identification of novel HIV-1 peptides.** (**a**) Schematic of HIV genome structure including major splice donor and acceptor sites, ORFs and exons. (**b**) Schematic of *Pol* iORF located within exon 2. (**c**) Schematic of *Vif* iORF located within exon 3. (**d**) Schematic showing translation products of *Pol* or *Vif* iORF spliced at D2/D3 respectively to different acceptor sites. (**e**) Schematic showing translation frames of exon 5 with novel

ORF colored in light red and unique peptide in pink. Annotated fragmentation spectrum and delineated sequence fragmentation of (**f**) novel peptide encoded in exon 5 and (**g**) detected peptide of *Env* ORF after further translation across the D4A7 splice site, indicative of the confidence of the peptide identification. (**h**) Potential alternative source of the detected peptides from a cryptic D2bA5 splice event leading to transcript Gp41 2.

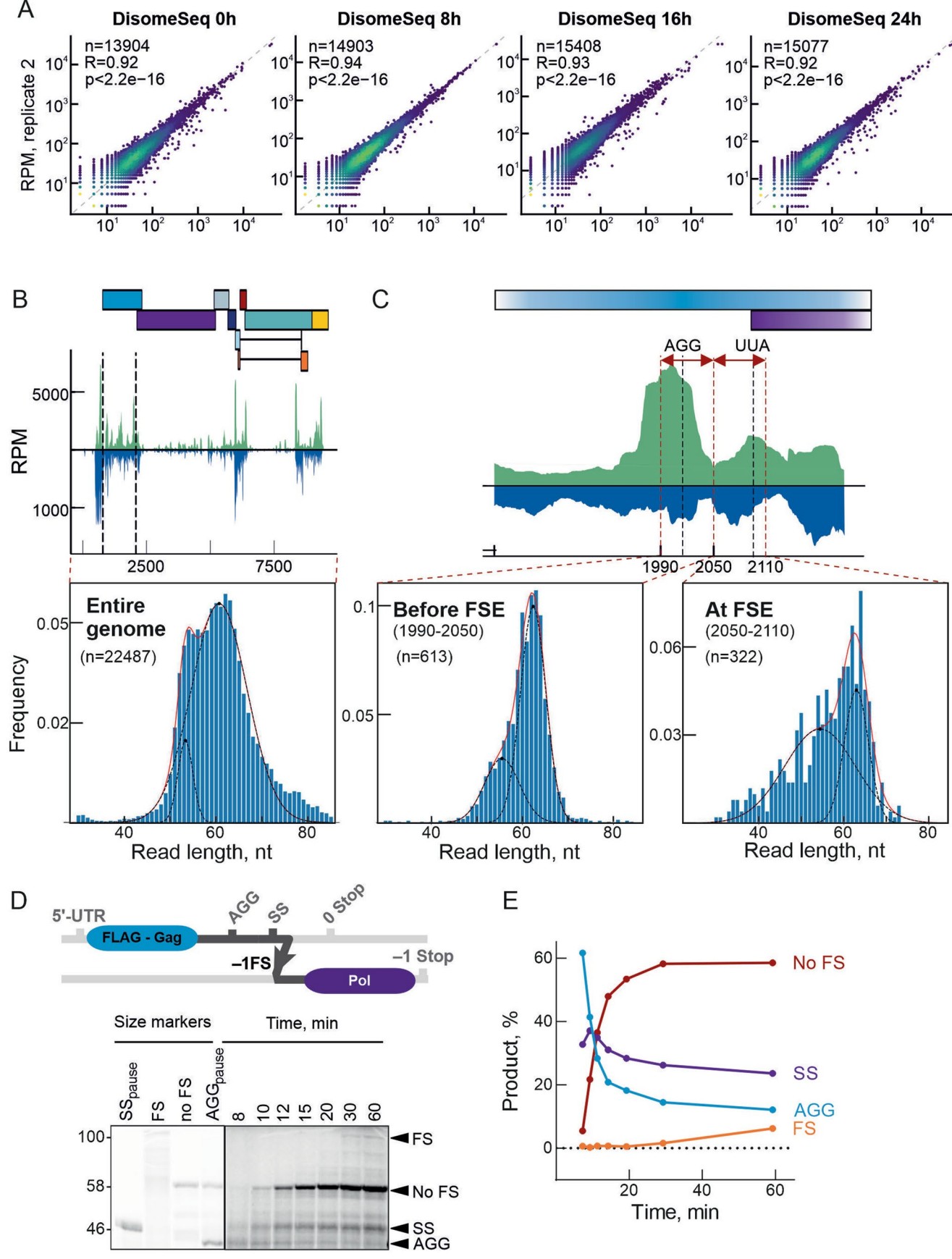

**Extended Data Fig. 6 | See next page for caption.**

**Extended Data Fig. 6 | DisomeSeq analysis and confirmation of ribosomal pausing in HIV-1.** (**a**) Scatterplots representing the correlation and reproducibility among duplicates of DisomeSeq analysis at each timepoint of infection. Reads per million (RPM), as assigned to individual genomic features by featureCounts[1]. Number of reads (n), significance (*p* value) and Pearson correlation coefficient (R) values are indicated for each scatterplot (higher density points shown in blue-green). Read length distribution of reads obtained from disome profiling libraries prepared from SupT1 cell lysates for (**b**) entire HIV-1 genome and (**c**) reads ending before FSE (1990–2050) and reads starting at FSE (2050–2110). (**d**) Schematics of the N-terminal FLAG-tagged frameshifting reporter consisting of nucleotides 64–2687 (Δ1870–1881) of the HIV-1 genome (up). Time course of translation of reporter mRNA and controls in RRL (bottom). Briefly, *in vitro* transcribed reporter RNA was translated in RRL, after 3 min further initiation was halted by the addition of harringtonine, and aliquots were removed at various times and analyzed by SDS–PAGE. RNA size markers were similarly translated and analyzed to show the expected size of the ribosomal pause product (See also Methods). (**e**) Quantification of the products of the western blot analysis.

# Reporting Summary

## Statistics

For all statistical analyses, confirm that the following items are present in the figure legend, table legend, main text, or Methods section.

| n/a | Confirmed | |
|---|---|---|
| ☐ | ☒ | The exact sample size (*n*) for each experimental group/condition, given as a discrete number and unit of measurement |
| ☐ | ☒ | A statement on whether measurements were taken from distinct samples or whether the same sample was measured repeatedly |
| ☐ | ☒ | The statistical test(s) used AND whether they are one- or two-sided<br>*Only common tests should be described solely by name; describe more complex techniques in the Methods section.* |
| ☐ | ☒ | A description of all covariates tested |
| ☐ | ☒ | A description of any assumptions or corrections, such as tests of normality and adjustment for multiple comparisons |
| ☐ | ☒ | A full description of the statistical parameters including central tendency (e.g. means) or other basic estimates (e.g. regression coefficient) AND variation (e.g. standard deviation) or associated estimates of uncertainty (e.g. confidence intervals) |
| ☐ | ☒ | For null hypothesis testing, the test statistic (e.g. *F*, *t*, *r*) with confidence intervals, effect sizes, degrees of freedom and *P* value noted<br>*Give P values as exact values whenever suitable.* |
| ☒ | ☐ | For Bayesian analysis, information on the choice of priors and Markov chain Monte Carlo settings |
| ☒ | ☐ | For hierarchical and complex designs, identification of the appropriate level for tests and full reporting of outcomes |
| ☐ | ☒ | Estimates of effect sizes (e.g. Cohen's *d*, Pearson's *r*), indicating how they were calculated |

*Our web collection on statistics for biologists contains articles on many of the points above.*

## Software and code

Policy information about availability of computer code

| Data collection | Polysome profiling - monitoring absorbance at 260 and 280 nm TriAX Flow Cell version 1.56A<br>qRT-PCR - CFX Maestro software version 2.3<br>Flow cytometry - NovoCyte Quanteon (ACEA) instrument (Novoexpress (version 1.6.2)).<br>In vitro translation- ImageJ 1.53c<br>Nanopore sequencing - MinKNOW version 23.04.3 |
|---|---|
| Data analysis | Graphpad prism 9 software (version 9.5.1) (https://www.graphpad.com/scientific-software/prism/);<br>Polysome profiles and qRT PCR - Data exported to Excel and visualized in Graphpad.<br>Flow cytometry: Mean fluorescence intensities by Novoexpress software (version 1.6.2) Mean data exported to Excel and FE calculated with Excel - Visualized in Graphpad<br>Riboseq data -PRICE pipeline (https://github.com/erhard-lab/price), DeltaTE pipeline (https://github.com/SGDDNB/translational_regulation).<br>DMS-Map data - RNA Framework v2.7.2, DREEM clustering algorithm (https://codeocean.com/capsule/6175523/tree/v1_), StructureEditor (version 1.0)<br>NanoLC-MS/MS - Peaks Studio  Xpro 11<br>Nanopore sequencing - Guppy version 6.5.7,   Isoquant version 3.3.0, Biopython 1.79 |

For manuscripts utilizing custom algorithms or software that are central to the research but not yet described in published literature, software must be made available to editors and reviewers. We strongly encourage code deposition in a community repository (e.g. GitHub). See the Nature Portfolio guidelines for submitting code & software for further information.

## Data

Policy information about availability of data

All manuscripts must include a data availability statement. This statement should provide the following information, where applicable:
- Accession codes, unique identifiers, or web links for publicly available datasets
- A description of any restrictions on data availability
- For clinical datasets or third party data, please ensure that the statement adheres to our policy

> High-throughput sequencing data have been submitted to Gene Expression Omnibus (GEO) and are available under the accession number GSE244468. The mass spectrometry proteomics data have been deposited to the ProteomeXchange Consortium via the PRIDE partner repository with the dataset identifier PXD058232, which will be made available upon online publication. Further information and requests for resources and reagents should be directed to and will be fulfilled by the lead contact neva.caliskan@helmholtz-hiri.de.

## Research involving human participants, their data, or biological material

Policy information about studies with human participants or human data. See also policy information about sex, gender (identity/presentation), and sexual orientation and race, ethnicity and racism.

| | |
|---|---|
| Reporting on sex and gender | Not applicable |
| Reporting on race, ethnicity, or other socially relevant groupings | Not applicable |
| Population characteristics | Not applicable |
| Recruitment | Not applicable |
| Ethics oversight | Not applicable |

Note that full information on the approval of the study protocol must also be provided in the manuscript.

# Field-specific reporting

Please select the one below that is the best fit for your research. If you are not sure, read the appropriate sections before making your selection.

☒ Life sciences ☐ Behavioural & social sciences ☐ Ecological, evolutionary & environmental sciences

For a reference copy of the document with all sections, see nature.com/documents/nr-reporting-summary-flat.pdf

# Life sciences study design

All studies must disclose on these points even when the disclosure is negative.

| | |
|---|---|
| Sample size | No sample-size calculation was done. n=3 independent replicates for all in-vitro or cell line experiments, which is the standard sample size for statistically significant biological tests. For all sequencing experiments, n = 2 independent replicates were made for all high-throughput sequencing experiments, including ribosome and disome profiling and RNA-Seq analyses. This is in line with the standard protocols in the field. This sample size is sufficient for the statistical analysis conducted by PRICE and DeltaTE pipelines. Detailed information about teh sample numbers for each experiment is provided in the manuscript. |
| Data exclusions | No data was excluded from the analysis. |
| Replication | The reproducibility of our experiments was confirmed, with 2 replicates tested for deep sequencing experiments. For the in vitro ribosome pausing and dual fluorescence assays were conducted in triplicates. All 3 replicate values are shown on the graph for dual fluorescence reporter assays, and representative of the 3 replicates is shown for the ribosome pausing. |
| Randomization | Randomization is not relevant to this study because the samples were not allocated into seperate experimental groups. |
| Blinding | Blinding is not applicable to this study because all data were acquired by machines or by custom/publicly available scripts. |

# Reporting for specific materials, systems and methods

We require information from authors about some types of materials, experimental systems and methods used in many studies. Here, indicate whether each material, system or method listed is relevant to your study. If you are not sure if a list item applies to your research, read the appropriate section before selecting a response.

## Materials & experimental systems

| n/a | Involved in the study |
|---|---|
| ☒ | ☐ Antibodies |
| ☐ | ☒ Eukaryotic cell lines |
| ☒ | ☐ Palaeontology and archaeology |
| ☒ | ☐ Animals and other organisms |
| ☒ | ☐ Clinical data |
| ☒ | ☐ Dual use research of concern |
| ☒ | ☐ Plants |

## Methods

| n/a | Involved in the study |
|---|---|
| ☒ | ☐ ChIP-seq |
| ☐ | ☒ Flow cytometry |
| ☒ | ☐ MRI-based neuroimaging |

## Eukaryotic cell lines

Policy information about cell lines and Sex and Gender in Research

| | |
|---|---|
| Cell line source(s) | HEK293T cells (gift from Prof. Jörg Vogel, HIRI-HZI; originally from ATCC), SupT1 cells ((NIH HIV Reagent Program, Division of AIDS, NIAID, NIH: Sup-T1 Cells, ARP-100, contributed by Dr. Dharam Ablashi) |
| Authentication | Cell lines were not aunthenticated |
| Mycoplasma contamination | All cell lines negative for mycoplasma contamination (checked via PCR with primers specific for mycoplasma as well as confirmed in RNASeq) |
| Commonly misidentified lines (See ICLAC register) | No commonly misidentified cell lines used |

## Plants

| | |
|---|---|
| Seed stocks | n/a |
| Novel plant genotypes | n/a |
| Authentication | n/a |

## Flow Cytometry

### Plots

Confirm that:

☒ The axis labels state the marker and fluorochrome used (e.g. CD4-FITC).

☒ The axis scales are clearly visible. Include numbers along axes only for bottom left plot of group (a 'group' is an analysis of identical markers).

☒ All plots are contour plots with outliers or pseudocolor plots.

☒ A numerical value for number of cells or percentage (with statistics) is provided.

### Methodology

| | |
|---|---|
| Sample preparation | HEK293 cells were transiently transfected using polyethylenimine (PEI) according to manufacturer's instructions using a 1:12 DNA:PEI ratio with either the control construct or the 1FS construct encoding for the dual-fluorescence EGFP-mCherry translation reporter as outlined in Figure 6. Cells were harvested at 24 h post-transfection. After washing with phosphate-buffered saline (PBS). |
| Instrument | NovoCyte Quanteon (ACEA) |
| Software | NovoExpress Software (version 1.6.2) |
| Cell population abundance | Cells were gated for FSC/SSC to ignore cell debris, which were approximately 70-80% of the population, indicating healthy cells. Then the mean GFP and mcherry intensity of these cells were measured. |

Gating strategy   Cells were gated for FSC/SSC to capture the bulk of the healthy cell population and and further analyzed for the mean intensities of EGFP (FITC channel) and mCherry (Texas Red channel). Figure exemplifying the gating strategy is now provided as Supplementary Figure 1.

☒ Tick this box to confirm that a figure exemplifying the gating strategy is provided in the Supplementary Information.

