## [Peer Review File · Nature Structural & Molecular Biology]

The translational landscape of HIV-1 infected cells reveals key gene regulatory principles

Corresponding Author: Professor Neva Caliskan

Version 0:

Decision Letter:

8th Dec 2023

Dear Professor Caliskan,

Thank you again for submitting your manuscript "The translational landscape of HIV-1 infected cells reveals novel gene regulatory principles". I sincerely apologize for the unusual delay in responding, which resulted from the difficulty in obtaining suitable referee reports. Nevertheless, we now have comments (below) from the 3 reviewers who evaluated your paper. In light of those reports, we remain interested in your study and would like to see your response to the comments of the referees, in the form of a revised manuscript.

You will see that while all 3 reviewers find the work interesting and timely, Reviewer #1 requests clarifications about data reproducibility and more detailed presentation of the results, as well as strengthening the links between the data presented and the conclusions drawn, and discussion of the BiorXiv papers. Reviewers #2 and #3 both request better substantiation of the extended frameshift structure, and Reviewer #3 further asks for strengthening and better description of the optical tweezer experiments. Please be sure to address/respond to all concerns of the referees in full in a point-by-point response and highlight all changes in the revised manuscript text file.

We appreciate the requested revisions additional experimentation. We thus expect to see your revised manuscript within 3-4 months. If you cannot send it within this time, please let us know. We will be happy to consider your revision as long as nothing similar has been accepted for publication at NSMB or published elsewhere. Should your manuscript be substantially delayed without notifying us in advance and your article is eventually published, the received date would be that of the revised, not the original, version.

Reporting Summary:
<https://www.nature.com/documents/nr-reporting-summary.pdf>

- that unprocessed scans are clearly labelled and match the gels and western blots presented in figures.
- that control panels for gels and western blots are appropriately described as loading or sample processing controls
- all images in the paper are checked for duplication of panels and for splicing of gel lanes.

Please note that all key data shown in the main figures as cropped gels or blots should be presented in uncropped form, with

molecular weight markers. These data can be aggregated into a single supplementary figure. While these data can be displayed in a relatively informal style, they must refer back to the relevant figures. These data should be submitted with the last revision, prior to acceptance, but you may want to start putting it together at this point.

SOURCE DATA: we request that authors provide, in tabular form, the data underlying the graphical representations used in figures. This is to further increase transparency in data reporting, as detailed in this editorial (<http://www.nature.com/nsmb/journal/v22/n10/full/nsmb.3110.html>). Spreadsheets can be submitted in excel format. Only one (1) file per figure is permitted; thus, for multi-paneled figures, the source data for each panel should be clearly labeled in the Excel file; alternately the data can be provided as multiple, clearly labeled sheets in an Excel file. When submitting files, the title field should indicate which figure the source data pertains to. We encourage our authors to provide source data at the revision stage, so that they are part of the peer-review process.

We require deposition of coordinates (and, in the case of crystal structures, structure factors) into the Protein Data Bank with the designation of immediate release upon publication (HPUB). Electron microscopy-derived density maps and coordinate data must be deposited in EMDB and released upon publication. Deposition and immediate release of NMR chemical shift assignments are highly encouraged. Deposition of deep sequencing and microarray data is mandatory, and the datasets must be released prior to or upon publication. To avoid delays in publication, dataset accession numbers must be supplied with the final accepted manuscript and appropriate release dates must be indicated at the galley proof stage. Please find the complete NRG policies on data availability at <http://www.nature.com/authors/policies/availability.html>.

Link Redacted

Sincerely,
Sara

Sara Osman, Ph.D.
Associate Editor
Nature Structural & Molecular Biology

Referee expertise:

Referee #1: RNA and viral translation biology

Referee #2: Ribosome profiling

Referee #3: Single molecule RNA studies

Reviewers' Comments:

Reviewer #1:

Remarks to the Author:

This manuscript entitled "the translational landscape of HIV-1 infected cells reveals novel regulatory principles" by Kibe et al. reports a ribosome profiling study along HIV-1 infection, complemented with transcriptomic and proteomic results. Although its title may be a bit emphatic, this manuscript presents several interesting and novel findings, notably the discovery of two new iORF. Many state of the art experiments have been performed, and this undeniably represents the most up to date and thorough report on HIV translation. However, some parts of the manuscript are very confusing, and the criteria that lead the authors to emphasize some of their results rather than others need to be clarified. A reproducibility issue has to be addressed too. Finally, some information and results are missing, they should be added. Points to be addressed are listed below

L 68-69: It would be interesting to summarize here what is the overall conclusion of all these reports (and others?). In addition, this is actually not completely true since the manuscript of Labaronne et al. has been available on bioRxiv for some time. Their conclusion may have to be summarized here too.

There are no legends and not even any label or title to the supplementary tables. Please provide this information.

I may have missed them but I could not find the RNAseq results.

Ribosome profiling experiments have been duplicated, however nothing is said about the reproducibility of these experiments. Could the author show/comment about the correlation of the data. Then it is not clear what are the results that are presented: an average of both replicates or replicates were so similar that only the result for one of them is shown?

L104-130 How did the “interesting” ontology fields (those discussed in the manuscript) have been chosen? Number of genes regulated in the pathway? Strong regulation? Else?

L104-130 how are the enrichment calculated? From one time point to another, or always in reference to the 0 time point (mock control)?

L120-121 it is not clear to me what this sentence means.

L128-130 As translation is the focus of this study, it would be interesting to know which are the 14 genes involved? Why only 10 of them have been selected for TE calculation?

L131-132 Are all the regulated pathways present in this table? How were the selected the GO for which the TE has been calculated?

L140-142 As far as I understand this is not what table 2 says. I see RPL22, RPS14 at 8h, RPL26, RPL13A, RPS9 at 16h and RPL13A, RPS9 at 24h

L143-147 It is hard to evaluate these assertions without the ma-seq data. In deed if relying only on calculated TE, about as many genes are upregulated than downregulated (72 up vs 23 down at 8h hours, 126 down vs 95 down at 16h)

L149-185: I very well understand that polysome profile experiments are tricky to quantify and may not be 100% reproducible. However, in a study focusing on translational regulation this is a bit of a concern. Consequently, several conclusions appear at least disputable:

i) It is concluded that overall the translation is inhibited at 16h and 24h, but looking at the ribosome profile I would say that there is more translation at 16h and even at 24h than at 8 hours (that cannot be explained by the 0,5% to 2,5% of HIV RNA engaged in translation at these time points as determined by Ribo-Seq).

ii) It is concluded that GADPH (and ACTB) are actively translated at 8h, but are subsequently removed from the active polysome fraction. However, taking into account the experimental variation, the differences between profiles at different time point, or with the mock control are rather subtle.

iii) I may have missed something but if HIV RNA represent 0.5% and 2.5% of the ribosome-associated RNA, can a “preferential translation of viral over host genes” be claimed?

All conclusions drawn sound very reasonable, but they are hardly supported by the data. I found this chapter and the conclusions drawn very confusing, the authors need to clarify this chapter.

Figure 2F : indicate that the gray line is the mock control.

L224-225 The figure may be deceptive, but I see mostly signal on Rev, and not on Tat. Actually there are no more reads on Tat than on vpu or nef (figure 3B). Please clarify

L230-232: In deed the sharp increase on Gag is observed between 16 and 24 hours. But doesn't this come in contradiction with the sharp decrease of the % of unspliced HIV-1 RNA in the polysome fraction and its relocation to the 40S fraction between 16h and 24h as seen in figure 2G or supp 2E? Please clarify

L234-293: This chapter about the identification of potential alternative translation initiation events is very interesting and uncovers (at least) to novel iORF in HIV-1. However, several things need to be clarify for an easier reading and a better understanding of how these results were obtained and their relevance.

When looking at figure 4A where are reported the RPMharr (Please define RPM), it seems that the overwhelming majority of the reads, are in the 5'UTR (on non AUG start codon), then on rev 5' exon, then in rev second exon (or env may be), then in vpu, then in the Gag coding region and finally in pol and vif. Results about the 5'UTR are not in the supplementary table; could you please provide them? Only the results about Pol and vif are discussed. The examination of the supplementary table (Please legend the supplementary table) reveals that this is because the p-value calculated by PRICE is not <0.005 for all of the other results. Please explain the calculation of this p value and why it leads to the discard of the vast majority of the results obtained.

Nothing is said about the results obtained on cellular mRNA. I understand that this is not the scope of this publication, but few indicators of the relevancy of the results about the cellular transcriptome could give weight to the novel findings about HIV

L256: “Downstream” instead of “upstream”?

L266-272: My understanding from L260-263 is that the authors suggest that this iORF may be expressed from a D1-A2 splicing? In this context, the experiment performed in supp4E is confusing. If this is to suggest that an alternative translation event such as a frameshift or an internal ribosome entry may take place, it would require some additional controls.

294-319: Why are only the results obtained for the FSE region discussed? Are the others not statistically relevant? Could the authors comment on the other strong disome peaks (within Gag, env or nef)?

Figure 6: The color coding of the probing results should be indicated. 6C: HIV-1 instead of “SARS-CoV-2” should be replaced by HIV-1? 6D: What is the NT mutant? Indicate the delta ext mutant boundaries on 6B?

The DMS probing data actually poorly supports at least three of the pairings proposed

Supp figure 7: It is not clear what is the difference between “contour length changes” and “total contour length changes”

L488-489: As far as I understand it “ASO targeting of the lower stem” slightly decrease, rather than enhance the frameshifting efficiency?

615-616 I have not found any of those, could you please indicate where they can be found

Reviewer #2:

Remarks to the Author:

This manuscript reports on a timecourse of RiboSeq translational profiling of HIV-1 infected T cells. Infection causes early translational changes in host genes, accompanied by later changes in mRNA abundance. Viral RNAs remain highly translated as host translation declines, and uncharacterized viral ORFs are described. These include uORFs as well as internal ORFs resulting from non-canonical splicing events. Ribosome occupancy around the HIV-1 frameshift shows unexpected patterns, including ribosome accumulation and stacking upstream of the frameshift site itself. Ribosome stacking leads to the proposal of an extended structural element around the frameshift site, supported by DMS probing in cells and by single-molecule folding trajectories.

This manuscript provides a valuable resource for the study of HIV, and notably proposes a significant addition to our understanding of the HIV-1 frameshift. The genomic data are high quality and supported by validation data, for example by independent proteomic validation of iORF-encoded peptides as well as reporter constructs. While data presented are strong, additional tests would strengthen the argument for the extended frameshift structure—one of the most substantial and exciting findings in this work. Certain other points seem to over-interpret the data presented. I would support publication of a suitably revised version of the manuscript addressing the concerns below.

1. Host cell translational profiling identifies changes in ribosomal protein translation. Such changes are often indicative of changes in 5' TOP mRNA translation downstream of changes in mTOR activity. It would be valuable to look at this specifically for a well-established collection of 5' TOP transcripts; either gathering evidence for such a change, or refuting it, would be interesting.
2. The sucrose density gradients in Figure 2 are over-interpreted, with too little consideration of alternate explanations. For instance, in Fig 2F, the shift of GAPDH into the heaviest fractions is interpreted to “represent stress granules” (l. 169) whereas a rather similar shift for fully-spliced HIV-1 transcripts are interpreted to indicate that they are “actively translated”. If similar data are consistent with either repression (Fig 2F) or active translation (Fig 2G), then the data don't support one or the other alternative. Translation shutoff experiments could resolve this question, but as this does not seem to be a central point for the paper, simply providing a more consistent interpretation would address this concern.
3. A few related concerns arise about the analysis of pausing and codon usage in Supp Figure 3. First, cell lysates were prepared after chilling cells to 4 °C for 3 minutes and exposing them to cycloheximide, which seems likely to distort measured ribosome positions. Second, the analysis focuses on extreme (> 2 standard deviation) pauses, but the discussion focuses on changes in tRNA pools, which are expected to produce broad shifts in per-codon occupancy rather than extreme pauses. My sense is that the scope of conclusions here are limited in any case.
4. Viral gene expression measurements in Figure 3 rely on deconvolution of the different splice isoforms. Is it possible to use RSEM or similar tools to carry out more robust deconvolution?
5. 54 nt seems too short for a footprint from a proper disome. Even in yeast, where footprints are slightly shorter, 54 nt footprints are attributed to quality control intermediates whereas proper disomes are more than 60 nt [10.1016/j.cell.2014.02.006].
6. It would be valuable to present the length profile of disome footprints at and around the programmed frameshift relative to the overall length distribution.
7. In discussing Supp Figure 6B-C, it is written that “a persistent translational pause” suggests “potential ribosome drop-off”. I think it is meant that a persistent protein product at this position suggests drop-off? This would seem to better explain the profiling and in vitro translation data.
8. In discussing frameshift reporters (ll. 394-395), a control is described that has both the canonical slippery site (SS) and the new candidate slippery site (SS*) mutated, but no data from this control is presented. It seems important to present these data, which would indicate the level of “background mCherry signal” present in the absence of any slippery sites.
9. No non-targeting ASO control is presented for the data arguing that an ASO can disrupt frameshifting by disrupting the extended structural element.
10. Likewise, some minimal mutational analysis of the proposed novel stem-loop would be valuable; compensatory mutations could provide strong evidence for a functionally important structure, and would also produce an ASO-resistant variant.
11. As a minor point, the text (ll. 120-121) reads “...prominent changes in RNAseq read without changes.” Presumably this is an editing error and it is intended to claim, roughly that there are no RiboSeq changes beyond what can be explained by RNAseq?

Reviewer #3:

Remarks to the Author:

The study entitled "The translational landscape of HIV-1 infected cells reveals novel gene regulatory principles" involves several sequencing and biochemical/biophysical techniques to investigate HIV-1 frameshifting both in vivo and in vitro. There are several interesting aspects of this study. Among them are the modulation of the transcriptome over time by HIV-1 infection and the potential fine tuning of frame shifting efficiencies by an upstream structural element. The combination of experimental approaches is very interesting and provides valuable insight on the role of -1 frame shifting in the life cycle of HIV-1. Overall, there is an appropriate analysis of the resultant sequencing data along with speculation founded in prior work that provides strength to their conclusions. I was somewhat confused during the presentations of results as it related to the extended structural frameshift site. They mention previous work as it related to their DMS probing, but I feel the discussion of the differences was insufficient. In terms of the optical tweezers experiments, I do not find these were well described. In the figure legend they mention SARS-CoV-2, I missed that part in the paper where they were looking at that. They need to be more specific about what region of the HIV-1 sequence they put between the handles. This wasn't clear. The assignment of the higher force unfolding event could easily be performed by unfolding the hairpin alone. This wasn't done. The figure is very small and it is challenging to observe the I to U transition. The authors discuss the hysteresis in the first unfolding transition, but it is hard to evaluate that as compared to the second unfolding transition. The numbers presented show similar error between the two transition steps. Why is one exhibiting more hysteresis than the other? Overall, I don't find the results and discussion on this second part related to the extended frame shifting site as well reasoned or compelling as the first section. I am not convinced the optical tweezers experiments are in complete agreement with the proposed extended sequence structure.

Version 1:

Decision Letter:

30th Jul 2024

Dear Dr. Caliskan,

Thank you again for submitting your manuscript "The translational landscape of HIV-1 infected cells reveals novel gene regulatory principles". Please accept my sincere apologies for the extremely long delay in sending you a decision on your study. As explained in previous correspondence, the original Reviewer #3 was not able to return a report on the revised manuscript. Because the referee had unique expertise in single molecule studies, and mechanobiology, we could not move forward with this expertise covered. Unfortunately, it was extremely difficult to recruit an alternative referee who was available to comment on the revised study. It is most unfortunate when such a situation occurs, and we are very sorry for this very long delay.

You will see that the original Reviewers #1 and #2 don't have any additional comments. The new arbitrating Reviewer #4 appreciates the study and that the optical tweezers experiments are not a central part of the manuscript. However, after assessing the original submission, and the revised manuscript, the referee notes that some important points remain that must be addressed - and that addressing these are essential to support the proposed model. It is unfortunate to have additional reviewer requests at this point in the peer-review process. However, we found the reviewers concerns fully justified in their thoughtful report, and agree that addressing these will be important for the readership of NSMB.

Please be sure to address and respond to all concerns of the referee in full in a point-by-point response and highlight all changes in the revised manuscript text file. If you have comments that are intended for editors only, please include those in a separate cover letter. We are committed to providing a fair and constructive peer-review process. Do not hesitate to contact us if there are specific requests from the reviewers that you believe are technically impossible or unlikely to yield a meaningful outcome.

We expect to see your revised manuscript within 6 weeks. If you cannot send it within this time, please contact us to discuss an extension; we would still consider your revision, provided that no similar work has been accepted for publication at NSMB or published elsewhere.

Reporting Summary:

When submitting the revised version of your manuscript, please pay close attention to our <https://www.nature.com/nature-portfolio/editorial-policies/image-integrity> Digital Image Integrity Guidelines and to the following points below:

Please note that all key data shown in the main figures as cropped gels or blots should be presented in uncropped form, with molecular weight markers. These data can be aggregated into a single supplementary figure item. While these data can be displayed in a relatively informal style, they must refer back to the relevant figures. These data should be submitted with the final revision, as source data, prior to acceptance, but you may want to start putting it together at this point.

Data availability: this journal strongly supports public availability of data. All data used in accepted papers should be available via a public data repository, or alternatively, as Supplementary Information. If data can only be shared on request, please explain why in your Data Availability Statement, and also in the correspondence with your editor. Please note that for some data types, deposition in a public repository is mandatory - more information on our data deposition policies and available repositories can be found below:

<https://www.nature.com/nature-research/editorial-policies/reporting-standards#availability-of-data>

We require deposition of coordinates (and, in the case of crystal structures, structure factors) into the Protein Data Bank with the designation of immediate release upon publication (HPUB). Electron microscopy-derived density maps and coordinate data must be deposited in EMDb and released upon publication. Deposition and immediate release of NMR chemical shift assignments are highly encouraged. Deposition of deep sequencing and microarray data is mandatory, and the datasets must be released prior to or upon publication. To avoid delays in publication, dataset accession numbers must be supplied with the final accepted manuscript and appropriate release dates must be indicated at the galley proof stage.

Link Redacted

Sincerely,

Carolina Perdigoto, PhD
Chief Editor
Nature Structural & Molecular Biology
orcid.org/0000-0002-5783-7106

Referee expertise:

Reviewers' Comments:

Reviewer #1:

Remarks to the Author:

The manuscript has been greatly improved and may be published

Reviewer #2:

Remarks to the Author:

Revisions have addressed my concerns with the original submission. This manuscript provides new insights into HIV translation and provides a valuable resource and I support its publication.

Reviewer #4:

Remarks to the Author:

In this study, Kibe et al. investigate HIV-1 frameshifting by combining multiple sequencing and biochemical techniques. Following reviewer #3 comments, I will focus my review on the single-molecule studies. In the first version of the manuscript, the authors employed single-molecule optical tweezers to understand the structural basis of the stalling event. Using DSS, they predicted that the HIV-1 RNA would fold as a three-way junction, including a GC-rich frameshift stem-loop (Fig. 6B). They used optical tweezers to pull on the frameshift site of HIV-1, which exhibited two distinct unfolding events: 1) a first low force ~ 10 pN event corresponding to a contour length change of ~ 64.5 nm; 2) a second higher force ~ 18 pN event involving a ~ 17 nm contour length change. Based, on these unfolding events, and the fact that the low force one showed a large unfolding/refolding hysteresis, the authors argued that the first event corresponded to the unfolding of the sequence encompassing the bulge (which has a more complex fold), while the high force event to the unfolding of the frameshift hairpin (which is rich in GC pairing, and therefore more stable).

Reviewer #3 raised several concerns regarding the clarity of the results and their interpretation, with which I agree. In particular, reviewer #3 suggested pulling on the frameshift stem-loop alone, to unambiguously assign that high-force event in the full-length HIV-1 RNA to the unfolding of the GC-rich hairpin. Of note, the nanomechanics of the frameshift stem loop were previously investigated using optical tweezers (Ritchie et al RNA, 2017). The authors have included two figures in their reply to reviewer #3, including new experiments using the frameshift hairpin. However, these new data do not fully agree with the authors' interpretation of the unfolding events observed for the full-length HIV-1 RNA—the high-force event in the full-length molecule is significantly longer than that of the isolated frameshift hairpin. Moreover, intriguingly, all data and discussion regarding the single-molecule experiments (former Fig. 6C, and Supp. Fig. 7) have now been removed from the manuscript and are only discussed in the rebuttal letter. While I understand that these data are not a central aspect of the study, the structural determination of the frameshift site is important for the conclusions of the manuscript, and the optical tweezers experiments can indeed provide relevant insights in this regard, which are currently inconsistent. Therefore, I believe the authors should provide a convincing explanation of the two unfolding events they observed in their initial optical tweezers experiments, which currently cannot be assigned to the unfolding of the frameshift hairpin, and the remainder of the molecule.

In the new experiments using the isolated frameshift stem-loop, the authors reproduce the results from Ritchie et al, observing a single unfolding event at ~ 17 pN corresponding to the unraveling of a contour length of ~ 12 nm, or 26 nucleotides, assuming ~ 0.45 nm per nucleotide—which is the standard value for RNA. This agrees with the unfolding of the GC-rich upper stem, as reported previously by Ritchie et al. However, the high force event in their experiments using the full-length HIV-1 RNA renders a contour length increase of ~ 17 nm, and therefore ~ 37 nucleotides, a significantly longer structure than that of the upper stem alone (the authors previously incorrectly assigned this contour length increase to the unfolding of 26 nucleotides, but that is not possible). The authors need to explain the origin of this discrepancy as the high force event in the full-length RNA needs to necessarily involve a larger structure, or perhaps even a more complex one. Interestingly, Ritchie et al reported that a small fraction of their curves ($\sim 0.1\%$) involved a contour length increase of ~ 17 nm (although occurring at a much higher force, ~ 30 pN), likely due to the formation of a tertiary structure involving the lower stem region of the hairpin. In the context of the full-length RNA molecule, this structure or a similar one could form more easily, perhaps due to the flexibility of the adjacent region of the bulge, and account for this extra contour length. The authors could demonstrate this by pulling on a larger region of the frameshift hairpin, aiming to obtain a single ~ 17 nm event that matches the high force event observed in the full-length molecule. Similarly, the authors could remove the frameshift stem-loop from their RNA molecule (as it seems to unfold separately from the rest of the molecule, it should not be a problem), and investigate whether the unfolding/folding of this region of the HIV-1 RNA matches the low force event observed in their initial experiments. This way, the authors could convincingly assign each of the unfolding events to well-defined elements within their proposed structure.

Version 2:

Decision Letter:

Our ref: NSMB-A48201B

11th Oct 2024

Dear Dr. Caliskan,

Thank you for submitting your revised manuscript "The translational landscape of HIV-1 infected cells reveals novel gene regulatory principles" (NSMB-A48201B). It has now been seen by the single molecule biophysics referee and their comments are below.

The reviewer finds that all their concerns have been addressed - the reviewer specifically notes that the tweezers experiments fully support your main findings and encourages you to include this data in the final study. We would agree with the referee that including those datasets would make for stronger paper overall, however, we will leave these decision to you.

In line of this feedback, we will be happy in principle to publish it in Nature Structural & Molecular Biology, pending minor revisions to comply with our editorial and formatting guidelines.

We are now performing detailed checks on your paper and will send you a checklist detailing our editorial and formatting requirements in about two weeks. Please do not upload the final materials and make any revisions until you receive this additional information from us.

Sincerely,

Carolina Perdigoto, PhD
Chief Editor
Nature Structural & Molecular Biology
orcid.org/0000-0002-5783-7106

Reviewer #4 (Remarks to the Author):

The authors have successfully addressed my concerns, and the new single-molecule data presented in the rebuttal letter explains convincingly their conclusions regarding the proposed structure of the extended frameshift structure. However, I still fail to understand (particularly in light of this last response) why they have decided not to include these data in the manuscript. I understand that the single-molecule data is not key to the paper's conclusions, but it would definitely strengthen their structural interpretation as presented in Fig. 6, especially because these data was included in the first submission (and I believe now is way more convincing). While a more in-depth single-molecule study could stand as a separate work, the initial experiments should be included in this manuscript.

Version 3:

Decision Letter:

3rd Dec 2024

Dear Dr. Caliskan,

We are now happy to accept your revised paper "The translational landscape of HIV-1 infected cells reveals key gene regulatory principles" for publication as a Article in Nature Structural & Molecular Biology.

Your paper will be published online soon after we receive proof corrections and will appear in print in the next available issue. You can find out your date of online publication by contacting the production team shortly after sending your proof corrections.

Please note that *Nature Structural & Molecular Biology* is a Transformative Journal (TJ). Authors may publish their research with us through the traditional subscription access route or make their paper immediately open access through payment of an article-processing charge (APC). Authors will not be required to make a final decision about access to their article until it has been accepted. [Find out more about Transformative Journals](https://www.springernature.com/gp/open-research/transformative-journals)

Kind regards,
Florian

Dr Florian Ullrich
Senior Editor, Nature
Consulting Editor, Nature Structural & Molecular Biology
ORCID 0000-0002-1153-2040

Point by Point Responses

We thank the reviewers for their time and efforts and for their thoughtful and constructive comments that allowed us to strengthen our manuscript. Point by point replies to individual reviewer comments is below. The original reviews and specific points are reproduced (in blue) for convenience.

Reviewer #1:

This manuscript entitled “the translational landscape of HIV-1 infected cells reveals novel regulatory principles” by Kibe et al. reports a ribosome profiling study along HIV-1 infection, complemented with transcriptomic and proteomic results. Although its title may be a bit emphatic, this manuscript presents several interesting and novel findings, notably the discovery of two new iORF. Many state of the art experiments have been performed, and this undeniably represents the most up to date and thorough report on HIV translation. However, some parts of the manuscript are very confusing, and the criteria that lead the authors to emphasize some of their results rather than others need to be clarified. A reproducibility issue has to be addressed too. Finally, some information and results are missing, they should be added. Points to be addressed are listed below

L 68-69: It would be interesting to summarize here what is the overall conclusion of all these reports (and others?). In addition, this is actually not completely true since the manuscript of Labaronne et al. has been available on bioRxiv for some time. Their conclusion may have to be summarized here too.

We previously discussed the main findings of Labaronne in the discussion, but as suggested we now briefly mention the main conclusions of non-canonical initiation strategies employed by the virus (revised L 54-56 and 66-69 below) also in the introduction.

“A recent ribosome profiling study also reported extensive uORF-mediated non-AUG translation in HIV-1, which add to the myriad of non-canonical translation strategies employed by the virus to control gene expression”

Furthermore, we have summarized the findings of previous reports (revised lines XX-XX) and changed the phrasing to better reflect the previous work.

“Previously many studies have elucidated specific aspects of HIV-1 gene expression including alternative splicing, non-canonical initiation events and programmed ribosome frameshifting, yet the translational landscape of host and viral mRNAs during infection remains underexploited.”

There are no legends and not even any label or title to the supplementary tables. Please provide this information.

We thank the reviewer for bringing this to our attention. We have now included legends for the supplementary tables, as requested.

Supp. Table 1: Overview of gene ontology terms associated with differentially regulated genes during HIV-1 infection. For gene ontology (GO) a PANTHER overrepresentation test for biological processes was performed. The results were obtained with a hierarchical order between GO terms. This table contains the top 15 enriched gene ontology terms in the hierarchy level 1, sorted by false discovery rate (FDR) for the differentially expressed genes with a RiboSeq fold change above- (ribo_up) or below (ribo_down) zero, for each HIV-1 infection time point. The table includes the numbers of genes in our dataset found in the biological process (**input_list.number_in_list**), the fold enrichment in comparison to the expected number of genes, FDR score, p-value, Gene IDs, the original process as well as the umbrella term.

Supp. Table 2: List of ORFs predicted in the host and virus via PRICE analysis. This table contains the extensive list of open reading frames predicted in both host and viral genomes as characterized by the PRICE bioinformatics pipeline. The table includes columns that specify the unique Gene ID, the ORF ID, ORF location within the genome, the first codon within the ORF, ORF type, and the PRICE specific Start score and range-score. Additional quantitative data provided include the *p*-value, the number of mapped reads (fractional for multimapping reads) per experimental condition, and the total number of reads for each ORF.

Supp. Table 3: Riboseq and RNASeq reads mapped to the HIV-1 genome spanning donor splice sites D2 and D3 with different acceptor sites. RiboSeq and RNASeq reads were mapped splice aware with the STAR-aligner to the HIV-1 genome. For stringency, no mismatches in RiboSeq mapping reads were allowed in this analysis. Furthermore, only RiboSeq reads with a read length ≥ 20 were considered. The reads spanning the D2-AX and D3-AX splice sites were counted and are listed under the respective splice acceptor site. Because ribosome protected fragments are only around 30 nt long, we further filtered for reads having at least 10 nucleotides on either side of the splicing event (min10 columns). Splice

sites are characterized by a sudden drop (donor) or increase (acceptor) of read coverage. Therefore, the coverage 1 nt before and 1 nt after the donor sites was investigated. The difference between those coverages allows an estimation of how many splice site overlapping reads are expected.

Supp. Table 4: Proteome analysis of HIV-1 infected cells via mass-spectrometry. This table includes the list of different peptides identified via mass-spectrometry of 24h uninfected and HIV-1- infected cells, analyzed with PEAKS Studio Xpro (Bioinformatics Solutions Inc., Canada).

Supp. Table 5: **Nanopore sequencing of HIV-1 transcripts associated with mass spectroscopy detected peptides.** This table includes the splice sites, isoforms and the number of corresponding reads found at each time point of infection (n=2). Colors indicate whether the isoform is likely to express one or both of the detected peptides.

Supp. Table 5: Nanopore sequencing of HIV-1 transcripts associated with mass spectroscopy detected peptides. This table includes the splice sites, isoforms and the number of corresponding reads found at each time point of infection (n=2). Colors indicate whether the isoform is likely to express one or both of the detected peptides.

I may have missed them but I could not find the RNAseq results.

We used RNAseq data in conjunction with RiboSeq data to obtain gene translation efficiencies (TE). The transcriptional changes can be seen in the main manuscript for host genes in **Figure 1** (X-axis) and for the HIV-1 genome in **Figure 3A** (RPM, light blue). The respective methodology can be found in material and methods parts 'NGS sequencing analysis' and 'Differential gene expression analysis'.

The raw data for RNA-seq are now available under [GSE244468](https://www.ncbi.nlm.nih.gov/geo/query/acc.cgi?acc=GSE244468) with the reviewer token gxytgeukpzqfleb.

Ribosome profiling experiments have been duplicated, however nothing is said about the reproducibility of these experiments. Could the author show/comment about the correlation of the data. Then it is not clear what are the results that are presented: an average of both replicates or replicates were so similar that only the result for one of them is shown?

This is an important point that we have now addressed by calculating Pearson correlation coefficients for all experiments (RiboSeq, RNAseq and DisomeSeq) (**Rebuttal Figure 1**).

Rebuttal Figure 1: Scatterplots of reads per million (RPM) assigned to individual genomic features, showing the correlation and reproducibility between duplicates of RiboSeq (A), RNASeq (B) and DisomeSeq (C). Number of reads (n), significance (p value) and Pearson correlation coefficients are indicated for each scatterplot.

The results of these analyses show the experiments to be highly reproducible (R ranging from 0.91-0.96) which are included as **Supplemental Figures 1 and 6**.

We also now provide the complete PRICE analysis in the **Source Data** which includes codon-resolved details for the average as well as each RiboSeq replicate.

L104-130 How did the “interesting” ontology fields (those discussed in the manuscript) have been chosen? Number of genes regulated in the pathway? Strong regulation? Else?

We have now refined our gene ontology analysis for clarity (**Figure 1C**). Briefly, the PANTHER GO enrichment analysis returns hierarchical gene ontology terms from most specific (level 0) to broader parental terms. For each timepoint, we analyzed two gene lists – RiboSeq fold-change (\log_2) >0 and RiboSeq fold-change (\log_2) <0 of the significantly differentially expressed genes (*i.e* genes above or below 0 of the Y-axis in **Figure 1B**). Next, we identified the top 15 gene ontology terms (by fold enrichment, hierarchy level 1) for each analysis, grouping them into broader categories such as cholesterol metabolism, immune response, apoptosis, translation and others. We have now highlighted genes with >1 fold change (\log_2) in

translational efficiency (TE) and colored them based on the umbrella terms (Revised Figure 1C), genes with >1.2 fold change and $-\log_{10}(p_{adj}) > 3.15$ in TE are labeled. Additionally, we provide pie charts as graphical representations for these categories, showing the gene distribution percentage of the highlighted genes for each infection time-point. **Supp. Table 2** provides detailed selected GO term classifications and their groupings. The **Source Data** includes all the PANTHER GO enrichment analysis prior to selection.

Rebuttal Figure 2 (Revised figure 1): Global transcriptional and translational changes in HIV-1 infected cells. (A) Schematic representation of the procedure to monitor transcript abundance and translation in HIV-1 infected cells. Briefly, SupT1 cells were infected or not with HIV-1-iGFP (NL4.3 strain). At 0, 8, 16 and 24 hours post infection (hpi), cells were lysed to recover the cytoplasmic fraction and prepared ribosome profiling and RNA-seq libraries were subjected to high-throughput sequencing. (B) Scatterplots of the fold-change (\log_2) in cytoplasmic RNaseq and RiboSeq levels of HIV-1 infected cells compared to the mock-sample (8h uninfected cells) at each time point of infection. Only genes that were called to be significantly differentially expressed (8 hpi: $n=1,254$, 16 hpi: $n=1,312$, 24 hpi: $n=2,257$) are shown. Genes are colored based on fold changes of Ribo, RNA, and TE (yellow: change in TE that

counteracts the change in RNA, red: change in Ribo, with no change in RNA leading to change in TE, blue: change in RNA and Ribo at the same rate, with no change in TE, purple: change in TE that acts with the effect of transcription) (C) Volcano plots of differential translation efficiency (TE) of HIV-1 infected cells compared to the uninfected sample at each time point of infection. Genes with >1 fold change in TE are colored based on gene ontology terms (biological process) grouped as umbrella terms (See also **Methods**) and genes with highest fold change in TE are marked. (Inlet) Piecharts representing the umbrella terms of differentially regulated genes at all time points. Each slice of the pie chart represents the percentage of genes in the particular term.

L104-130 how are the enrichment calculated? From one time point to another, or always in reference to the 0 time point (mock control)?

The enrichment or fold change is calculated by comparing each timepoint to the same mock sample which is 8h uninfected SupT1 cells (**Figure 1B**). We have now included a detailed explanation in the figure legend as well as the Materials and Methods.

“ Scatter-plots of the fold-change (\log_2) in cytoplasmic RNASeq and RiboSeq levels of HIV-1 infected cells compared to the mock-sample (8h uninfected cells) at each time point of infection. Only genes that were called to be significantly differentially expressed (8 hpi: n=1,254, 16 hpi: n=1,312, 24 hpi: n=2,257) are shown.”

L120-121 it is not clear to me what this sentence means.

Thank you for pointing this text deletion error, we corrected it. Revised lines 115-117 ‘Interestingly, a substantial portion, ~60% (787 genes), demonstrated changes in translation as revealed by RiboSeq (depicted in red), without any corresponding changes in RNA levels, leading to significant alterations in TE.’

L128-130 As translation is the focus of this study, it would be interesting to know which are the 14 genes involved? Why only 10 of them have been selected for TE calculation?

The translational efficiency (TE) of all genes was calculated as a part of the deltaTE pipeline analysis and is provided in the Source Data. The genes that were significantly differentially expressed are shown in the scatterplots of **Figure 1B**. The 14 genes involved in translation, identified as consistently downregulated across all timepoints, include RPS8, EEF1B2, EEF1D, RPS21, RPL22L1, RPS10, RPL34, RPL23, RPL36A, RPL35, RPL12, RPS14, RPS17, RPL11. In **Figure 1C**, we focused on highlighting a subset of genes that exhibited the most notable changes in TE within the defined umbrella terms. However, we have now mentioned this clearly in the figure legend.

“Volcano plots of differential translation efficiency (TE) of HIV-1 infected cells compared to the uninfected sample at each time point of infection. Genes with >1 fold change (\log_2) in TE are colored based on gene ontology terms (biological process) grouped as umbrella terms (See also **Methods**) and genes with >1.2 fold change and $-\log_{10}(p_{adj}) > 3.15$ in TE are labeled.”

Furthermore, we have illustrated the TE fold changes of all the translation-related genes in the volcano plots of **Supp. Figure 2A**.

For the reviewer’s convenience, we show the revised Supp. Fig. 2 (**Rebuttal Figure 3**),

Rebuttal Figure 2 (Revised Supp. figure 2): Related to Figure 2. (A) Volcano plots of differential translation efficiency (TE) of HIV-1 infected cells compared to the uninfected sample at each time point of infection. The downregulated translation-related genes are marked at each time-point. (B) Normalized ratio of area under the curve (AUC) calculated for 80S and polysome peaks of the polysome profiles as seen in Figure 2B, for each timepoint of infection. Each column averages two independent replicates, and both replicate values are shown as points. (C) Relative abundance of ACTB along the polysome profile at each time point of infection. The depicted curve represents a smoothed interpolation derived from mean values of two qPCR replicates and both replicate values are shown as points. The

dashed lines mark the 80S (9-12) and polysome (13-18) containing fractions, respectively. (D) Normalized ratio of area under the curve (AUC) calculated for 80S and polysome peaks of the relative abundance mRNA profiles as seen in Figure 2C, D and Supp. Figure 2C, for each timepoint of infection. For GAPDH, mock represents uninfected cells at each timepoint. For ACTB, mock represents 24h uninfected cells. Each bar averages two independent replicates, and both replicate values are shown as points. (E) Change in translation efficiency (calculated using RiboSeq data) of HIV genes – Gag, Pol and host genes – GAPDH, ACTB compared to 8 hours post infection.

L131-132 Are all the regulated pathways present in this table? How were the selected the GO for which the TE has been calculated?

In **Figure 1C**, we selected the top 15 gene ontology terms (by fold enrichment, hierarchy level 1) at each PANTHER GO enrichment analysis and grouped them into umbrella terms (see the comment above). The top GO terms and their grouping has been detailed in **Supp. Table 1**. The entire GO enrichment analysis can be found in the **Source Data**. We detailed the manuscript main text (L123-152).

L140-142 As far as I understand this is not what table 2 says. I see RPL22, RPS14 at 8h, RPL26, RPL13A, RPS9 at 16h and RPL13A, RPS9 at 24h

The old **Supp. Table 2** contained only the selected (colored) genes from **Figure 1**. We now removed **Supp. Table 2** and rephrased the text.

“Moreover, we observed a consistent decrease in the translational efficiency of 14 genes associated with translation, including specific elongation factors and ribosomal proteins like RPL35, RPS21, RPS9, and RPS14 across all infection stages (**Supp. Figure 2A and Source data**).”

The TE of all genes (as part of the deltaTE pipeline analysis) can be now found in the **Source Data**. We have also shown the fold change in Δ TE of all translation-related genes significantly differentially expressed in our dataset as a part of **Supp. Fig 2**.

L143-147 It is hard to evaluate these assertions without the rna-seq data. Indeed if relying only on calculated TE, about as many genes are upregulated than downregulated (72 up vs 23 down at 8h hours, 126 down vs 95 down at 16h)

As requested by the reviewer, we have now included a comprehensive list of transcriptional changes in **Supp. Table 1**. and the raw data for RNA-seq are available under [GSE244468](https://www.ncbi.nlm.nih.gov/geo/query/acc.cgi?acc=GSE244468) (reviewer token gxytgeukpzqfleb).

Additionally, we have carefully reevaluated our conclusions to better reflect our results and ensuring a more accurate representation of the gene regulation.

L149-185: I very well understand that polysome profile experiments are tricky to quantify and may not be 100% reproducible. However, in a study focusing on translational regulation this is a bit of a concern.

Consequently, several conclusions appear at least disputable:

i) It is concluded that overall the translation is inhibited at 16h and 24h, but looking at the ribosome profile I would say that there is more translation at 16h and even at 24h than at 8 hours (that cannot be explained by the 0,5% to 2,5% of HIV RNA engaged in translation at these time points as determined by Ribo-Seq).

We thank the reviewer for his thoughtful comment. We conclude that the host translation initiation is affected because (i) the infected cells consistently exhibited less material (OD600) in polysomes compared to their uninfected counterparts at all examined timepoints. (**Figure 2B and Supp. Figure 2B**), and (ii) essential house-keeping genes GAPDH and ACTB were less enriched in polysome fractions relative to the free and 80S fractions (**Figure 2C, Supp. Figure 2C and D**).

While we acknowledge an apparent increase in polysomes at 16 and 24 hours in **Figure 2B**, this likely reflects continued cellular growth over the course of the experiment, given the ~30-hour doubling time of Sup-T1 cells. To account for this, we counted the cells and use a standardized 40 million cells to perform lysis and polysome fractionation for each time point. However, despite this standardization, any person performing gradient fractionation would be aware that in practice it isn't the case, especially when it comes to time-point comparison. For this reason, we calculated the polysome/80S AUC ratio (**Sup. Figure 2B**) which should be robust to changes in cell amounts. Again, at all time-points the infected cells had relatively less polysomes compared to 80S, suggesting an impact on translation initiation.

ii) It is concluded that GAPDH (and ACTB) are actively translated at 8h, but are subsequently removed from the active polysome fraction. However, taking into account the experimental variation, the differences between profiles at different time point, or with the mock control are rather subtle.

We acknowledge that the effect may be considered subtle. However, this trend was consistently observed for the housekeeping genes across two replicates and multiple time

points. It is important to highlight that the data in **Figure 2 C, D and Supp. Figure 2C** are normalized for the total amount of RNA across the fractions, so the analysis should be robust to changes in cell amounts.

To more clearly delineate these trends, we have now introduced a smoothed curve representation of the qRT-PCR data in **Figure 2**. Based on the polysome profiles we see that the 80S corresponds to fraction 9-12 and polysomes to fractions 13-18. For the polysome profiles as well as the relative abundance profiles, we have calculated the area under the curve (AUC) of the 80S and polysome fractions using the trapezoidal rule, including the polysome/80S ratio to minimize experimental variation (**Supp. Figure 2B and D**). We see the ratio of polysome/80S decreases at 16 and 24h in the polysome profile (**Supp. Figure 2B**). Further analysis of the qRT-PCR fractions reveals a decrease in host mRNA within polysomes at these later time points. Conversely, the translation of HIV-1 mRNAs remains active throughout the infection stages, with a minor decline in partially spliced mRNA at 16 hours. These findings have now been integrated into the main manuscript (**Rebuttal Figure 3 and 4**).

Rebuttal Figure 4 (Revised Figure 2): HIV-1 infection affects host translation initiation. (A) Schematic representation of the procedure to monitor effect of HIV-1 infection on global host translation. Briefly, cytoplasmic lysates of un/infected SupT1 cells at different timepoints of infection were sedimented through a 5-45% sucrose gradients and fractions were collected while continuously monitoring absorbance at $\lambda = 260$ nm. RNA was isolated from each of these fractions for subsequent qRT-PCR of specific genes. (B) Polysome profile of infected cells compared to the Mock-sample at each time point of infection ($n=2$; representative profile of one replicate is shown). Relative abundance of (C) *GAPDH* and (D) HIV-1 RNAs along the polysome profile at each time point of infection ($n=2$). The grey line represents mock control (uninfected cells) at respective time point. The depicted curve represents a smoothed interpolation derived from mean values of two qPCR replicates and both replicate values are shown as points. **See also Supp. Figure 2.**

iii) I may have missed something but if HIV RNA represent 0.5% and 2.5% of the ribosome-associated RNA, can a “preferential translation of viral over host genes” be claimed?

We do not mean to imply that all of the cellular translation machinery is repurposed for HIV-1. Instead, we suggest that host translation is downregulated but HIV-1 RNAs are nevertheless efficiently translated.

As mentioned previously, we show this through (i) a lower polysome/80S ratio in infected cells compared to uninfected cells, (ii) a redistribution of housekeeping genes away from polysome fractions, (iii) HIV-1 RNAs predominantly in the polysome fractions at all time-points (with the exception of the unspliced RNA at 24 hrs for other reasons), (iv) increasing TE of HIV-1 genes with time by Ribo/RNA-seq (v) a decrease in TE of housekeeping genes with time by Ribo/RNA-seq. The fact that HIV-1 RNA is such a small proportion of cellular RNA does not preclude its preferential translation.

All conclusions drawn sound very reasonable, but they are hardly supported by the data. I found this chapter and the conclusions drawn very confusing, the authors need to clarify this chapter.

We thank the reviewer for their feedback. We have carefully re-evaluated and amended our initial conclusions, particularly where we previously suggested initial host response at 8 hpi is only translationally regulated. The revised section now more accurately mirrors the findings presented in our study:

L150-152: “Overall, these results indicate reprogramming of the host transcription and translation upon HIV-1 infection, particularly for genes related to cellular stress, rRNA processing and translation.

For subsequent findings, we have thoroughly re-examined our results and now provide stronger data analysis, which has strengthened our claim that HIV-1 infection notably impacts host translation initiation. To enhance the manuscript's overall clarity and provide a more precise interpretation of our findings, we have now refined the presentation of these results and their accompanying conclusions in the ms (**Revised Figure 2 and Supp. Figure 2; Rebuttal Figure 3 and 4 in this document**).

Figure 2F : indicate that the gray line is the mock control.

Thank you, done.

L224-225 The figure may be deceptive, but I see mostly signal on Rev, and not on Tat. Actually, there are no more reads on Tat than on vpu or nef (figure 3B). Please clarify

We agree with the reviewer, we have altered the text to remove the reference to Tat.

The text now reads “At 16 hpi, RiboSeq reads mostly corresponded to one of the ‘early’ HIV-1 regulatory proteins Rev, which is translated from fully-spliced HIV-1 transcripts”.

L230-232: Indeed the sharp increase on Gag is observed between 16 and 24 hours. But doesn't this come in contradiction with the sharp decrease of the % of unspliced HIV-1 RNA in the polysome fraction and its relocation to the 40S fraction between 16h and 24h as seen in figure 2G or supp 2E? Please clarify.

We note that, Figure 2D shows the *relative* distribution of unspliced RNA in polysome vs non-polysome fractions by RT-qPCR, including its redistribution. Figure 3 shows the *absolute* number of Gag RiboSeq reads increasing at 16 and 24h. Thus, there is no contradiction because the absolute increase in unspliced RNA in cells is sufficiently large to overcome a relative redistribution of unspliced into the non-polysome fraction. Furthermore, we next applied a smoothed curve to the qRT-PCR data and calculated the area under the curve for these specific fractions. Although there is a decrease in the polysome/80S ratio from 16 to 24h, we can confirm that the unspliced HIV-1 RNAs are still predominantly associated with polysomal fractions, indicating their active translation. Additionally, the ratio is higher for unspliced HIV-1 mRNA than the host mRNAs at the respective infection time-point (**Supp. Figure 2D**).

As pointed by the reviewer, indeed, between 16-24 hours, it is possible that sufficient amount of gag has been produced, as evidenced by the peaking of translational efficiency around 16hpi (**Supp. Figure 2E**). The increased availability of unspliced RNA henceforth could provide genomes for further viral assembly. This can thus lead to repartitioning to lighter fractions, as translation of unspliced RNAs is proposed to hinder viral packaging. We have now rephrased the text as follows (Revised L178-186),

“Interestingly, at 8 and 16 hpi the majority of the unspliced HIV-1 RNAs were predominately located in the polysomes, but at 24 hpi, a proportion of unspliced RNAs repartitioned into the lighter fractions (Fractions 2-8). The increased availability of untranslated unspliced RNA could provide genomes for assembly into viral particles, as active translation of unspliced RNA is proposed to inhibit viral packaging⁵⁵ (**Figure 2D** (right) and **Supp. Figure 2D**).—This observation is further supported by a slight decrease in the polysome-to-monosome ratio at 24 hpi as compared to earlier timepoints, although still remains significantly higher compared to that of host mRNAs.”

L234-293: This chapter about the identification of potential alternative translation initiation events is very interesting and uncovers (at least) to novel iORF in HIV-1. However, several things need to be clarify for an easier reading and a better understanding of how these results were obtained and their relevance.

When looking at figure 4A where are reported the RPMharr (Please define RPM), it seems that the overwhelming majority of the reads, are in the 5'UTR (on non AUG start codon), then on rev 5' exon, then in rev second exon (or env may be), then in vpu, then in the Gag coding region and finally in pol and vif. Results about the 5'UTR are not in the supplementary table; could you please provide them? Only the results about Pol and vif are discussed. The examination of the supplementary table (Please legend the supplementary table) reveals that this is because the p-value calculated by PRICE is not <0.005 for all of the other results. Please explain the calculation of this p value and why it leads to the discard of the vast majority of the results obtained.

We thank the reviewer for their thoughtful observation and careful look at our data. We did indeed select the ORFs with p-value <0.005 , ensuring these are significant iORFs separated from noise. The p-values were calculated by the PRICE pipeline. Shortly, PRICE tests whether the signal of a specific ORF candidate exceeds noise levels, by using a hypothesis test based on the generalized binomial distribution. In **Supp. Table 2** we provide the non-filtered alternative ORF predictions, where every potential candidate is listed. Further details of p-value determination can be found in the original publication.

Due to missing genome annotation, the HIV 5'UTR was initially not included in the PRICE analysis. We had manually looked at each Harrington peak in the HIV-1 5' UTR for potential uORFs. We have now corrected our PRICE analysis and included the region from 450-789 as Gag 5'UTR. Indeed, we found one uORF (749-779) with p-value <0.005 and two uORFs with p-value <0.05 in the HIV-1 5'UTR. We have updated **Figure 4B**, **Supp. Table 2** and the manuscript accordingly. However, upon reanalysis of our mass-spec data we could not find any peptide corresponding to any of the uORFs, thus further studies are required to prove its translation and role during HIV-1 infection.

Supplemental table legends are included. Sorry for the inconvenience.

Nothing is said about the results obtained on cellular mRNA. I understand that this is not the scope of this publication, but few indicators of the relevancy of the results about the cellular transcriptome could give weight to the novel findings about HIV.

The non canonical as well as canonical ORFs in the host identified by PRICE are included in **Supp. Table 2**. As suggested by the reviewer, we now include the oncogene MDM2 (**Supp. Figure 4D**), which is known to be translationally regulated by two upstream open reading frames (PMID: 12730202), both of which could be detected by PRICE (ENSG00000135679 – p-value <0.005).

L256: "Downstream" instead of "upstream"?

Yes, changed.

L266-272: My understanding from L260-263 is that the authors suggest that this iORF may be expressed from a D1-A2 splicing? In this context, the experiment performed in supp4E is confusing. If this is to suggest that an alternative translation event such as a frameshift or an internal ribosome entry may take place, it would require some additional controls.

We appreciate the reviewer's attention to detail and acknowledge the need for clarity regarding translation of the *Vif* internal open reading frame (iORF).

To investigate the translational capability of the iORF within the *Vif* gene, we designed a reporter assay using a construct with the CMV promoter, followed by a 50-nucleotide sequence from the *Vif* gene encompassing the iORF region, and succeeded by the mCherry reporter gene. To establish a baseline (0-frame control), we aligned the mCherry sequence in the same frame as the *Vif* gene, with a stop-codon in the +1 frame (corresponding to the iORF's frame), simulating conventional 0-frame translation. Conversely, for the experimental construct, mCherry was positioned in the +1 frame with stop codons in the 0-frame, thereby indicating mCherry expression as evidence of iORF translation.

We compared mCherry expression between the experimental construct and the 0-frame control to assess the iORF's translation initiation and elongation capability as compared to the conventional 0-frame translation. Given that our plasmid construct does not integrate into the genome, it precludes splicing; thus, the chosen 50-nucleotide segment serves as a hypothetical "pre-spliced snippet," representing the sequence that would ideally remain post-splicing, considering the splice acceptor site is situated 48 nucleotides upstream of the iORF start codon.

We agree that further experiments with additional controls are essential to conclusively determine the iORF's translation and its functional significance in HIV-1 regulation. The current findings provide preliminary evidence supporting the existence and possible translation of the iORF through the described mechanism, yet further research is required to fully elucidate its role. We now have shifted the *in vitro* confirmation of *Vif* iORF to **Supp. Figure 4**.

294-319: Why are only the results obtained for the FSS region discussed? Are the others not statistically relevant? Could the authors comment on the other strong disome peaks (within Gag, env or nef)?

While we indeed observed several interesting pause sites across the HIV-1 genome, our focus on the FSS (FSS) region was driven by its current active investigation within our lab and the limited time available for comprehensive analysis. The FSS element, in particular, represents a significant area of interest due to its potential regulatory roles.

We recognize the significance of disome peaks within the *Gag* and *Nef* regions, which warrant further investigation for their roles in co-translational folding as seen previously for ribosome collisions (PMID: 33402206). Notably, our use of a VSVG pseudotyped HIV-1 NL4-3 Gag-iGFP ΔEnv virus, which includes a frameshift mutation in the Env gene, led to accumulation of disomes. The functional relevance of this peak within the Env region remains uncertain. Other disome peaks in our data align with splice junctions or overlapping reading frames, which complicates their analysis due to the ambiguity regarding their specific open reading frame associations (**Rebuttal figure 5**). This complexity underscores the need for further detailed exploration to ascertain their precise roles and implications.

To ensure the manuscript reflects the presence and potential significance of these additional stacking events, we have incorporated the following statement (L 322-324):

"Several prominent disome peaks were observed in the *Gag-Pol*, and *Nef* regions, which could be crucial for the correct co-translational folding of the proteins (PMID: 33402206)"

Rebuttal figure 5: (A) DisomeSeq depth profiles of HIV-1 at 24 hours post infection. The frameshift region and the region of overlapping *Tat-Rev* reading frames are boxed. (B) Zoom into the HIV-1 frameshift region at the interface of the *gag* and *pol* genes showing the AGG disome peak. (C) Zoom into the overlapping *Tat-Rev* reading frames where the highest disome peak is boxed. (D) Close-up view of the boxed region which shows the location of splice sites.

Figure 6: The color coding of the probing results should be indicated. 6C: HIV-1 instead of SARS-CoV-2 should be replaced by HIV-1. 6D: What is the NT mutant? Indicate the delta ext mutant boundaries on 6B?

We have made all of these critical changes as requested:

- We have included a color scale We have replaced SARS-CoV-2 with HIV-1
- NT refers to 'non-targeting' ASO. The figure legend has been updated.
- We have now included this information in the figure legend.

For the reviewer's convenience, the updated figure and figure legend is below:

Rebuttal Figure 6 (Revised Figure 6): **Experimental and functional analysis of extended HIV-1 frameshift site.** (A) Translational landscape surrounding the HIV-1 frameshift site (FS) at 24 hours post infection (mean of both replicates). P-site codons were predicted using PRICE analysis. P-site codon bars are colored according to their relative frame (Gag ORF is in the 0 frame). Codons of the highest peak upstream of the SS (AGG) and in the SS (UUA) are denoted. Sequence logo of 4903 HIV

sequences is shown with the amino acids sequence associated to the 0 frame. Below these sequences, the potential alternative frameshift SS (SS*), the -1 frame stop codons and the canonical SS are shown in blue, yellow and red boxes, respectively. (B) RNA structure model of the region surrounding the HIV-1 FS, from nucleotides 1551 to 1741, as predicted by DMS data. The DMS reactive nucleotides are colored red while unreactive are blue. Gray nucleotides correspond to nucleotides without DMS data. The ASO1 binding site is shown. (C) Schematic representation of the dual-fluorescence frameshift reporter constructs and quantification of the relative frameshift efficiency (FE) of different HIV-1 FS mutants. EGFP and mCherry are separated by both a self-cleaving 2A peptide and a stop codon in frame with the EGFP. As a result, 0 frame translation produces only EGFP, whereas -1FS produces both EGFP and mCherry. The ratio of mCherry to EGFP fluorescence is used to quantify the FE. Each mutant characteristic is highlighted below the bar graph. Included is a control where both SS and SS* are mutated to define the baseline mCherry signal, with subsequent data presented adjusted for this baseline. HIV-1 Δ ext corresponds to the structure containing the nucleotides 1629 to 1672. The non-targeting (NT) antisense oligonucleotide (ASO) and ASO1 were used against the WT HIV-1 FS. Data point represent the mean \pm s.d. (n = 3 independent experiments). P values were calculated using an ordinary unpaired one-sided ANOVA comparing every mutant FE to the WT.

The DMS probing data actually poorly supports at least three of the pairings proposed
Supp figure 7:

We have provided supporting evidence from DMS probing and ASO experiments that the frameshift element of the HIV-1 is more complex than the previously described hairpin. Our predicted structure is a three-way junction with a large central bulge. Whilst the canonical frameshift hairpin and the base of the extended frameshift are well supported by DMS and functional data by ASO, we agree that not all stems are well supported by the DMS probing data.

We have now clarified this point by adjusting the text (L364-372):

“According to our DMS probing in cells, this extended RNA structure is predicted to fold in a three-way junction with a large central bulge. We find good support for the anchoring stem (nucleotides 2022-2031 base paired to nucleotides 2152-2161) and the conventional FSS (2099-2110 base paired to nucleotides 2115-2126). However, neither the alternate lower stem (2037-2039 base paired to 2146-2148) described by Low et al, 2014 (Ref 26), nor the lower FSS (2092-2096 base paired to 2130-2134) are well supported, indicating that these stems may not be folded in the majority of molecules in their native, dynamic state in the cell (Figure 6B).”

It is not clear what is the difference between “contour length changes” and “total contour length changes”

We apologize for the unclear description. The “total contour length” refers to the size of the RNA at its maximal possible extension (i.e. when completely unfolded and under tension, like a string). The “contour length changes” refers to the difference in size from one structure to the other one. The total contour length change refers to difference in size from the **F**olded state to the **U**nfolded state. Here, for an RNA of ~ 140 nts, we expect a total contour length change of ~82 nm if both extremities are base-paired. This value can decrease if some nucleotides in the extremities are unpaired (if different RNA folds exist, for example).

L488-489: As far as I understand it “ASO targeting of the lower stem” slightly decrease, rather than enhance the frameshifting efficiency?

We apologize for the mistake. As stated in the results, antisense oligonucleotide targeting decreased frameshifting efficiency by 40%.

The corrected manuscript now reads “ASO-targeting of the lower stem of the alternative fold of the HIV-1 RNA decreased efficiency of frameshifting, which also points to a functional role for the upstream RNA of the canonical FSS.”

L615-616 I have not found any of those, could you please indicate where they can be found

Due to a delay with getting a GEO accession number, this could not be provided at the initial submission. The data can now be found under GSE244468 with the GEO token gxytgeukpzqfleb.

Reviewer #2:

Remarks to the Author:

This manuscript reports on a timecourse of RiboSeq translational profiling of HIV-1 infected T cells. Infection causes early translational changes in host genes, accompanied by later changes in mRNA abundance. Viral RNAs remain highly translated as host translation declines, and uncharacterized viral ORFs are described. These include uORFs as well as internal ORFs resulting from non-canonical splicing events. Ribosome occupancy around the HIV-1 frameshift shows unexpected patterns, including ribosome accumulation and stacking upstream of the frameshift site itself. Ribosome stacking leads to the proposal of an extended

structural element around the frameshift site, supported by DMS probing in cells and by single-molecule folding trajectories.

This manuscript provides a valuable resource for the study of HIV, and notably proposes a significant addition to our understanding of the HIV-1 frameshift. The genomic data are high quality and supported by validation data, for example by independent proteomic validation of iORF-encoded peptides as well as reporter constructs. While data presented are strong, additional tests would strengthen the argument for the extended frameshift structure—one of the most substantial and exciting findings in this work. Certain other points seem to over-interpret the data presented. I would support publication of a suitably revised version of the manuscript addressing the concerns below.

1. Host cell translational profiling identifies changes in ribosomal protein translation. Such changes are often indicative of changes in 5' TOP mRNA translation downstream of changes in mTOR activity. It would be valuable to look at this specifically for a well-established collection of 5' TOP transcripts; either gathering evidence for such a change, or refuting it, would be interesting.

We thank the reviewer for their thoughtful suggestion. We looked at the translation of mRNAs with 5' TOP motifs namely ribosomal proteins, *EEF1D*, and *EEF1B2*. (PMID: 9023110). Interestingly, our analysis of the translational efficiencies of these mRNAs revealed a general decrease upon HIV-1 infection, suggesting an impact of the infection on 5' TOP motif translation (**Rebuttal Figure 3**). We then checked the expression of LARP1, which is a key protein involved in mTOR mediated regulation of TOP mRNAs (PMID: 33398329). However, the expression of LARP1 was not significantly changed upon infection in our dataset, albeit a slight downregulation in both RNA- and RiboSeq at 24h, indicating that the modulation of TOP mRNA translation is likely not due to changes in expression of LARP1.

Upon examining mTOR pathway regulators from the dataset available on the Harmonizome database, we observed slight expression changes in the early phase of infection (8 hpi). While mRNAs of some mTOR activators such as *PRKCA*, *RAG1&2* and *RICTOR* showed increased expression, *MTOR* and other effectors experienced a decrease. Given the absence of the HIV-1 env protein, which is known to activate mTORC1, our findings do not definitively establish the direction of mTOR pathway modulation by HIV-1 (PMID: 32354054).

Rebuttal Figure 7: Scatterplots of the fold-change (\log_2) in cytoplasmic RNASeq and RiboSeq levels of HIV-1 infected cells compared to the mock-sample (8h uninfected cells) at each time point of infection. Only genes that were called to be significantly differentially expressed are shown. Genes involved in the mTOR pathway identified by the Harmonizome database are highlighted.

2. The sucrose density gradients in Figure 2 are over-interpreted, with too little consideration of alternate explanations. For instance, in Fig 2F, the shift of GAPDH into the heaviest fractions is interpreted to “represent stress granules” (l. 169) whereas a rather similar shift for fully-spliced HIV-1 transcripts are interpreted to indicate that they are “actively translated”. If similar data are consistent with either repression (Fig 2F) or active translation (Fig 2G), then the data don’t support one or the other alternative. Translation shutoff experiments could resolve this question, but as this does not seem to be a central point for the paper, simply providing a more consistent interpretation would address this concern.

We appreciate the reviewer's critical evaluation and acknowledge the importance of a consistent interpretation of the data.

In addressing the shift observed for GAPDH mRNA and fully-spliced HIV transcripts, it is important to note that the fully-spliced HIV transcript predominantly shifts within the polysomal fractions (14-18), suggesting active translation. In contrast, the GAPDH mRNA shifts from polysomal fractions (16-18) to heavier fractions beyond 18, which could indicate engagement with non-polysomal complexes, potentially stress granules. However, we agree with the reviewer that our initial interpretation was speculative and required further evidence.

To clarify and ensure the scientific accuracy of our findings, we have omitted data beyond fraction 18 to avoid over-interpretation. Additionally, we have refined our analysis of the effect of HIV-1 on host translation initiation, incorporating a smoothed curve representation of the qRT-PCR data in Rebuttal Figures 3C and 4C,D. The area under the curve (AUC) for 80S and polysome fractions, calculated using the trapezoidal rule, and the polysome/80S ratio have been included to reduce experimental variability. This clarified analysis shows a discernible

trend of decreasing host mRNA ratios in polysomes as infection progresses. Conversely, the fully-spliced HIV-1 mRNAs remain predominantly within polysomes across all time points, while partially-spliced mRNAs exhibit variable translation activity, we see active polysomal translation at 8hpi, a slight drop in the ratio at 16h which bounces back at 24hpi. For unspliced HIV-1 mRNAs, a shift towards lighter fractions is observed at 24hpi, but active translation is still predominant in the polysomes as compared to the monosomes at all time points.

In summary, we have enhanced the data analysis and presentation to better support our conclusions and provide a more precise interpretation in line with the reviewer's feedback.

The modified results can be found above in **Figure 2 (Rebuttal Figure 4) and Supp. Figure 2 (Rebuttal Figure 3)**.

3. A few related concerns arise about the analysis of pausing and codon usage in Supp Figure 3. First, cell lysates were prepared after chilling cells to 4 °C for 3 minutes and exposing them to cycloheximide, which seems likely to distort measured ribosome positions. Second, the analysis focuses on extreme (> 2 standard deviation) pauses, but the discussion focuses on changes in tRNA pools, which are expected to produce broad shifts in per-codon occupancy rather than extreme pauses. My sense is that the scope of conclusions here are limited in any case.

We acknowledge the concern regarding the cycloheximide treatment. It is important to note cycloheximide was added to media and cells were incubated at 37°C for 5 min, followed by rapid pelleting at 10,000 RPM at 4°C for 3 minutes, washed with ice-cold PBS and flash-frozen. Cells were then lysed with lysis buffer on ice, consistent with conventional ribosome profiling procedures. This method should ideally avoid cycloheximide induced distortions (PMID: 34429433). Furthermore, our read length and frame distribution as well as number of reads within coding transcripts were all in good agreement.

In Suppl. Fig 3, our aim was to check for specific stalling patterns, as performed in other studies (PMID: 34056595). One of the interesting findings of this analysis is the isoleucine specific stalling we observe at the P site. We think that this could be of interest for the HIV-1 and translation field, hence we are eager to keep the data in the supplementals. However, we agree with the reviewer and have removed the sentence linking the A-site stalling events in our study to possible presence of rare codons or to limitations in the amino acid pools and tRNA availability, to avoid over interpretation.

4. Viral gene expression measurements in Figure 3 rely on deconvolution of the different splice isoforms. Is it possible to use RSEM or similar tools to carry out more robust deconvolution? Due to the short length of ribosome profiling reads it is not possible to assign the vast majority of reads to specific isoforms via sequence alone. However, PRICE analysis allowed us to assign reads to specific HIV-1 genes based on frame information.

5. 54 nt seems too short for a footprint from a proper disome. Even in yeast, where footprints are slightly shorter, 54 nt footprints are attributed to quality control intermediates whereas proper disomes are more than 60 nt [10.1016/j.cell.2014.02.006].

Although, our major peak was ranging from 59-64nt, we do observe a shoulder at around 54-57nt. Recent human disome profiling studies with P1 nuclease identified presence of ~50nt disome protected fragments (45-60nt) that they classified at sub-disomes, whereas the true disomes are >60nt (PMID: 37783882). Previously, it was shown that RNase I digestion can produce both a ~58 nt true disome RPF and a shorter 3' truncated ~51 nt RPF with cleavage at the unoccupied A-site of the leading stalled ribosome (PMID: 32615089). These studies could explain the presence of the shoulder at 54-57nt.

6. It would be valuable to present the length profile of disome footprints at and around the programmed frameshift relative to the overall length distribution.

We thank the reviewer for this interesting point. We carefully analyzed our disome peak at and around the HIV-1 FSS (below, **Rebuttal Figure 7**). Before the FSS the majority of the disomes lie in the 60-64nt range, which aligns well with closely stacked ribosomes (**Rebuttal Figure 7C left**). At the FSS, the majority of disome reads remain within the 60-64 nucleotide window, however we see a fraction of shorter footprints with a median length of about 55-57 nt, which could be attributed to quality control intermediates as also suggested by the reviewer in point 5 PMID: 24581494 (**Rebuttal Figure 7C right**). The relatively lower number of reads at the FSS suggests that ribosomes may not be stacking extensively (consistent Riboseq reads at these positions). We have now included this read length distribution analysis in our **Supp. Figure 6**.

Rebuttal Figure 7 (Revised Supp. Figure 6): (A) Scatterplots showing the correlation and reproducibility between duplicates of DisomeSeq at each timepoint of infection. Reads per million (RPM), as assigned to individual genomic features by featureCounts. Number of reads (n), significance (*p* value) and Pearson correlation coefficients (R) are indicated for each scatterplot. Lighter colors indicate higher point density. Read length distribution of transcripts reads obtained from disome profiling libraries prepared from SupT1 cell lysates for (B) entire HIV-1 genome and (C) reads ending before FSS (1990-2050) and reads starting at FSS (2050-2110). (D) Schematics of the N-terminal FLAG-tagged frameshifting reporter consisting of the nucleotides 64 – 2687 (Δ 1870 - 1881) of the HIV-1 genome (up). Time course of translation of reporter mRNA and controls in RRL (bottom). Briefly, *in vitro* transcribed reporter RNA was translated in RRL, after 3 min further initiation was halted by the addition of harringtonine, and aliquots were removed at various times and analyzed by SDS-PAGE. Size marker RNAs were similarly translated and analyzed to show the expected size of the ribosomal pause product

(See also **Methods**). (E) The % product was quantified from the western blot and is plotted against time.

7. In discussing Supp Figure 6B-C, it is written that “a persistent translational pause” suggests “potential ribosome drop-off”. I think it is meant that a persistent protein product at this position suggests drop-off? This would seem to better explain the profiling and in vitro translation data.

We thank the reviewer and the text is changed as suggested.

8. In discussing frameshift reporters (ll. 394-395), a control is described that has both the canonical slippery site (SS) and the new candidate slippery site (SS*) mutated, but no data from this control is presented. It seems important to present these data, which would indicate the level of “background mCherry signal” present in the absence of any slippery sites.

We have indeed used a control in which both the canonical slippery site (SS) and the new candidate slippery site (SS*) are mutated to establish a baseline for background mCherry fluorescence. For convenience, the results presented in Figure 6D have been adjusted for mCherry background signal. We have also incorporated another control that includes both slippery sites (SS*/SS) with 0-frame stop codons ahead of mCherry in-frame in order to measure the background mCherry production. To improve clarity, we have now explicitly mentioned the background controls within the main text and the corresponding figure legend (L386-387 and **Methods**). **Rebuttal Figure 8** displays the frameshift efficiency data prior to background subtraction.

Rebuttal figure 8: Schematic representation of the dual-fluorescence frameshift reporter construct and quantification of the relative frameshift efficiency (FE) of different HIV-1 FS mutants. EGFP and mCherry are separated by both a self-cleaving 2A peptide and a stop codon in frame with the EGFP. As a result, 0 frame translation produces only EGFP, whereas -1FS produces both EGFP and mCherry. The ratio of mCherry to EGFP fluorescence is used to quantify the FE. Each mutant characteristics are highlighted below the bar graph. The non-targeting (NT) antisense oligonucleotide (ASO) and ASO1 were used against the WT HIV-1 FS. Data point represent the mean \pm s.d. ($n=3$ independent experiments). *P* values were calculated using an ordinary unpaired one-sided ANOVA comparing every mutant FE to the WT.

9. No non-targeting ASO control is presented for the data arguing that an ASO can disrupt frameshifting by disrupting the extended structural element.

We apologize for the confusion. The NT in Fig. 6D represents the non-targeting control. We have now included this description in the figure legend.

10. Likewise, some minimal mutational analysis of the proposed novel stem-loop would be valuable; compensatory mutations could provide strong evidence for a functionally important structure and would also produce an ASO-resistant variant.

We appreciate the reviewer's suggestion for further mutational analysis of the novel stem-loop structure. We have indeed conducted preliminary mutational studies with optical tweezers and

DMS probing, which contribute valuable insights into the alternative structures of the HIV-1 FSS. In Rebuttal figure 9, we show the optical tweezer analysis of the Δ ext mutant or minimal frameshift stimulatory structure (nucleotides 2083 to 2126 from Figure 6B), which confirms the presence of an extended alternative RNA fold. However, we believe that the full scope and implications of these findings warrant a separate manuscript that is currently in preparation. Therefore, we have decided to focus this manuscript on the current results, providing functional proof for the existence of the extended frameshift stimulatory structure.

11. As a minor point, the text (ll. 120-121) reads "...prominent changes in RNAseq read without changes." Presumably this is an editing error and it is intended to claim, roughly that there are no RiboSeq changes beyond what can be explained by RNAseq? Thank you for pointing this text deletion error, we corrected it.

'At 16 and 24 hpi, we observed prominent changes in RNAseq reads without corresponding changes in RiboSeq reads.'

Reviewer #3:

Remarks to the Author:

The study entitled "The translational landscape of HIV-1 infected cells reveals novel gene regulatory principles" involves several sequencing and biochemical/biophysical techniques to investigate HIV-1 frameshifting both in vivo and in vitro. There are several interesting aspects of this study. Among them are the modulation of the transcriptome over time by HIV-1 infection and the potential fine tuning of frame shifting efficiencies by an upstream structural element. The combination of experimental approaches is very interesting and provides valuable insight on the role of -1 frame shifting in the life cycle of HIV-1. Overall, there is an appropriate analysis of the resultant sequencing data along with speculation founded in prior work that provides strength to their conclusions. I was somewhat confused during the presentations of results as it related to the extended structural frameshift site. They mention previous work as it related to their DMS probing, but I feel the discussion of the differences was insufficient. In terms of the optical tweezers experiments, I do not find these were well described. **In the figure legend they mention SARS-CoV-2, I missed that part in the paper where they were looking at that.** They need to be more specific about what region of the HIV-1 sequence they put between the handles. This wasn't clear. The assignment of the higher force unfolding event could easily be performed by unfolding the hairpin alone. This wasn't done. The figure is very small and it is challenging to observe the I to U transition. The authors discuss the hysteresis in the first unfolding transition, but it is hard to evaluate that as compared to the second unfolding

transition. The numbers presented show similar error between the two transition steps. Why is one exhibiting more hysteresis than the other? Overall, I don't find the results and discussion on this second part related to the extended frame shifting site as well reasoned or compelling as the first section. I am not convinced the optical tweezers experiments are in complete agreement with the proposed extended sequence structure.

We acknowledge the reviewers' insights regarding the complexity of the optical tweezers' analyses and need for additional mutational studies. We agree with the reviewer that the optical tweezers graph is confusing without additional analysis including other HIV-1 RNA variants. Yet, hope they would also appreciate that structure is only one of the aspects of this comprehensive work on HIV translation. Our conclusions on the alternative RNA fold and its function is well supported by the other DMS-Seq and cellular reporter analysis.

We think that extensive single molecule analysis of this RNA element will be an important aspect that we would like to further investigate in the future with other RNA variants. However, such extensive work goes beyond the scope of the present work and would be more suitable to follow up properly in an independent manuscript.

To demonstrate the rigor of our OT findings, we have prepared the response below to address the issues raised by the reviewer. We hope it will bring clarity.

Supp Figure 7: related to Figure 6. Distributions of the forces (A and C) and contour length change (B and D) of the minimal (A and B) and extended (C and D) HIV-1 WT FSS. The unfolding and refolding distributions are colored in red and blue, respectively. The force (pN)

and contour length distributions indicate the force upon which an unfolding or refolding event happens with its associated extension.

Rebuttal Figure 9: Optical tweezers experiment with the WT HIV-1 FSS and Δ_{ext} mutant (minimal frameshift stimulatory element)(nucleotides 2083 to 2126 in Figure 6B). Red curves correspond to the unfolding, blue curves to the refolding. Both the refolding and unfolding events happen at ~ 18 pN, with little to no hysteresis.

“In terms of the optical tweezers experiments, I do not find these were well described. In the figure legend they mention SARS-CoV-2, I missed that part in the paper where they were looking at that. They need to be more specific about what region of the HIV-1 sequence they put between the handles. This wasn't clear. The assignment of the higher force unfolding event could easily be performed by unfolding the hairpin alone. This wasn't done.”

In the single molecule measurements performed, we used an HIV-1 sequence encompassing the nucleotides 2022 to 2161 on the genome of reference “GenBank: K03455.1”. This sequence is expected to fold in a similar way as the one presented by Mouzakis *et al*, 2013 (PMID: 23248007). Our own chemical probing experiments allowed us to model an RNA structure for the Frameshift element (**Figure 6B**) and we tried to confirm this model through optical tweezers measurement, but also to assign forces required to unfold each of the stem present in our model, as the stability of these stems might be important for the frameshifting mechanism.

Regarding the force assignment, the FSS stem-loop alone was studied in optical tweezers by Ritchie *et al*, 2017 (PMID: 28522581). They found that the FSS stem-loop unfold/refolds at ~ 18 pN (Figure 1C of their paper). Despite differences between our system and theirs (buffer

composition and longer handles), we have also observed an unfolding/refolding similar to the one they showed for the minimal FSS, with a similar distance/contour length change.

We also performed optical tweezers experiments of the minimal FSS (**Rebuttal Figure 9**). The results indicated that the minimal FSS unfolds and refolds at the same force (~ 18 pN) and induces a contour length change of ~ 12 nm. The contour length changes are consistent with the opening/refolding of a structure of ~ 20 nts, which is the expected size of the FSS.

Regarding the extended FSS, both the DMS-Map data and the optical tweezers experiments are consistent with the presence of the minimal FSS structure in the extended FSS. However, in the extended FSS, the minimal FSS is not the sole unfolding event and happens last (still ~ 18 pN, but ~ 17 nm), as the rest of the structure needs to unfold first. This happens at ~ 10 pN and induces a contour length change of ~ 65 nm (or 110 nts).

Overall, our optical tweezers experiment combined with the chemical probing data highlighted the presence of at least two structural elements in our extended FSS model. Upon exerting tension on the RNA, the anchoring stem will open at ~ 10 pN, inducing the opening of the slippery sequence stem-loop at once, as this stem-loop seems way less stable than the anchoring and the minimal FSS. If the force exerted on the RNA structure increases, it triggers the unwinding of the most stable stem-loop in this element, the FSS stem-loop. An hysteresis is observed during the refolding of the slippery sequence and anchoring stem-loops, as expected for a complex fold such as a 3-way junction.

Reviewer #4:

Remarks to the Author:

In this study, Kibe et al. investigate HIV-1 frameshifting by combining multiple sequencing and biochemical techniques. Following reviewer #3 comments, I will focus my review on the single-molecule studies. In the first version of the manuscript, the authors employed single-molecule optical tweezers to understand the structural basis of the stalling event. Using DSS, they predicted that the HIV-1 RNA would fold as a three-way junction, including a GC-rich frameshift stem-loop (Fig. 6B). They used optical tweezers to pull on the frameshift site of HIV-1, which exhibited two distinct unfolding events: 1) a first low force ~ 10 pN event corresponding to a contour length change of ~ 64.5 nm; 2) a second higher force ~ 18 pN event involving a ~ 17 nm contour length change. Based, on these unfolding events, and the fact that the low force one showed a large unfolding/refolding hysteresis, the authors argued that the first event corresponded to the unfolding of the sequence encompassing the bulge (which has a more complex fold), while the high force event to the unfolding of the frameshift hairpin (which is rich in GC pairing, and therefore more stable).

Reviewer #3 raised several concerns regarding the clarity of the results and their interpretation, with which I agree. In particular, reviewer #3 suggested pulling on the frameshift stem-loop alone, to unambiguously assign that high-force event in the full-length HIV-1 RNA to the unfolding of the GC-rich hairpin. Of note, the nanomechanics of the frameshift stem loop were previously investigated using optical tweezers (Ritchie et al RNA, 2017). The authors have included two figures in their reply to reviewer #3, including new experiments using the frameshift hairpin. However, these new data do not fully agree with the authors' interpretation of the unfolding events observed for the full-length HIV-1 RNA—the high-force event in the full-length molecule is significantly longer than that of the isolated frameshift hairpin. Moreover, intriguingly, all data and discussion regarding the single-molecule experiments (former Fig. 6C, and Supp. Fig. 7) have now been removed from the manuscript and are only discussed in the rebuttal letter. While I understand that these data are not a central aspect of the study, the structural determination of the frameshift site is important for the conclusions of the manuscript, and the optical tweezers experiments can indeed provide relevant insights in this regard, which are currently inconsistent. Therefore, I believe the authors should provide a convincing explanation of the two unfolding events they observed in their initial optical tweezers experiments, which currently cannot be assigned to the unfolding of the frameshift hairpin, and the remainder of the molecule.

In the new experiments using the isolated frameshift stem-loop, the authors reproduce the results from Ritchie et al, observing a single unfolding event at ~ 17 pN corresponding to the unraveling of a contour length of ~ 12 nm, or 26 nucleotides, assuming ~ 0.45 nm per nucleotide—which is the standard value for RNA. This agrees with the unfolding of the GC-rich upper stem, as reported previously by Ritchie et al. However, the high force event in their experiments using the full-length HIV-1 RNA renders a contour length increase of ~ 17 nm, and therefore ~ 37 nucleotides, a significantly longer structure than that of the upper stem alone (the authors previously incorrectly assigned this contour length increase to the unfolding of 26 nucleotides, but that is not possible). The authors need to explain the origin of this discrepancy as the high force event in the full-length RNA needs to necessarily involve a larger structure, or perhaps even a more complex one. Interestingly, Ritchie et al reported that a small fraction of their curves ($\sim 0.1\%$) involved a contour length increase of ~ 17 nm (although occurring at a much higher force, ~ 30 pN), likely due to the formation of a tertiary structure involving the lower stem region of the hairpin. In the context of the full-length RNA molecule, this structure or a

similar one could form more easily, perhaps due to the flexibility of the adjacent region of the bulge, and account for this extra contour length. The authors could demonstrate this by pulling on a larger region of the frameshift hairpin, aiming to obtain a single ~17 nm event that matches the high force event observed in the full-length molecule. Similarly, the authors could remove the frameshift stem-loop from their RNA molecule (as it seems to unfold separately from the rest of the molecule, it should not be a problem), and investigate whether the unfolding/folding of this region of the HIV-1 RNA matches the low force event observed in their initial experiments. This way, the authors could convincingly assign each of the unfolding events to well-defined elements within their proposed structure.

Response:

We thank reviewer #4 for their constructive comments regarding the optical tweezers analysis. Based on their comments, we provide a detailed comparison of our results to the Ritchie *et al.* optical tweezers study of the HIV FSE. We also present some preliminary data with an antisense oligonucleotide (ASO FSE) to disrupt the frameshift stem-loop. It is important to note that a ribosomal pause was initially observed in our ribosome profiling analysis, which informed our structure predictions based on *in vivo* DMS-Seq reactivity profiles. Since the DMS analysis was performed in infected cells, it provides the most accurate information on the viral RNA fold as it exists during viral replication. Furthermore, the predicted structure's function was validated by dual-fluorescence frameshifting assays in cells. In our original study, we provided additional information of the minimal frameshift element (FSE_{min}) and extended folds (FSE_{ext}) *in vitro* using force spectroscopy. This was removed in the revised version as we now intend to provide a more complete follow-up that is beyond the scope of the current study. We wish to emphasize that the results and the conclusions of our study do not depend on *in vitro* force spectroscopy analysis.

In our initial reply to Reviewer #3, we presented force spectroscopy data for both the FSE_{min} and FSE_{ext}. The respective structures are represented in the **Rebuttal Figure 1**, alongside the RNA secondary structure model presented by Ritchie *et al.*, 2017.

Rebuttal Figure 1: Schematic representation of the three RNA structure constructs, (A) the minimal frameshift element used in our study (FSE_{min}), (B) the frameshift element studied in Ritchie *et al.*, and (C) the extended frameshift structure (FSE_{ext}). The FSE stem-loop in red corresponds to the canonical FSE, the yellow sequence corresponds to the slippery sequence and the ASO targeting the canonical FSE stem-loop is shown in green.

We pulled on the RNA sequences corresponding to FSE_{min} and FSE_{ext} using DNA/RNA hybrid handles and obtained force-distance curves (**Rebuttal Figure 2A**). For FSE_{min}, we observed a single unfolding and refolding event at a force of ~ 18 pN (17.8 ± 1.5 pN), with a contour length change of ~ 12 nm (12.4 ± 4.2 nm), consistent with Ritchie *et al.* (**Rebuttal Figure 2B**). For the extended FSE, as presented previously, two unfolding events were detected: a low-force/high contour length event and a high-force/low contour length event. The high-force unfolding event

of this FSE_{ext} is similar to the single unfolding/refolding event of the FSE_{min} (Force 17.6 ± 1.2 pN vs 17.8 ± 1.5 pN), but with a higher contour length (17.2 ± 5.9 nm vs 12.4 ± 4.2 nm) (**Rebuttal figure 3A**). The higher contour length in FSE_{ext} compared to FSE_{min} likely arises from the addition 7bp basal portion of the FSE_{ext} , which cannot form in FSE_{min} .

Rebuttal Figure 2: (A) Schematic of the optical tweezers construct with the DNA/RNA hybrid between two polystyrene beads. (B) Histograms of the force distribution (left panels) and contour length change (middle panels) for the unfolding (red) and refolding (blue) steps observed in for the HIV-1 FSE_{min} with (bottom panels) or without (top panels) antisense oligonucleotides targeting the canonical FSE stem-loop. The right panels represent examples of refolding and unfolding force-distance curves. The refolding and unfolding curves are shifted in the X axis for clarity.

To unambiguously assign this high-force event in the HIV-1 FSE_{ext} to the FSE stem loop, we used an antisense oligonucleotide (ASO FSE: 5'-TTC CCT GGC CTT CCC-3') targeting the 3' part of the structure (**Rebuttal Figure 1**, in green). Incubation of the FSE_{min} RNA/DNA hybrid with 2000-fold excess ASO FSE, confirmed correct targeting of the FSE minimal stem-loop: the force event at ~ 18 pN disappeared in the presence of the ASO, replaced by folding at ~ 12 pN, indicating impaired folding of the canonical FSE (**Rebuttal Figure 2C**). Some refolding curves do not show any detectable event. In FSE_{ext} , 90.5% of curves showed a single step with ASO versus 4.2% without it. Furthermore the disappearance of the high force event, confirmed the 17 pN event corresponds to the same RNA element in both constructs (**Rebuttal Figure 3B**).

Rebuttal Figure 3: Histograms of the force distribution (left panels) and contour length change (middle panels) for the unfolding (red) and refolding (blue) steps observed in for the HIV-1 FSE_{ext} with (bottom panels) or without (top panels) antisense oligonucleotides targeting the canonical FSE stem-loop. The right panels represent examples of refolding and unfolding force-distance curves. The refolding and unfolding curves are shifted in the X axis for clarity.

In our calculation of the contour length, we used 0.59 nm per base as the standard value for RNA, which is the same value used by Ritchie *et al.* in their calculations (Materials and Methods, FEC analysis, page 1382). Accordingly, the results of FSE_{min} agree with the results of the previous study by Ritchie *et al.* Our FSE_{ext} construct showed identical force but increased contour length, indicating the extended RNA to have a longer stem length, however we do not observe pseudoknot folding (~ 30 pN in Ritchie *et al.*). In vitro, the difference between the optical tweezers constructs likely arises from the structural context around FSE in FSE_{ext} . Here, we employed a much longer construct than Ritchie *et al.* While the standard deviation for FSE_{ext} contour length overlaps with Ritchie *et al.*'s values for the FSE, the large variance suggests the need for further detailed investigation into the structural features of FSE_{ext} , which we are currently doing as part of a separate upcoming study.

To calculate the size of the structure, we need to take a few parameters into account:

- most dsRNA molecules adopt an “A-form” double helix structure, with ~ 2.6 nm (d_T) helix diameter (Arias-Gonzales, J. (2014), *Single-molecule portrait of DNA and RNA double helices. Integr. Biol.*, 6(10), 904-925. DOI:10.1039/C4IB00163J)
- while using force, the distance between 2 stacked-bases is ~ 0.59 nm (Lc^{nt})
- the minimal theoretical stem-loop must comprise at least 2 nucleotides involved in base-pairing and 3 nucleotides in the loop.

In the study by Ritchie *et al.*, the authors calculated the expected contour length change of their structure with the formula: $\Delta Lc = Lc^{nt} \times n_{nt} \div d_T$

with Lc^{nt} being the expected distance between two stretched nucleotides, n_{nt} the number of nucleotides in the structure and d_T the initial distance between the two closing nucleotides of the structure.

According to their formula, the expected ΔLc for our FSE_{min} is: $0.59 * 26 - 2.6 = 12.7$ nm, closely matching what we observed. For FSE_{ext}, with 136 nucleotides, the expected ΔLc is 77 nm, consistent with the observed sum of two unfolding steps (64.5 nm + 17 nm = 81 nm).

In summary, our additional data confirm that the high-force event in FSE_{ext} corresponds to the FSE stem-loop, with observed discrepancies in the FSE_{ext} likely due to structural context differences around the FSE. These structural complexities warrant further mutational studies, which we believe merit a separate, more focused investigation.